# Fundamental Theory of Torsion Gravity

Luca Fabbri 

DIME, Metodi e Modelli Matematici, Università di Genova, Via all'Opera Pia 15, 16145 Genova, Italy;
fabbri@dime.unige.it

**Abstract:** In this work, we present the general differential geometry of a background in which the space–time has both torsion and curvature with internal symmetries being described by gauge fields, and that is equipped to couple spinorial matter fields having spin and energy as well as gauge currents: torsion will turn out to be equivalent to an axial-vector massive Proca field and, because the spinor can be decomposed in its two chiral projections, torsion can be thought as the mediator that keeps spinors in stable configurations; we will justify this claim by studying some limiting situations. We will then proceed with a second chapter, where the material presented in the first chapter will be applied to specific systems in order to solve problems that seems to affect theories without torsion: hence the problem of gravitational singularity formation and positivity of the energy are the most important, and they will also lead the way for a discussion about the Pauli exclusion principle and the concept of macroscopic approximation. In a third and final chapter, we are going to investigate, in the light of torsion dynamics, some of the open problems in the standard models of particles and cosmology which would not be easily solvable otherwise.

**Keywords:** torsion-gravity; electrodynamics; spinors

## 1. Introduction

In fundamental theoretical physics, there are a number of principles that are assumed, and, among them, one of the most important is the principle of covariance, stating that the form of physical laws must be independent from the coordinate system employed to write them. Covariance is mathematically translated into the instruction that such physical laws have to be written in tensorial forms.

On the other hand, because physical laws describe the shape and evolution of fields, differential operators must be used; because of covariance, all derivatives in the field equations have to be covariant: thus covariant derivatives must be defined. In its most general form, the covariant derivative of, say, a vector, is given by

$$D_\alpha V^\nu = \partial_\alpha V^\nu + V^\sigma \Gamma^\nu_{\sigma\alpha}$$

where the object $\Gamma^\alpha_{\nu\sigma}$ is called connection, it is defined in terms of the transformation law needed for the derivative to be fully covariant, and it has three indices: the upper index and the lower index on the left are the indices involved in the shuffling of the components of the vector, whereas the lower index on the right is the index related to the coordinate with respect to which the derivative is calculated eventually. Hence, there appears to be a clear distinction in the roles played by the left and the right of the lower indices, and therefore the connection cannot be taken to have any kind of symmetry property for indices transposition involving the two lower indices at all.

The fact that, in the most general case, the connection has no specific symmetry implies that the antisymmetric part of the connection is not zero, and it turns out to be a tensor: this is what is known as torsion tensor.

The circumstance for which the torsion tensor is not zero does not follow from arguments of generality alone, but also from explicit examples: for instance, torsion does

describe some essential properties of Lie groups, as it was discussed by Cartan [1–4]. Cartan has been the first who pioneered into the study of torsion, and this is the reason why today torsion is also known as the Cartan tensor.

When back at the end of the 19th century, Ricci-Curbastro and Levi–Civita developed absolute differential calculus, or tensor calculus, they did it by assuming zero torsion to simplify computations, and the geometry they eventually obtained was entirely based on the existence of a Riemann metric: this is what we call Riemann geometry. Nothing in this geometry is spoiled by letting torsion take its place in it, the only difference being that now the metric would be accompanied by torsion as the fundamental objects of the geometry: the final setting is what is called Riemann–Cartan geometry.

Granted that, from a general mathematical perspective, torsion is present, one may wonder if there can be physical reasons for torsion to be zero. Physical arguments to prove that torsion must equal zero were indeed proposed in the past. However, none of them appeared to be free of fallacies or logical inconsistencies. A complete list with detailed reasons for their failure can be found in [5].

That torsion should not be equal to zero even in physical contexts is again quite general. In fact, by writing the RC geometry in anholonomic bases, the torsion can be seen as the strength of the potential arising from gauging the translation group, much in the same way in which the curvature is the strength of the potential arising from gauging the rotation group, as shown by Sciama and Kibble [6,7]. What Sciama and Kibble proved was that torsion is not just a tensor that could be added, but a tensor that must be added, besides curvature, in order to have the possibility to completely describe translations, besides rotations, in a full Poincaré gauge theory of physics [8].

At the beginning of the 20th century, when Einstein developed his theory of gravity, he did it by assuming zero torsion because, when torsion vanishes, the Ricci tensor is symmetric and therefore it can be consistently coupled to the symmetric energy tensor, realizing the identification between the space–time curvature, and its energy content expressed by Einstein field equations: this is the basic spirit of Einstein gravity. Today, we know that in physics there is also another quantity of interest called spin, and that, in its presence, the energy is no longer symmetric, so nowadays having a non-symmetric Ricci tensor besides a Cartan tensor would allow for a more exhaustive coupling in gravity, where the curvature would still be coupled to the energy but now torsion would be coupled to the spin: such a scheme would realize the identification between the space–time curvature and its energy content expressed by Einstein field equations and the identification between space–time torsion and its spin content expressed by the Sciama–Kibble field equations as the Einstein–Sciama–Kibble torsion gravity.

The ESK theory of gravity is thus the most complete theory describing the dynamics of the space–time, and, because torsion is coupled to the spin in the same spirit in which curvature is coupled to energy, then it is the theory of space–time in which the coupling to its matter content is achieved most exhaustively. The central point of the situation is therefore brought to the question asking whether there actually exists something possessing both spin and energy as a form of matter, which can profit from the setting that is provided by the ESK gravity.

As a matter of fact, such a theory not only exists, but it is also very well known, the Dirac spinorial field theory.

With so much insight, it is an odd circumstance that there be still such a controversy about the role of torsion besides that of curvature in gravity, and there may actually be several reasons for it. The single most important one may be that Einstein gravity was first published in the year 1916 when no spin was known and, despite being then insightful to set the torsion tensor to zero, when Dirac came with a theory of spinors comprising an intrinsic spin in 1928, the successes of Einstein theory of gravity were already too great to make anyone wonder about the possibility of modifying it.

Of course, this is no scientific reason to hinder research, but, sociologically, it can be easy to understand why one would not lightly go to look beyond something good,

especially today that the successes of the Einstein theory of gravitation have become practically complete.

In the present report, we would like to change this tendency by considering torsion in gravity coupled to spinor fields and showing all advantages that we can get, from theoretical consistency, to phenomenological applications.

Thus, in a second chapter, we will investigate the theoretical advantages obtained from the torsion–spin interactions. These will span from the revision of the Hawking–Penrose theorem about the inevitability of gravitational singularity formation to some discussion about the positivity of the energy, passing through the Pauli exclusion principle and the concept of macroscopic approximation.

In a third and final chapter, we are going to employ the presented theory to assess some of the known open problems in the standard models of particles and cosmology.

*One: Fundamental Theory*

The first chapter will be about presenting the fundamental theory, and it will be divided into three sections: in the first section, we will define all the kinematic quantities and see how they can be dynamically coupled. It will be followed by a second section in which we will deepen the study about what is torsion and the spinor fields and the way they interact. A third section will be about studying limiting situations that can allow us to get even more information about the torsion–spin coupling.

## 2. Torsion Gravity for Spinor Fields

In this first section, we introduce the physical theory that will be our reference throughout the entire work: we start with the most general geometric introduction of the kinematic quantities. In addition, we will continue by establishing their link in terms of dynamical field equations.

### 2.1. Geometry and Its Matter Content

To build the geometric background on which to define kinematic fields, we start with the symmetry principle at the basis of any theory in physics: covariance under the most general transformation of coordinates. We will see in what way from such a general environment a natural definition of matter field will spontaneously emerge.

#### 2.1.1. Tensor and Gauge Fields

The principle of covariance under the most general transformation of coordinates is possible one of the most self-evident principles in all of physics: it states that our way of writing equations might a priori be conditioned in principle by the coordinates we choose, but observable properties should not feel affected by coordinate artifacts brought by us. This means that of all possible manners we have to write physics, there must be one that is not influenced by any choice of coordinates, or in other words, this specific way of writing physics has to be invariant between different coordinate systems.

To see how this is possible, we start with the following definition. Suppose that a certain physical quantity can be described in terms of the object $T^{\alpha\cdots\sigma}_{\rho\cdots\zeta}$ such that it is written as $T^{\alpha\cdots\sigma}_{\rho\cdots\zeta}(x)$ with respect to coordinates $x$, and it is written as $T'^{\alpha\cdots\sigma}_{\rho\cdots\zeta}(x')$ with respect to coordinates $x'$ in general, and suppose that

$$T'^{\alpha\cdots\sigma}_{\rho\cdots\zeta} = \frac{\partial x^{\beta}}{\partial x'^{\rho}} \cdots \frac{\partial x^{\theta}}{\partial x'^{\zeta}} \frac{\partial x'^{\alpha}}{\partial x^{\nu}} \cdots \frac{\partial x'^{\sigma}}{\partial x^{\tau}} T^{\nu\cdots\tau}_{\beta\cdots\theta}$$

where $x' = x'(x)$ determines the passage from the first to the second system of coordinates. If this happens, such a quantity is called a tensor. Now, suppose that one specific property of this quantity be described as

$$T^{\nu\cdots\tau}_{\beta\cdots\theta} = 0$$

in the first system of coordinates. According to the above definition, then, we have that

$$T'^{\alpha\ldots\sigma}_{\rho\ldots\zeta} = \frac{\partial x^\beta}{\partial x'^\rho}\cdots\frac{\partial x^\theta}{\partial x'^\zeta}\frac{\partial x'^\alpha}{\partial x^\nu}\cdots\frac{\partial x'^\sigma}{\partial x^\tau}\,T^{\nu\ldots\tau}_{\beta\ldots\theta} =$$
$$= \frac{\partial x^\beta}{\partial x'^\rho}\cdots\frac{\partial x^\theta}{\partial x'^\zeta}\frac{\partial x'^\alpha}{\partial x^\nu}\cdots\frac{\partial x'^\sigma}{\partial x^\tau}\,0 \equiv 0$$

and thus

$$T'^{\alpha\ldots\sigma}_{\rho\ldots\zeta} = 0$$

showing that the same property pertains to that quantity also in the second system of coordinates as well. In this way, we have that, if the property of a quantity is encoded as the vanishing of a tensor, then we can be certain that such a property pertains to that quantity regardless of the system of coordinates. In addition, this is just covariance.

The principle of covariance is therefore implemented in the geometry by the straightforward requirement that this geometry be written in terms of tensors. Therefore, let be given two systems of coordinates as $x$ and $x'$ related by the most general coordinate transformation $x' = x'(x)$ and a set of functions of these coordinates written with respect to the first and the second system of coordinates as $T(x)$ and $T'(x')$ and such that, for a coordinate transformation, they are related by

$$T'^{\alpha\ldots\sigma}_{\rho\ldots\zeta} = \text{sign det}\left(\frac{\partial x'}{\partial x}\right)\frac{\partial x^\beta}{\partial x'^\rho}\cdots\frac{\partial x^\theta}{\partial x'^\zeta}\frac{\partial x'^\alpha}{\partial x^\nu}\cdots\frac{\partial x'^\sigma}{\partial x^\tau}\,T^{\nu\ldots\tau}_{\beta\ldots\theta} \tag{1}$$

Then, this quantity is called *tensor* or *pseudo-tensor*, according to whether the sign of the determinant of such a transformation is positive or negative. For a tensor with at least two upper or two lower indices, we might switch the two indices obtaining a tensor called *transposition* of the original tensor in those two indices, and if it happens to be equal to the initial tensor up to the sign plus or minus, we say that the tensor is *symmetric* or *antisymmetric* in those two indices, respectively. Given a tensor with at least one upper and one lower index, we can consider one of the upper and one of the lower indices forcing them to have the same value and performing the sum over every possible value of those indices obtaining a tensor called *contraction* in those indices, and this process can be repeated until we reach a tensor whose contraction is zero, called *irreducible*. Particular cases are tensors having one index called *vectors*, while tensors without any index are called *scalars*. Tensors with the same index configuration can be *summed* and any two tensors can be *multiplied* in a component-by-component way, according to the usual rules of algebraic calculus as they are well known.

There is therefore no need to spend more time in the algebraic properties of tensors. However, differential properties of tensors need some deepening. The problem with differentiation applied to the case of tensors is that such an operation spoils the transformation law of a tensor in very general circumstances. Thus, if we want to construct an operation that is able to generalize the usual derivative up to a derivative that respects covariance, we must begin by noticing that, because a tensor is a set of fields, in general, it will have two types of variations: the first is due to the fact that tensors fields are fields, coordinates dependent, and so a local structure must be present as

$$_{\text{local}}\Delta T^{\alpha_1\ldots\alpha_j}_{\beta_1\ldots\beta_i} = T^{\alpha_1\ldots\alpha_j}_{\beta_1\ldots\beta_i}(x') - T^{\alpha_1\ldots\alpha_j}_{\beta_1\ldots\beta_i}(x) =$$
$$= \partial_\mu T^{\alpha_1\ldots\alpha_j}_{\beta_1\ldots\beta_i}(x)\delta x^\mu$$

at the first order infinitesimal; the second is due to the fact that tensors fields are tensors, so a system of components, and thus a re-shuffling of the different components must be allowed according to

$$\text{structure}\Delta T^{\alpha_1...\alpha_j}_{\beta_1...\beta_i} = T'^{\alpha_1...\alpha_j}_{\beta_1...\beta_i} - T^{\alpha_1...\alpha_j}_{\beta_1...\beta_i} =$$
$$= [(\delta\Gamma^{\alpha_1}_\theta T^{\theta...\alpha_j}_{\beta_1...\beta_i} + ... + \delta\Gamma^{\alpha_j}_\theta T^{\alpha_1...\theta}_{\beta_1...\beta_i}) -$$
$$- (\delta\Gamma^{\theta}_{\beta_1} T^{\alpha_1...\alpha_j}_{\theta...\beta_i} + ... + \delta\Gamma^{\theta}_{\beta_i} T^{\alpha_1...\alpha_j}_{\beta_1...\theta})]$$

as the most general form in which this can be done while respecting the fact that the differential structure requires the linearity and the Leibniz rule, and again at the first order of infinitesimal. In full, we have

$$\Delta T^{\alpha_1...\alpha_j}_{\beta_1...\beta_i} = \text{local}\Delta T^{\alpha_1...\alpha_j}_{\beta_1...\beta_i} + \text{structure}\Delta T^{\alpha_1...\alpha_j}_{\beta_1...\beta_i} =$$
$$= \partial_\mu T^{\alpha_1...\alpha_j}_{\beta_1...\beta_i}(x)\delta x^\mu +$$
$$+ [(\delta\Gamma^{\alpha_1}_\theta T^{\theta...\alpha_j}_{\beta_1...\beta_i} + ... + \delta\Gamma^{\alpha_j}_\theta T^{\alpha_1...\theta}_{\beta_1...\beta_i}) -$$
$$- (\delta\Gamma^{\theta}_{\beta_1} T^{\alpha_1...\alpha_j}_{\theta...\beta_i} + ... + \delta\Gamma^{\theta}_{\beta_i} T^{\alpha_1...\alpha_j}_{\beta_1...\theta})]$$

at the first order infinitesimal. Thus, defining $\delta\Gamma^\alpha_\beta = \Gamma^\alpha_{\beta\mu}\delta x^\mu$ and dividing by $\delta x^\mu$, we obtain that

$$D_\mu T^{\alpha_1...\alpha_j}_{\beta_1...\beta_i} = \partial_\mu T^{\alpha_1...\alpha_j}_{\beta_1...\beta_i} +$$
$$+ (\Gamma^{\alpha_1}_{\theta\mu} T^{\theta...\alpha_j}_{\beta_1...\beta_i} + ... + \Gamma^{\alpha_j}_{\theta\mu} T^{\alpha_1...\theta}_{\beta_1...\beta_i}) -$$
$$- (\Gamma^{\theta}_{\beta_1\mu} T^{\alpha_1...\alpha_j}_{\theta...\beta_i} + ... + \Gamma^{\theta}_{\beta_i\mu} T^{\alpha_1...\alpha_j}_{\beta_1...\theta})$$

after taking the limit. This is the most general form of potential covariant derivative. To see that this derivative is indeed covariant, we have to require that $\Gamma^\alpha_{\beta\mu}$ transforms with a specific non-tensorial transformation law such as to compensate for the non-tensorial transformation law of the partial derivative. In the simplest case of one tensorial index, we have that the derivative is

$$D_\iota V^\alpha = \partial_\iota V^\alpha + V^\beta\Gamma^\alpha_{\beta\iota}$$

whose transformation law is given by

$$\frac{\partial x^\beta}{\partial x'^{\beta'}}\frac{\partial x'^{\alpha'}}{\partial x^\alpha}(\partial_\beta V^\alpha + V^\rho\Gamma^\alpha_{\rho\beta}) = \frac{\partial x^\beta}{\partial x'^{\beta'}}\frac{\partial x'^{\alpha'}}{\partial x^\alpha}D_\beta V^\alpha =$$
$$= (D_\beta V^\alpha)' = (\partial_\beta V^\alpha + V^\rho\Gamma^\alpha_{\rho\beta})' = \partial_{\beta'} V^{\alpha'} + V^{\rho'}\Gamma'^{\alpha'}_{\rho'\beta'} =$$
$$= \frac{\partial x^\theta}{\partial x'^{\beta'}}\frac{\partial}{\partial x^\theta}\left(\frac{\partial x'^{\alpha'}}{\partial x^\alpha}V^\alpha\right) + \frac{\partial x'^{\rho'}}{\partial x^\rho}V^\rho\Gamma'^{\alpha'}_{\rho'\beta'} =$$
$$= \frac{\partial x^\theta}{\partial x'^{\beta'}}\frac{\partial x'^{\alpha'}}{\partial x^\alpha}\frac{\partial V^\alpha}{\partial x^\theta} + \frac{\partial x^\theta}{\partial x'^{\beta'}}\frac{\partial}{\partial x^\theta}\frac{\partial x'^{\alpha'}}{\partial x^\alpha}V^\alpha + \frac{\partial x'^{\rho'}}{\partial x^\rho}V^\rho\Gamma'^{\alpha'}_{\rho'\beta'}$$

in which terms with the derivatives disappear. Then,

$$\frac{\partial x^\beta}{\partial x'^{\beta'}}\frac{\partial x'^{\alpha'}}{\partial x^\alpha}V^\rho\Gamma^\alpha_{\rho\beta} = \frac{\partial x^\theta}{\partial x'^{\beta'}}\frac{\partial}{\partial x^\theta}\frac{\partial x'^{\alpha'}}{\partial x^\alpha}V^\alpha + \frac{\partial x'^{\rho'}}{\partial x^\rho}V^\rho\Gamma'^{\alpha'}_{\rho'\beta'}$$

and since this has to hold for any vector

$$\frac{\partial x^\beta}{\partial x'^{\beta'}}\frac{\partial x'^{\alpha'}}{\partial x^\alpha}\Gamma^\alpha_{\rho\beta} = \frac{\partial x^\theta}{\partial x'^{\beta'}}\frac{\partial}{\partial x^\theta}\frac{\partial x'^{\alpha'}}{\partial x^\rho} + \frac{\partial x'^{\rho'}}{\partial x^\rho}\Gamma'^{\alpha'}_{\rho'\beta'}$$

which is the non-tensorial transformation that the set of coefficients $\Gamma^{\alpha}_{\rho\beta}$ must have to ensure that the full derivative transforms as a tensor in this very specific case with a vector field. However, quite remarkably, the very same non-tensorial transformation of $\Gamma^{\alpha}_{\rho\beta}$ can be used for each term in the most general form of derivative for a generic tensor, and so the obtained result is completely general.

The set of coefficients $\Gamma^{\alpha}_{\rho\beta}$ have no specific symmetry properties in the lower indices, and consequently we have that we can write

$$\Gamma^{\alpha}_{\mu\nu} \equiv \tfrac{1}{2}(\Gamma^{\alpha}_{\mu\nu}+\Gamma^{\alpha}_{\nu\mu})+\tfrac{1}{2}(\Gamma^{\alpha}_{\mu\nu}-\Gamma^{\alpha}_{\nu\mu})$$

where the transformation properties of the full object is inherited by the first part, which is symmetric in the two lower indices, and it can be indicated as

$$\Lambda^{\alpha}_{\mu\nu}=\tfrac{1}{2}(\Gamma^{\alpha}_{\mu\nu}+\Gamma^{\alpha}_{\nu\mu})$$

while the second part

$$Q^{\alpha}{}_{\mu\nu} = \Gamma^{\alpha}_{\mu\nu} - \Gamma^{\alpha}_{\nu\mu}$$

transforms as a tensor such that $Q^{\alpha}{}_{\mu\nu} = -Q^{\alpha}{}_{\nu\mu}$, meaning that it is antisymmetric in its second pair of indices. Thus,

$$\Gamma^{\alpha}_{\mu\nu}=\Lambda^{\alpha}_{\mu\nu}+\tfrac{1}{2}Q^{\alpha}{}_{\mu\nu}$$

in the most general case. As in the covariant derivatives, the connection enters linearly, and the splitting in symmetric and antisymmetric parts sums up to a linear combination of the tensor $Q^{\alpha}{}_{\mu\nu}$ plus the terms linear in the symmetric connection, which therefore forms yet another type of covariant derivative that is defined according to

$$\nabla_{\mu}T^{\alpha_1\dots\alpha_j}_{\beta_1\dots\beta_i}=\partial_{\mu}T^{\alpha_1\dots\alpha_j}_{\beta_1\dots\beta_i}+$$
$$+(\Lambda^{\alpha_1}_{\theta\mu}T^{\theta\dots\alpha_j}_{\beta_1\dots\beta_i}+\dots+\Lambda^{\alpha_j}_{\theta\mu}T^{\alpha_1\dots\theta}_{\beta_1\dots\beta_i})-$$
$$-(\Lambda^{\theta}_{\beta_1\mu}T^{\alpha_1\dots\alpha_j}_{\theta\dots\beta_i}+\dots+\Lambda^{\theta}_{\beta_i\mu}T^{\alpha_1\dots\alpha_j}_{\beta_1\dots\theta})$$

and in it the fact that the symmetric connection is indeed symmetric allows for particularly simplified expressions in some special cases. For instance, taking the symmetric covariant derivative of a tensor with all lower indices gives

$$\nabla_{\mu}T_{\beta_1\dots\beta_i}=\partial_{\mu}T_{\beta_1\dots\beta_i}-\Lambda^{\theta}_{\beta_1\mu}T_{\theta\dots\beta_i}-\dots-\Lambda^{\theta}_{\beta_i\mu}T_{\beta_1\dots\theta}$$

which is particularly interesting because we see that the symmetric connection always saturates the same index in the upper position, so that, if we further specialize onto the case in which the tensor is completely antisymmetric, we obtain that

$$\nabla_{[\mu}T_{\beta\dots\rho]}=\nabla_{\mu}T_{\beta\dots\rho}-\nabla_{\beta}T_{\mu\dots\rho}+\dots-\nabla_{\rho}T_{\beta\dots\mu}=$$
$$=\partial_{\mu}T_{\beta\dots\rho}-\Lambda^{\sigma}_{\beta\mu}T_{\sigma\dots\rho}-\dots-\Lambda^{\sigma}_{\rho\mu}T_{\beta\dots\sigma}-$$
$$-\partial_{\beta}T_{\mu\dots\rho}+\Lambda^{\sigma}_{\mu\beta}T_{\sigma\dots\rho}+\dots+\Lambda^{\sigma}_{\rho\beta}T_{\mu\dots\sigma}+\dots$$
$$\dots-\partial_{\rho}T_{\beta\dots\mu}+\Lambda^{\sigma}_{\beta\rho}T_{\sigma\dots\mu}+\dots+\Lambda^{\sigma}_{\mu\rho}T_{\beta\dots\sigma}=$$
$$=\partial_{\mu}T_{\beta\dots\rho}-\partial_{\beta}T_{\mu\dots\rho}+\dots-\partial_{\rho}T_{\beta\dots\mu}=\partial_{[\mu}T_{\beta\dots\rho]}$$

where all symmetric connections cancelled off leaving an expression written only in terms of partial derivatives but that is a completely antisymmetric covariant derivative in the most general case. This is a very peculiar property of tensors having all lower indices and being completely antisymmetric in all of these indices, and there is an entire domain related to this type of tensors and covariant derivatives, in which tensors are known as forms and

the covariant derivatives are part of what is known as exterior calculus. Nevertheless, we will not discuss it here because we do not want to introduce even further mathematical concepts and after all forms and exterior derivatives are nothing but a specific type of tensors. We encourage the interested readers to study this domain on their own.

Thus, to summarize what we have done, we have that the set of functions $\Gamma^\rho_{\alpha\beta}$ transforming as

$$\Gamma'^\rho_{\sigma\tau} = \left( \Gamma^\alpha_{\mu\nu} - \frac{\partial x^\alpha}{\partial x'^\kappa} \frac{\partial^2 x'^\kappa}{\partial x'^\nu \partial x'^\mu} \right) \frac{\partial x'^\rho}{\partial x^\alpha} \frac{\partial x^\mu}{\partial x'^\sigma} \frac{\partial x^\nu}{\partial x'^\tau} \tag{2}$$

is called *connection*, and it can be decomposed as

$$\Gamma^\rho_{\alpha\beta} = \Lambda^\rho_{\alpha\beta} + \tfrac{1}{2} Q^\rho{}_{\alpha\beta} \tag{3}$$

where $\Lambda^\rho_{\alpha\beta}$ is a set of functions transforming according to the law of a connection but which are symmetric in the two lower indices called *symmetric connection* and

$$Q^\rho{}_{\alpha\beta} = \Gamma^\rho_{\alpha\beta} - \Gamma^\rho_{\beta\alpha} \tag{4}$$

which is a tensor antisymmetric in the two lower indices called *torsion tensor*. In terms of the connection, we may write the *covariant derivative* in the most general case as

$$D_\mu T^{\alpha_1\ldots\alpha_j}_{\beta_1\ldots\beta_i} = \partial_\mu T^{\alpha_1\ldots\alpha_j}_{\beta_1\ldots\beta_i} + \sum_{k=1}^{k=j} \Gamma^{\alpha_k}_{\sigma\mu} T^{\alpha_1\ldots\sigma\ldots\alpha_j}_{\beta_1\ldots\beta_i} -$$
$$- \sum_{k=1}^{k=i} \Gamma^\sigma_{\beta_k\mu} T^{\alpha_1\ldots\alpha_j}_{\beta_1\ldots\sigma\ldots\beta_i} \tag{5}$$

decomposing as

$$D_\mu T^{\alpha_1\ldots\alpha_j}_{\beta_1\ldots\beta_i} = \nabla_\mu T^{\alpha_1\ldots\alpha_j}_{\beta_1\ldots\beta_i} + \tfrac{1}{2}\sum_{k=1}^{k=j} Q^{\alpha_k}_{\sigma\mu} T^{\alpha_1\ldots\sigma\ldots\alpha_j}_{\beta_1\ldots\beta_i} -$$
$$- \tfrac{1}{2}\sum_{k=1}^{k=i} Q^\sigma_{\beta_k\mu} T^{\alpha_1\ldots\alpha_j}_{\beta_1\ldots\sigma\ldots\beta_i} \tag{6}$$

with spurious terms linear in the torsion tensor and

$$\nabla_\mu T^{\alpha_1\ldots\alpha_j}_{\beta_1\ldots\beta_i} = \partial_\mu T^{\alpha_1\ldots\alpha_j}_{\beta_1\ldots\beta_i} + \sum_{k=1}^{k=j} \Lambda^{\alpha_k}_{\sigma\mu} T^{\alpha_1\ldots\sigma\ldots\alpha_j}_{\beta_1\ldots\beta_i} -$$
$$- \sum_{k=1}^{k=i} \Lambda^\sigma_{\beta_k\mu} T^{\alpha_1\ldots\alpha_j}_{\beta_1\ldots\sigma\ldots\beta_i} \tag{7}$$

which is the *covariant derivative calculated with respect to the symmetric connection*. If we apply the last definition to the particular case of tensors with all lower indices and being completely antisymmetric, we get

$$\nabla_{[\nu} T_{\alpha\ldots\sigma]} = \partial_{[\nu} T_{\alpha\ldots\sigma]} \equiv (\partial T)_{\nu\alpha\ldots\sigma}, \tag{8}$$

which is still a tensor and such that it is completely antisymmetric called a *covariant curl* of the tensor field. When in the covariant derivative of a tensor with at least one upper index we contract the index of derivation with an upper index of the tensor field, we get what is known as *covariant divergence* in that index of the tensor field.

When we have introduced the concept of tensor, it naturally emerged that, in the definition, two types of indices were present, upper and lower, reflecting the fact that a tensor could transform according to two type of transformations, direct and inverse. However, these two types of transformation are two different forms of the same transformation, and so one should expect that the two types of indices be two different arrangements of the same system of components. Thus, there should be no difference in content if we move a given index up or down at will.

What this implies is that it should be possible to move indices up and down without losing or adding anything to the information content: this can be done by considering the

Kronecker tensor $\delta_\nu^\alpha$ and postulating the existence of two tensors $g_{\alpha\nu}$ and $g^{\alpha\nu}$ in general. Then, we can define the operation of raising and lowering of tensorial indices by considering that $A^\pi g_{\pi\nu}$ and $A_\pi g^{\pi\nu}$ are tensors that are related to the initial ones but with the index lowered and raised, respectively, and so we may define these two tensors as $A_\pi g^{\pi\nu} \equiv A^\nu$ and $A^\pi g_{\pi\nu} \equiv A_\nu$ as the same tensors but with the index moved in a different position with respect to the initial one. While it is certainly useful to have the possibility to perform such an operation, we also have to consider that such an operation has a two-fold ambiguity concerning the fact that, besides the contractions $A_\pi g^{\pi\nu} \equiv A^\nu$ and $A^\pi g_{\pi\nu} \equiv A_\nu$, we may have the contractions $A_\pi g^{\nu\pi} \equiv A^\nu$ and $A^\pi g_{\nu\pi} \equiv A_\nu$ too. In addition, we may decide to raise the previously lowered index to the initial position or lower the previously raise index to the initial position, so that the above ambiguity becomes four-fold with $A_\pi g^{\pi\nu} g_{\sigma\nu} \equiv A_\sigma$ and $A_\pi g^{\nu\pi} g_{\sigma\nu} \equiv A_\sigma$ as well as $A_\pi g^{\pi\nu} g_{\nu\sigma} \equiv A_\sigma$ and $A_\pi g^{\nu\pi} g_{\nu\sigma} \equiv A_\sigma$ as equally good possibilities that may be considered. On the other hand, requiring that raising one index up and then lowering that index down give back the initial tensor in all of the four possibilities leads to the following relationships

$$A_\mu(g^{\mu\sigma}g_{\sigma\kappa} - \delta_\kappa^\mu) = 0 \qquad A_\mu(g^{\sigma\mu}g_{\sigma\kappa} - \delta_\kappa^\mu) = 0$$
$$A_\mu(g^{\mu\sigma}g_{\kappa\sigma} - \delta_\kappa^\mu) = 0 \qquad A_\mu(g^{\sigma\mu}g_{\kappa\sigma} - \delta_\kappa^\mu) = 0$$

for any possible tensor $A_\mu$, so that

$$(g^{\mu\sigma}g_{\sigma\kappa} - \delta_\kappa^\mu) = 0 \qquad (g^{\sigma\mu}g_{\sigma\kappa} - \delta_\kappa^\mu) = 0$$
$$(g^{\mu\sigma}g_{\kappa\sigma} - \delta_\kappa^\mu) = 0 \qquad (g^{\sigma\mu}g_{\kappa\sigma} - \delta_\kappa^\mu) = 0$$

identically. Taking the differences

$$g^{\mu\sigma}(g_{\sigma\kappa} - g_{\kappa\sigma}) = 0 \qquad (g^{\sigma\mu} - g^{\mu\sigma})g_{\sigma\kappa} = 0$$
$$g^{\sigma\mu}(g_{\sigma\kappa} - g_{\kappa\sigma}) = 0 \qquad (g^{\sigma\mu} - g^{\mu\sigma})g_{\kappa\sigma} = 0,$$

we may work out that

$$g_{\alpha\kappa} = g_{\kappa\alpha}$$
$$g^{\alpha\kappa} = g^{\kappa\alpha}$$

together with the condition

$$g^{\sigma\mu}g_{\kappa\sigma} = \delta_\kappa^\mu$$

meaning that, seen as matrices, they are symmetric and one the inverse of the other, and so, in particular, they are non-degenerate, as it has been demonstrated in [9]. This implies that what has been introduced to raise lower or lower upper indices has all the features of a metric and therefore these two tensors can also be identified with the metric of the space–time. We remark that this is exactly the opposite to the normal approach, where the metric is postulated, and then it is realized that it can be used to move up and down indices of tensors. The equivalence of these two a priori unrelated operations looks profound.

The metric determinant $\det(g_{\mu\nu}) = g$ can never be zero, but it follows the transformation law

$$g' = \det\left|\frac{\partial x}{\partial x'}\right|^2 g$$

which is not the transformation law for a tensor. However, it can still be used to form a very important tensor as in the following. Consider in fact the non-tensorial quantity that is given by $\epsilon_{i_1 i_2 i_3 i_4}$ such that it is equal to the unity for an even permutation of (1234) and minus the unity for an odd permutation of (1234) and zero for a sequence that is not a permutation of (1234) at all. As this set of coefficients is completely antisymmetric with a

number of indices that is equal to the dimension, we have that it has only one independent component, transforming as

$$\frac{\partial x^{i_1}}{\partial x'^{i'_1}} \frac{\partial x^{i_2}}{\partial x'^{i'_2}} \frac{\partial x^{i_3}}{\partial x'^{i'_3}} \frac{\partial x^{i_4}}{\partial x'^{i'_4}} \epsilon_{i_1 i_2 i_3 i_4} = \epsilon_{i'_1 i'_2 i'_3 i'_4} \alpha$$

for a given $\alpha$ function to be determined. In addition, because the determinant of any generic matrix can always be written in terms of these coefficients according to the expression given by $\det M = \Sigma_{i_j} \epsilon_{i_1 i_2 i_3 i_4} M^{1 i_1} M^{2 i_2} M^{3 i_3} M^{4 i_4}$, then

$$\det \frac{\partial x}{\partial x'} = \frac{\partial x^{i_1}}{\partial x'^1} \frac{\partial x^{i_2}}{\partial x'^2} \frac{\partial x^{i_3}}{\partial x'^3} \frac{\partial x^{i_4}}{\partial x'^4} \epsilon_{i_1 i_2 i_3 i_4} = \epsilon_{1234} \alpha = \alpha$$

furnishing the $\alpha$ function. Thus, we have

$$\epsilon_{i'_1 i'_2 i'_3 i'_4} = \det \frac{\partial x'}{\partial x} \frac{\partial x^{i_1}}{\partial x'^{i'_1}} \frac{\partial x^{i_2}}{\partial x'^{i'_2}} \frac{\partial x^{i_3}}{\partial x'^{i'_3}} \frac{\partial x^{i_4}}{\partial x'^{i'_4}} \epsilon_{i_1 i_2 i_3 i_4}$$

which is non-tensorial, but its non-tensoriality perfectly matches that of the determinant of the metric. Therefore, we have that they compensate in the combined form

$$(g^{\frac{1}{2}} \epsilon_{\alpha \nu \sigma \tau})' = \text{sign} \det \left| \frac{\partial x'}{\partial x} \right| \frac{\partial x^{\beta}}{\partial x'^{\alpha}} \frac{\partial x^{\mu}}{\partial x'^{\nu}} \frac{\partial x^{\theta}}{\partial x'^{\sigma}} \frac{\partial x^{\rho}}{\partial x'^{\tau}} (g^{\frac{1}{2}} \epsilon_{\beta \mu \theta \rho}),$$

which is in fact the transformation that defines a pseudo-tensorial field. Notice, however, that, if we were to define the tensor with all lower indices as

$$\varepsilon_{\alpha \nu \sigma \tau} = \epsilon_{\alpha \nu \sigma \tau} |g|^{\frac{1}{2}}$$

the correspondent tensor with all upper indices would be given according to the following expression:

$$\varepsilon^{\alpha \nu \sigma \tau} = \epsilon^{\alpha \nu \sigma \tau} |g|^{-\frac{1}{2}}$$

in order for it to be consistently defined. This difference is necessary, as it can be seen from the fact that the quantity

$$\varepsilon^{i_1 i_2 i_3 i_4} \varepsilon_{j_1 j_2 j_3 j_4} = -\det \begin{vmatrix} \delta^{i_1}_{j_1} & \delta^{i_2}_{j_1} & \delta^{i_3}_{j_1} & \delta^{i_4}_{j_1} \\ \delta^{i_1}_{j_2} & \delta^{i_2}_{j_2} & \delta^{i_3}_{j_2} & \delta^{i_4}_{j_2} \\ \delta^{i_1}_{j_3} & \delta^{i_2}_{j_3} & \delta^{i_3}_{j_3} & \delta^{i_4}_{j_3} \\ \delta^{i_1}_{j_4} & \delta^{i_2}_{j_4} & \delta^{i_3}_{j_4} & \delta^{i_4}_{j_4} \end{vmatrix}$$

as it is very easy to check by performing a straightforward substitution and making all the direct calculations.

To summarize, the object $\delta^{\beta}_{\alpha}$ that is unity or zero according to whether the value of its indices is equal or not is the *unity tensor* mentioned. We assume the existence of two tensors $g_{\alpha \kappa}$ and $g^{\alpha \kappa}$ symmetric and such that

$$g^{\sigma \mu} g_{\kappa \sigma} = \delta^{\mu}_{\kappa} \tag{9}$$

called *metric tensors*. In addition, we define

$$\delta^{i_0 i_1 i_2 i_3}_{j_0 j_1 j_2 j_3} = \det \begin{vmatrix} \delta^{i_0}_{j_0} & \delta^{i_1}_{j_0} & \delta^{i_2}_{j_0} & \delta^{i_3}_{j_0} \\ \delta^{i_0}_{j_1} & \delta^{i_1}_{j_1} & \delta^{i_2}_{j_1} & \delta^{i_3}_{j_1} \\ \delta^{i_0}_{j_2} & \delta^{i_1}_{j_2} & \delta^{i_2}_{j_2} & \delta^{i_3}_{j_2} \\ \delta^{i_0}_{j_3} & \delta^{i_1}_{j_3} & \delta^{i_2}_{j_3} & \delta^{i_3}_{j_3} \end{vmatrix} \tag{10}$$

as a *completely antisymmetric unity tensor*. The quantity given by $\epsilon_{i_0 i_1 i_2 i_3}$ equal to the unity, minus unity, or zero according to whether $(i_0 i_1 i_2 i_3)$ is an even, odd, or not a permutation of $(0123)$ can be taken with the determinant of the metric $\det(g_{\mu\nu}) = g$ to define

$$\varepsilon^{\alpha\nu\sigma\tau} = \epsilon^{\alpha\nu\sigma\tau}|g|^{-\frac{1}{2}} \tag{11}$$

as well as

$$\varepsilon_{\alpha\nu\sigma\tau} = \epsilon_{\alpha\nu\sigma\tau}|g|^{\frac{1}{2}} \tag{12}$$

which are completely antisymmetric and such that

$$\varepsilon^{i_0 i_1 i_2 i_3}\varepsilon_{j_0 j_1 j_2 j_3} = -\delta^{i_0 i_1 i_2 i_3}_{j_0 j_1 j_2 j_3} \tag{13}$$

called *completely antisymmetric pseudo-tensors*. When a tensor with at least one index is multiplied by the metric tensor and the index is contracted with one index of the metric tensor, the result is a tensor in which the index has been *vertically moved*. In particular, if a tensor that is completely antisymmetric in $k$ indices is multiplied by the completely antisymmetric pseudo-tensors and the $k$ indices of the tensor are contracted with $k$ indices of the completely antisymmetric pseudo-tensors, the result is a pseudo-tensor antisymmetric in $(4-k)$ indices called *dual*.

We are now at a point where we have defined for tensors a covariant operation that respects all rules of differentiation as well as the tensorial structure and an operation for the vertical re-configuration of tensorial indices, and we may wonder what happens when both operations are taken in parallel. More precisely, if the vertical index configuration cannot change the information content of a tensor, then this must be true for any tensor, and, in particular, if the tensor is the covariant derivative of some other tensor. Consequently, it must be possible to define

$$g^{\alpha\beta}D_\mu T^{\nu...\zeta}_{\beta\rho\sigma...\theta} = D_\mu T^{\alpha\nu...\zeta}_{\rho\sigma...\theta}$$

which therefore implies

$$D_\mu T^{\alpha\nu...\zeta}_{\rho\sigma...\theta} = D_\mu(g^{\alpha\beta}T^{\nu...\zeta}_{\beta\rho\sigma...\theta}) = D_\mu g^{\alpha\beta}T^{\nu...\zeta}_{\beta\rho\sigma...\theta} +$$
$$+ g^{\alpha\beta}D_\mu T^{\nu...\zeta}_{\beta\rho\sigma...\theta}$$

so that we are left with the equation

$$D_\mu g^{\alpha\beta}T^{\nu...\zeta}_{\beta\rho\sigma...\theta} = 0$$

for any tensor, implying $D_\mu g^{\alpha\beta} = 0$ as well. This means that the metric tensor is covariantly constant. Conditions of vanishing of the covariant derivative of the metric tensor mean that the irrelevance of the indices disposition must be valid regardless of the differential order of the tensor. If we were to follow the common approach defining the metric first, these conditions would mean that the metric structure and the local structure will have to be independent. This is reasonable since, if a vector is constant, its norm should be constant too. It is interesting to notice that, since we have two types of covariant derivatives and because the present arguments hold, regardless of the specific covariant derivative, then we have to assume that both covariant derivatives of the metric tensor vanish as $D_\mu g_{\alpha\beta} = \nabla_\mu g_{\alpha\beta} = 0$ in general. In particular, we have that $D_\theta \varepsilon_{\alpha\beta\mu\nu} = \nabla_\theta \varepsilon_{\alpha\beta\mu\nu} = 0$ hold as well. If we are insisting that this happen, then there are very remarkable consequences that follow. To see this, expand

$$0 = D_\rho g_{\alpha\beta} = \partial_\rho g_{\alpha\beta} - g_{\alpha\mu}\Gamma^\mu_{\beta\rho} - g_{\mu\beta}\Gamma^\mu_{\alpha\rho}$$

and take the three different indices permutations combined together with the definition of torsion to get

$$\Gamma^\rho_{\alpha\beta} = \tfrac{1}{2}Q^\rho{}_{\alpha\beta} + \tfrac{1}{2}\left(Q_{\alpha\beta}{}^\rho + Q_{\beta\alpha}{}^\rho\right) +$$
$$+ \tfrac{1}{2}g^{\rho\mu}\left(\partial_\beta g_{\alpha\mu} + \partial_\alpha g_{\mu\beta} - \partial_\mu g_{\alpha\beta}\right)$$

in which $Q_{\rho\alpha\sigma}$ is the torsion tensor antisymmetric in the two lower indices, while $(Q_{\alpha\beta\rho} + Q_{\beta\alpha\rho})$ is a tensor symmetric in those indices, whereas the remaining coefficients written in terms of the partial derivatives of the metric tensor transform as a connection and they are symmetric in those very indices. This expression shows that the most general connection can be decomposed in terms of the torsion plus a symmetric connection, as we already knew from expression (3), but, in addition, it tells us the explicit form of $\Lambda^\rho_{\alpha\beta}$ as given by a symmetric combination of two torsions plus a symmetric connection entirely written in terms of the metric. It is essential to note that, if we want all possible connections to give rise to covariant derivatives, which, once applied onto the metric, give zero, then we have to restrict the torsion to verify

$$Q_{\alpha\beta\rho} = -Q_{\beta\alpha\rho}$$

spelling its complete antisymmetry [10]. The condition of metric-compatible connection extended to all connections implies the torsion to be completely antisymmetric, once again establishing a link between two structures that are a priori unrelated. The complete antisymmetry of torsion is equivalent to the existence of a single symmetric part of the connection, and therefore to the existence of a unique connection writable in terms of the metric alone. It is a remarkable fact that the torsion tensor could be reduced to be completely antisymmetric by employing a number of unrelated arguments as those presented in [11–14] and, although torsion might well not display such a symmetry, it is certainly intriguing to argue what the consequences are of this condition. We will see that some of these consequence are of paramount importance next.

Thus, we summarize by saying that the torsion tensor with all lower indices is taken to be completely antisymmetric and therefore it is possible to write it according to

$$Q_{\alpha\sigma\nu} = \tfrac{1}{6}W^\mu \varepsilon_{\mu\alpha\sigma\nu} \tag{14}$$

in terms of the $W^\mu$ pseudo-vector, therefore called the torsion pseudo-vector, while the connection

$$\Lambda^\rho_{\alpha\beta} = \tfrac{1}{2}g^{\rho\mu}\left(\partial_\beta g_{\alpha\mu} + \partial_\alpha g_{\mu\beta} - \partial_\mu g_{\alpha\beta}\right) \tag{15}$$

is symmetric and written entirely in terms of the partial derivatives of the metric tensor and, for this reason called the *metric connection*, so that

$$\Gamma^\rho_{\alpha\beta} = \tfrac{1}{2}g^{\rho\mu}\left[\left(\partial_\beta g_{\alpha\mu} + \partial_\alpha g_{\mu\beta} - \partial_\mu g_{\alpha\beta}\right) + \tfrac{1}{6}W^\nu \varepsilon_{\nu\mu\alpha\beta}\right] \tag{16}$$

is the most general connection and such a decomposition is equivalent to the validity of the following conditions:

$$\nabla_\theta \varepsilon_{\alpha\beta\mu\nu} \equiv D_\theta \varepsilon_{\alpha\beta\mu\nu} = 0 \tag{17}$$
$$\nabla_\mu g_{\alpha\beta} \equiv D_\mu g_{\alpha\beta} = 0 \tag{18}$$

called *metric-compatibility conditions for the connection*.

Thus far, we have defined tensors and the properties compatible with the derivation. It is now the time to see what happens when we go to a following order derivative.

We may proceed to calculate the commutator of two derivatives, which in the particular case of vectors is

$$[D_\alpha, D_\beta] T^\sigma = (\Gamma^\rho_{\alpha\beta} - \Gamma^\rho_{\beta\alpha}) D_\rho T^\sigma +$$
$$+ (\partial_\alpha \Gamma^\sigma_{\kappa\beta} - \partial_\beta \Gamma^\sigma_{\kappa\alpha} + \Gamma^\rho_{\kappa\beta} \Gamma^\sigma_{\rho\alpha} - \Gamma^\rho_{\kappa\alpha} \Gamma^\sigma_{\rho\beta}) T^\kappa$$

with no second derivatives. The only derivative term left is proportional to the torsion tensor $Q^\rho_{\mu\alpha}$ plus another

$$G^\sigma_{\kappa\alpha\beta} = \partial_\alpha \Gamma^\sigma_{\kappa\beta} - \partial_\beta \Gamma^\sigma_{\kappa\alpha} + \Gamma^\rho_{\kappa\beta} \Gamma^\sigma_{\rho\alpha} - \Gamma^\rho_{\kappa\alpha} \Gamma^\sigma_{\rho\beta}$$

which, although written in terms of the connection alone, is a tensor. With these expressions, we have

$$[D_\alpha, D_\beta] T^\sigma = Q^\rho_{\alpha\beta} D_\rho T^\sigma + G^\sigma_{\kappa\alpha\beta} T^\kappa$$

giving the commutator of vectors in particular. As it has been done for the connection and the most general covariant derivative, the interesting thing is that the definition of tensor $G^\sigma_{\kappa\alpha\beta}$ can be used in the most general case of the commutator of covariant derivatives. We also have

$$(\partial\partial T)_{\alpha\beta\rho\ldots\mu} = \partial_{[\alpha} (\partial T)_{\beta\rho\ldots\mu]} = \partial_{[\alpha} \partial_{[\beta} T_{\rho\ldots\mu]]} =$$
$$= \partial_{[\alpha} \partial_\beta T_{\rho\ldots\mu]} = 0$$

because partial derivatives always commute and therefore their commutator is always zero. Before we have had the opportunity to briefly talk about external calculus, where the external derivatives are used to calculate the border of a manifold, and the above expression refers to the fact that the border has a border that vanishes, or that there is no border of a border. Once again, apart from curiosity, there is no need to deepen these concepts in the following.

To summarize, from the connection, we may calculate

$$G^\sigma_{\kappa\alpha\beta} = \partial_\alpha \Gamma^\sigma_{\kappa\beta} - \partial_\beta \Gamma^\sigma_{\kappa\alpha} + \Gamma^\sigma_{\rho\alpha} \Gamma^\rho_{\kappa\beta} - \Gamma^\sigma_{\rho\beta} \Gamma^\rho_{\kappa\alpha} \tag{19}$$

which is a tensor antisymmetric in the last two indices and verifying the following cyclic permutation condition

$$D_\kappa Q^\rho_{\mu\nu} + D_\nu Q^\rho_{\kappa\mu} + D_\mu Q^\rho_{\nu\kappa} +$$
$$+ Q^\pi_{\nu\kappa} Q^\rho_{\mu\pi} + Q^\pi_{\mu\nu} Q^\rho_{\kappa\pi} + Q^\pi_{\kappa\mu} Q^\rho_{\nu\pi} -$$
$$- G^\rho_{\kappa\nu\mu} - G^\rho_{\mu\kappa\nu} - G^\rho_{\nu\mu\kappa} \equiv 0 \tag{20}$$

called the *curvature tensor* and decomposable as

$$G^\sigma_{\kappa\alpha\beta} = R^\sigma_{\kappa\alpha\beta} + \tfrac{1}{2} (\nabla_\alpha Q^\sigma_{\kappa\beta} - \nabla_\beta Q^\sigma_{\kappa\alpha}) +$$
$$+ \tfrac{1}{4} (Q^\sigma_{\rho\alpha} Q^\rho_{\kappa\beta} - Q^\sigma_{\rho\beta} Q^\rho_{\kappa\alpha}) \tag{21}$$

in terms of torsion and

$$R^\sigma_{\kappa\alpha\beta} = \partial_\alpha \Lambda^\sigma_{\kappa\beta} - \partial_\beta \Lambda^\sigma_{\kappa\alpha} + \Lambda^\sigma_{\rho\alpha} \Lambda^\rho_{\kappa\beta} - \Lambda^\sigma_{\rho\beta} \Lambda^\rho_{\kappa\alpha} \tag{22}$$

as a tensor antisymmetric in the last two indices and such that it verifies the cyclic permutation condition

$$R^\rho_{\kappa\nu\mu} + R^\rho_{\mu\kappa\nu} + R^\rho_{\nu\mu\kappa} \equiv 0 \tag{23}$$

called *metric curvature tensor*. By employing torsion and curvature, it is possible to demonstrate that we have

$$
\begin{aligned}
[D_\mu, D_\nu] T^{\alpha_1 \dots \alpha_j}_{\beta_1 \dots \beta_i} = Q^\eta{}_{\mu\nu} D_\eta T^{\alpha_1 \dots \alpha_j}_{\beta_1 \dots \beta_i} + \\
+ \sum_{k=1}^{k=j} G^{\alpha_k}{}_{\sigma\mu\nu} T^{\alpha_1 \dots \sigma \dots \alpha_j}_{\beta_1 \dots \beta_i} - \\
- \sum_{k=1}^{k=i} G^\sigma{}_{\beta_k \mu\nu} T^{\alpha_1 \dots \alpha_j}_{\beta_1 \dots \sigma \dots \beta_i}
\end{aligned}
\tag{24}
$$

as the expression for *commutator of covariant derivatives* of the tensor field. In particular, we have that

$$
\partial\partial T = 0
\tag{25}
$$

which is valid in the most general circumstance.

We have the validity of the following decomposition

$$
\begin{aligned}
R_{\kappa\rho\alpha\mu} = \tfrac{1}{2}(\partial_\alpha\partial_\rho g_{\mu\kappa} - \partial_\mu\partial_\rho g_{\kappa\alpha} + \partial_\mu\partial_\kappa g_{\alpha\rho} - \partial_\kappa\partial_\alpha g_{\mu\rho}) + \\
+ \tfrac{1}{4}g^{\sigma\nu}[(\partial_\rho g_{\alpha\nu} + \partial_\alpha g_{\rho\nu} - \partial_\nu g_{\rho\alpha})(\partial_\kappa g_{\mu\sigma} + \partial_\mu g_{\kappa\sigma} - \partial_\sigma g_{\kappa\mu}) - \\
- (\partial_\rho g_{\mu\nu} + \partial_\mu g_{\rho\nu} - \partial_\nu g_{\rho\mu})(\partial_\kappa g_{\alpha\sigma} + \partial_\alpha g_{\kappa\sigma} - \partial_\sigma g_{\kappa\alpha})]
\end{aligned}
\tag{26}
$$

showing the antisymmetry also in the first two indices as well as the symmetry involving all four indices

$$
R_{\rho\kappa\mu\nu} = R_{\mu\nu\rho\kappa}
\tag{27}
$$

and, as a consequence, the metric curvature tensor has one independent contraction $R_{\mu\sigma} = R^\rho{}_{\mu\rho\sigma}$, which is symmetric and called *Ricci metric curvature tensor* with contraction $R = R_{\mu\sigma} g^{\mu\sigma}$ called *Ricci metric curvature scalar*, so that, with torsion, we can write

$$
\begin{aligned}
G_{\kappa\rho\alpha\mu} = \tfrac{1}{2}(\partial_\alpha\partial_\rho g_{\mu\kappa} - \partial_\mu\partial_\rho g_{\kappa\alpha} + \partial_\mu\partial_\kappa g_{\alpha\rho} - \partial_\kappa\partial_\alpha g_{\mu\rho}) + \\
+ \tfrac{1}{4}g^{\sigma\nu}[(\partial_\rho g_{\alpha\nu} + \partial_\alpha g_{\rho\nu} - \partial_\nu g_{\rho\alpha})(\partial_\kappa g_{\mu\sigma} + \partial_\mu g_{\kappa\sigma} - \partial_\sigma g_{\kappa\mu}) - \\
- (\partial_\rho g_{\mu\nu} + \partial_\mu g_{\rho\nu} - \partial_\nu g_{\rho\mu})(\partial_\kappa g_{\alpha\sigma} + \partial_\alpha g_{\kappa\sigma} - \partial_\sigma g_{\kappa\alpha})] + \\
+ \tfrac{1}{12}\nabla^\eta W^\sigma(g_{\alpha\eta}\varepsilon_{\sigma\kappa\rho\mu} - g_{\mu\eta}\varepsilon_{\sigma\kappa\rho\alpha}) + \tfrac{1}{144}[W_\sigma W^\sigma(g_{\mu\rho}g_{\alpha\kappa} - g_{\mu\kappa}g_{\alpha\rho}) + \\
+ (W_\alpha W_\rho g_{\mu\kappa} - W_\mu W_\rho g_{\alpha\kappa} + W_\mu W_\kappa g_{\alpha\rho} - W_\alpha W_\kappa g_{\mu\rho})]
\end{aligned}
\tag{28}
$$

showing the antisymmetry in the first two indices, and, as a consequence, it has one independent contraction chosen as $G_{\mu\sigma} = G^\rho{}_{\mu\rho\sigma}$ called *Ricci curvature tensor* whose contraction $G = G_{\mu\sigma} g^{\mu\sigma}$ is called *Ricci curvature scalar*.

In addition, finally, we may consider the cyclic permutation of commutator of commutators of covariant derivatives and see that the results are geometric identities.

In general, we have that we can write

$$
\begin{aligned}
D_\mu G^\nu{}_{\iota\kappa\rho} + D_\kappa G^\nu{}_{\iota\rho\mu} + D_\rho G^\nu{}_{\iota\mu\kappa} + \\
+ G^\nu{}_{\iota\beta\mu} Q^\beta{}_{\rho\kappa} + G^\nu{}_{\iota\beta\kappa} Q^\beta{}_{\mu\rho} + G^\nu{}_{\iota\beta\rho} Q^\beta{}_{\kappa\mu} \equiv 0
\end{aligned}
\tag{29}
$$

for torsion and curvature valid as a geometric identity.

Thus far, we have introduced the concept of tensor and the way to move its indices, which we recall were coordinate indices. Coordinate indices are important since they are the type of indices involved in differentiation. However, on the other hand, tensors in coordinate indices always feel the specificity of the coordinate system. Tensorial equations do remain formally the same in all coordinate system, but the tensors themselves change in content while changing the coordinate system. The only types of tensors which, also in content, remain the same in all of the coordinate systems are the tensors that are identically equal to zero and the scalars. Zero tensors offer little information, but scalars can be

used to build a formalism in which tensors can be rendered, both in form and in content, completely invariant. This formalism is known as Lorentz formalism.

In Lorentz formalism, the idea is that of introducing a basis of vectors $\xi_a^\alpha$ having two types of indices: one type of indices (Greek) is the usual coordinate index referring to the component of the vector, whereas the other type of indices (Latin) is a new Lorentz index referring to which vector of the basis we are considering. Under the point of view of coordinate transformations, the coordinate index ensures the transformation law of a vector, but clearly the other index ensures some different type of transformation that we will next find to be a Lorentz transformation.

Consider, for example, the tensor given by $T_{\alpha\sigma}$ and multiply it by two of the vectors $\xi_a^\alpha$ of the basis contracting the coordinate indices together: so $T_{\alpha\sigma}\xi_a^\alpha\xi_s^\sigma = T_{as}$ is an object that according to a coordinate transformation law does not transform at all, thus it is completely invariant, and this is exactly what we wanted. For one tensor with upper indices, the procedure would be the same but just made in terms of the covectors $\xi_\alpha^a$ as clear. Converting a coordinate index to a Lorentz index and then back to the coordinate index requires that $\xi_b^\alpha\xi_\alpha^c = \delta_b^c$ and $\xi_k^\alpha\xi_\sigma^k = \delta_\sigma^\alpha$ as a simple consistency condition. Finally, the operation for moving Lorentz indices is performed in terms of the metric tensor in Lorentz form $g_{\alpha\sigma}\xi_a^\alpha\xi_s^\sigma = g_{as}$, but, because we can always ortho-normalize the basis, the metric tensor in Lorentz form is just the Minkowskian matrix $g_{as} = \eta_{as}$ as it is well known indeed. Once the basis $\xi_a^\sigma$ is assigned, we may pass to another basis $\xi_a^{\prime\sigma}$ linked to the initial according to the transformation $\xi_a^{\prime\sigma} = \Lambda_a^b\xi_b^\sigma$ with $\Lambda_a^b$ chosen as to preserve the structure of the Minkowskian matrix and so such that it has to yield $\eta = \Lambda\eta\Lambda^T$ known as Lorentz transformation and justifying the name of the formalism.

In conclusion, after that, the coordinate tensors are converted into the Lorentz tensors, they are scalars under a general coordinate transformation but tensors under the Lorentz transformations. In doing so, we have converted the most general formalism into an equivalent formalism in which, however, the structure of the transformation now is very specific, and it can be made explicit. It is, in fact, known from the theory of Lie groups that any continuous transformation is writable according to

$$\Lambda = e^{\frac{1}{2}\sigma^{ab}\theta_{ab}}$$

in which $\theta_{ab} = -\theta_{ba}$ are the parameters while $\sigma_{ab} = -\sigma_{ba}$ are the generators and which verify specific commutation relationships that depend on the specific transformation alone. In the case of Lorentz transformation, it is known that we have six parameters and six generators given by

$$(\sigma_{ab})_j^i = \delta_a^i\eta_{jb} - \delta_b^i\eta_{ja}$$

and verifying

$$[\sigma_{ab}, \sigma_{cd}] = \eta_{ad}\sigma_{bc} - \eta_{ac}\sigma_{bd} + \eta_{bc}\sigma_{ad} - \eta_{bd}\sigma_{ac}$$

in general. While the generators are peculiar of this so-called real representation, the commutations relationship are meant to be a general character of the Lorentz transformation. As such, they will always be the same for any representation of Lorentz transformations. This shall be the Lorentz transformation that we will employ next.

We may condense everything into the following statements, starting from the fact that given a Lorentz transformation $\Lambda$ the set of functions $T_{r_1\ldots r_j}^{a_1\ldots a_i}$ transforming as

$$T_{r_1'\ldots r_n'}^{\prime a_1'\ldots a_m'} = (\Lambda^{-1})_{r_1'}^{r_1}\ldots(\Lambda^{-1})_{r_n'}^{r_n}(\Lambda)_{a_1}^{a_1'}\ldots(\Lambda)_{a_m}^{a_m'}T_{r_1\ldots r_n}^{a_1\ldots a_m} \tag{30}$$

is a *tensor in Lorentz formalism*. Compared to the coordinate formalism, symmetry properties and contractions, as well as all algebraic operations, are given analogously.

However, again, Lorentz transformations can be local and so differential operations must be defined by introducing a connection. As we have done before, the connection must be introduced in general in terms of its transformation.

Therefore, once again, we summarize by saying that the set of functions $\Omega^a_{b\mu}$ such that, under a general coordinate transformation, transforming as a lower Greek index vector and under a Lorentz transformation transforming as

$$\Omega'^{a'}_{b'\nu} = \Lambda^{a'}_a \left[ \Omega^a_{b\nu} - (\Lambda^{-1})^a_k (\partial_\nu \Lambda)^k_b \right] (\Lambda^{-1})^b_{b'} \tag{31}$$

is called *spin connection*, and no decomposition nor in particular any torsion can be defined as no transposition of indices of different types is defined. With it, we have

$$D_\mu T^{a_1 \ldots a_i}_{r_1 \ldots r_j} = \partial_\mu T^{a_1 \ldots a_i}_{r_1 \ldots r_j} + \sum_{k=1}^{k=i} \Omega^{a_k}_{p\mu} T^{a_1 \ldots p \ldots a_i}_{r_1 \ldots r_j} -$$
$$- \sum_{k=1}^{k=j} \Omega^p_{r_k \mu} T^{a_1 \ldots a_i}_{r_1 \ldots p \ldots r_j} \tag{32}$$

as *covariant derivative* of tensors in Lorentz formalism.

As we have anticipated, the passage to this formalism is done with the $\xi^a_\sigma$ and $\xi^\sigma_a$ vectors while the vertical movement of Latin indices is done with the $\eta^{ab}$ matrix.

Thus, the passage from general coordinate formalism to the Lorentz formalism is made via the introduction of the bases of vectors $\xi^a_\sigma$ and $\xi^\sigma_a$ dual of one another

$$\xi^a_\mu \xi^\mu_r = \delta^a_r \tag{33}$$
$$\xi^a_\mu \xi^\rho_a = \delta^\rho_\mu \tag{34}$$

called *tetrad fields* and such that they verify the pair of ortho-normality conditions given by

$$g^{\alpha\sigma} \xi^a_\alpha \xi^b_\sigma = \eta^{ab} \tag{35}$$
$$g_{\alpha\sigma} \xi^\alpha_a \xi^\sigma_b = \eta_{ab} \tag{36}$$

as $\eta$ are the *Minkowskian matrices*, preserved by Lorentz transformations. With the dual bases, ortho-normal with respect to the Minkowskian matrices, we can take a tensor in coordinate formalism with at least one Greek index and multiply it by the basis contracting one Greek index with the Greek index of the bases therefore obtaining the tensor in Lorentz formalism with a Latin index, and with a vertical movement of Latin indices which is performed in terms of the Minkowskian matrix as it is expected.

Notice that, if these two formalisms are perfectly equivalent, then their covariant derivatives should be equivalent and in particular we should be able from the most general connection to derive the spin connection. Upon requiring that $D_\mu \xi^\alpha_a = 0$ as well as $D_\mu \eta_{ab} = 0$, we have exactly this.

In fact, in terms of the most general coordinate connection and tetrad fields, we can always write

$$\Omega^a_{b\mu} = \xi^\nu_b \xi^a_\rho \left( \Gamma^\rho_{\nu\mu} - \xi^\rho_k \partial_\mu \xi^k_\nu \right) \tag{37}$$

antisymmetric in the Lorentz indices and such that, from it, we can derive the torsion tensor according to

$$Q^a_{\ \mu\nu} = -(\partial_\mu \xi^a_\nu - \partial_\nu \xi^a_\mu + \xi^b_\nu \Omega^a_{b\mu} - \xi^b_\mu \Omega^a_{b\nu}) \tag{38}$$

as it is easy to see, and we have that conditions (37) and $\Omega_{ab\mu} = -\Omega_{ba\mu}$ are respectively equivalent to

$$D_\mu \xi^r_\alpha = 0 \tag{39}$$
$$D_\mu \eta_{ab} = 0 \tag{40}$$

as general *coordinate-Lorentz compatibility conditions*.

In Lorentz formalism, from the spin connection, we get

$$G^a{}_{b\alpha\beta} = \partial_\alpha \Omega^a_{b\beta} - \partial_\beta \Omega^a_{b\alpha} + \Omega^a_{k\alpha} \Omega^k_{b\beta} - \Omega^a_{k\beta} \Omega^k_{b\alpha} \tag{41}$$

as the *curvature tensor*. Then, we have that

$$[D_\mu, D_\nu] T^{r_1 \cdots r_j} = Q^\eta{}_{\mu\nu} D_\eta T^{r_1 \cdots r_j} + $$
$$+ \sum_{k=1}^{k=j} G^{r_k}{}_{p\mu\nu} T^{r_1 \cdots p \cdots r_j} \tag{42}$$

is the general coordinate covariant *commutator of covariant derivatives* of the tensor field in Lorentz formalism.

As it should be expected by now, we have that

$$G^a{}_{b\mu\nu} = \xi^a_\alpha \xi^\beta_b G^\alpha{}_{\beta\mu\nu} \tag{43}$$

showing that the curvature tensor in Lorentz formalism is antisymmetric both in coordinate indices and in Lorentz indices, and so as a consequence the curvature also in this formalism has the same independent contractions which are therefore $G_{b\sigma} = G^a{}_{b\rho\sigma} \xi^\rho_a$ for the *Ricci curvature tensor* and $G = G_{a\sigma} \xi^\sigma_p \eta^{ap}$ for the *Ricci curvature scalar*.

After index renaming, we get

$$D_\mu G^a{}_{j\kappa\rho} + D_\kappa G^a{}_{j\rho\mu} + D_\rho G^a{}_{j\mu\kappa} + $$
$$+ G^a{}_{j\beta\mu} Q^\beta{}_{\rho\kappa} + G^a{}_{j\beta\kappa} Q^\beta{}_{\mu\rho} + G^a{}_{j\beta\rho} Q^\beta{}_{\kappa\mu} \equiv 0 \tag{44}$$

with curvature in Lorentz form as a geometric identity.

In this way, we conclude the introduction of the most general covariant formalism with the further conversion into the specific Lorentz formalism, in which the Lorentz transformation has been made explicit in terms of its real representation. We will soon see that another representation is possible. However, before this, we introduce gauge fields.

Our main goal is going to be focusing on the fact that fields may be complex, and so it makes sense to ask what symmetries can be established for these fields: if a field is complex, there arises the issue of phase transformations and, correspondingly, it is possible to construct a calculus that is in all aspects analogous to the one we just built.

Thus, given a real function $\alpha$, we have that a complex field that transforms according to the transformation

$$\phi' = e^{iq\alpha} \phi \tag{45}$$

is called *gauge field* of *q charge*, with algebraic operations defined as for geometric tensors.

Let it be given a covector field $A_\nu$ such that, for a phase transformation, it transforms according to the law

$$A'_\nu = A_\nu - \partial_\nu \alpha \tag{46}$$

then this vector is called *gauge potential*. With it,

$$D_\mu \phi = \partial_\mu \phi + iq A_\mu \phi \tag{47}$$

is said to be the *gauge derivative* of the gauge field.

For the gauge fields, we may introduce the operation of complex conjugation without the necessity of introducing any additional structure, and hence, for a gauge field of $q$ charge, the complex conjugate gauge field has $-q$ charge.

There is no decomposition of the gauge potential into more fundamental elements. In fact, complex conjugation is compatible with gauge derivatives automatically.

From the gauge connection, we define

$$F_{\alpha\beta} = \partial_\alpha A_\beta - \partial_\beta A_\alpha \tag{48}$$

that is such that $F = \partial A$ and so it is a tensor which is antisymmetric and invariant by a gauge transformation called *gauge strength*. With it, we have that

$$[D_\mu, D_\nu]\phi = iqF_{\mu\nu}\phi \tag{49}$$

is the *commutator of gauge derivatives* of gauge fields.

Clearly, the gauge strength cannot be decomposed in terms of more fundamental underlying structures.

Furthermore, we have that

$$\partial_\nu F_{\alpha\sigma} + \partial_\sigma F_{\nu\alpha} + \partial_\alpha F_{\sigma\nu} = 0 \tag{50}$$

or equivalently $\partial F = 0$ as a gauge geometric identity.

There is a point that needs to be elucidated regarding the definition of the Maxwell strength. As this expression can be generalized up to

$$F_{\alpha\beta} = \nabla_\alpha A_\beta - \nabla_\beta A_\alpha,$$

then one may wonder if some non-minimal coupling could be invoked to write it as

$$F'_{\alpha\beta} = D_\alpha A_\beta - D_\beta A_\alpha = \nabla_\alpha A_\beta - \nabla_\beta A_\alpha + Q_{\alpha\beta\rho}A^\rho$$

which would violate gauge invariance. Therefore, which one between $F_{\alpha\beta}$ and $F'_{\alpha\beta}$ should be considered? The answer is actually quite simple conceptually, and it is that, in a theory of electrodynamics established within a purely geometric context, the Maxwell strength is not just a curl of a vector but the specific curl of a vector that comes as the formal expression of the curvature of two covariant derivatives. In this sense, it is clear that $F'_{\alpha\beta}$ as compared to $F_{\alpha\beta}$ has a lesser geometric meaning. Moreover, the form $F_{\alpha\beta}$ is also the one for which the geometric identities (50) called Cauchy identities are valid. In addition, so this is the only form that will interest us in the following.

This concludes the introduction of gauge fields, based on a parallel with geometric tensor. We shall now move to a following part in which these two formalisms will be merged into a single one known as spinorial formalism.

### 2.1.2. Spinorial Fields

In the previous parts, we have introduced tensor fields and the way to pass from coordinate into Lorentz indices, specifying that, with such a conversion, we also had the conversion of the most general coordinate transformation into the specific Lorentz transformation: the advantage of this specific Lorentz transformation is that, although it had been introduced in real representation, nevertheless, it can also be written in other representations like most notably the complex representation. In such representation, we will see that gauge fields find place naturally.

In order to find a Lorentz transformation in complex representation, we specify that these transformations are classified by semi-integer labels known as spin, and here we consider the simplest $\frac{1}{2}$-spin case: so, for the complex generators, we select those whose irreducible form is given in terms of two-dimensional matrices. General results from the theory of Lie groups tell us that the Lorentz transformation can be written according to the following form:

$$\boldsymbol{\Lambda} = e^{\frac{1}{2}\sigma^{ab}\theta_{ab}}$$

where $\theta^{ab} = -\theta^{ba}$ are the parameters as given above and $\sigma_{ab} = -\sigma_{ba}$ are the generators verifying

$$[\sigma_{ab}, \sigma_{cd}] = \eta_{ad}\sigma_{bc} - \eta_{ac}\sigma_{bd} + \eta_{bc}\sigma_{ad} - \eta_{bd}\sigma_{ac}$$

as commutation relationships. The actual form for these Lorentz generators in the case of the complex irreducible two-dimensional matrices is known to be given in terms of the Pauli matrices

$$\sigma^1 = \begin{pmatrix} 0 & 1 \\ 1 & 0 \end{pmatrix} \quad \sigma^2 = \begin{pmatrix} 0 & -i \\ i & 0 \end{pmatrix} \quad \sigma^3 = \begin{pmatrix} 1 & 0 \\ 0 & -1 \end{pmatrix}$$

according to

$$\sigma_{\pm}^{0A} = \pm\tfrac{1}{2}\sigma^A$$
$$\sigma_{AB} = -\tfrac{i}{2}\varepsilon_{ABC}\sigma^C$$

as a straightforward check would demonstrate. We notice that, in the passage from real to complex representation, a two-fold multiplicity has arisen since two opposite expressions are possible for the boosts and thus for the Lorentz transformation in full. This ambiguity can be overcome by having these two irreducible two-dimensional generators merged into a single reducible four-dimensional generators

$$\sigma^{0A} = \tfrac{1}{2}\begin{pmatrix} -\sigma^A & 0 \\ 0 & \sigma^A \end{pmatrix}$$
$$\sigma_{AB} = -\tfrac{i}{2}\varepsilon_{ABC}\begin{pmatrix} \sigma^C & 0 \\ 0 & \sigma^C \end{pmatrix}$$

which still verify the Lorentz commutation algebra. Such a merging also has the advantage that, with four-dimensional matrices, it is possible to introduce

$$\begin{pmatrix} 0 & \mathbb{I} \\ \mathbb{I} & 0 \end{pmatrix} = \gamma^0$$
$$\begin{pmatrix} 0 & \sigma^K \\ -\sigma^K & 0 \end{pmatrix} = \gamma^K$$

in terms of which the four-dimensional generators are

$$\sigma^{ab} = \tfrac{1}{4}\left[\gamma^a, \gamma^b\right]$$

and where

$$\{\gamma^a, \gamma^b\} = 2\eta^{ab}\mathbb{I}$$

in terms of the Minkowskian matrix. This way of writing four-dimensional matrices constitutes an advantage because we can see a manifest (1 + 3)-dimensional space–time form in the last two expressions. Then, it is possible to employ these last two expressions to derive a whole list of useful identities involving these matrices. To begin, we have

$$\sigma_{ab} = -\tfrac{i}{2}\varepsilon_{abcd}\pi\sigma^{cd}$$

which implicitly defines the $\pi$ matrix. This matrix is the one usually indicated like a gamma with an index five as originally it was used to study five-dimensional theories, but, because we will always be in the space–time, the index five for us has no meaning and

so we will use a notation with no index at all. Notice that, with this definition, we have extinguished all possible matrices since the matrices

$$\mathbb{I} \qquad \gamma^a \qquad \sigma^{ab} \qquad \gamma^a \pi \qquad \pi$$

are 16 linearly independent matrices spanning the space of four-dimensional matrices, and so they form a basis for such a space. These matrices are called Clifford matrices and they will have great importance. We have that

$$\gamma_0 \gamma_a^\dagger \gamma_0 = \gamma_a$$
$$\gamma_0 \sigma_{ab}^\dagger \gamma_0 = -\sigma_{ab}$$
$$\pi^\dagger = \pi$$

specifying the behavior of the Clifford matrices under conjugation. By direct inspection, one can easily see that

$$\gamma_a \gamma_b = \eta_{ab}\mathbb{I} + 2\sigma_{ab}$$

as well as

$$\gamma_i \gamma_j \gamma_k = \gamma_i \eta_{jk} - \gamma_j \eta_{ik} + \gamma_k \eta_{ij} + i\varepsilon_{ijkq}\pi\gamma^q$$

showing that products of, however, many gamma matrices can always be reduced to the product of at most two of them. Therefore, there is no need to compute the product of three or more gamma matrices. Because $\varepsilon_{0123} = 1$, we have $\pi = i\gamma^0\gamma^1\gamma^2\gamma^3$ and so

$$\{\pi, \gamma_a\} = 0$$
$$[\pi, \sigma_{ab}] = 0$$

as expected. In fact, this representation is reducible, and then Schur's lemma ensures us that there must exist one matrix different from the identity commuting with all the generators of the group. By working with all the previous identities, one can find

$$[\gamma_i, \sigma_{jk}] = \gamma_k \eta_{ij} - \gamma_j \eta_{ik}$$
$$\{\gamma_i, \sigma_{jk}\} = i\varepsilon_{ijkq}\pi\gamma^q$$

and similarly

$$\{\sigma_{ab}, \sigma_{cd}\} = \tfrac{1}{2}\left(\eta_{ad}\eta_{bc}\mathbb{I} - \eta_{ac}\eta_{bd}\mathbb{I} + i\varepsilon_{abcd}\pi\right)$$
$$[\sigma_{ab}, \sigma_{cd}] = \eta_{ad}\sigma_{bc} - \eta_{ac}\sigma_{bd} + \eta_{bc}\sigma_{ad} - \eta_{bd}\sigma_{ac}$$

as other fundamental identities. The list may go on, but, for our purposes, there is no need to reach products with more gamma matrices. The last identity tells us that the $\sigma_{ab}$ matrices are the generators of the Lorentz algebra as expected. As already said, the parameters are the same we had in the Lorentz formalism since real and complex representations are merely two different forms of the same transformation. This transformation is thus given by

$$\mathbf{\Lambda} = e^{\frac{1}{2}\sigma^{ab}\theta_{ab}}$$

in its most general form. However, in view of studying complex fields, we know that the complex phase transformation $e^{iq\alpha}$ must also be introduced. Therefore, we have

$$\mathbf{\Lambda}e^{iq\alpha} = e^{\left(\frac{1}{2}\sigma^{ab}\theta_{ab} + iq\alpha\mathbb{I}\right)} = \mathbf{S}$$

as the Lorentz-phase transformation in its most complete form possible. This form is also called spinorial transformation. It is what we will employ to define the spinorial fields $\psi$ as

the column of four complex functions that are scalars for coordinate transformations while transforming according to $\psi' = S\psi$ under the spinorial transformations.

We may now summarize by saying that, given the most general spinorial transformation $S$, the column and row of complex scalars $\psi$ and $\overline{\psi}$ transforming as

$$\psi' = S\psi \qquad \overline{\psi}' = \overline{\psi}S^{-1} \tag{51}$$

are called *spinorial fields*. Operations of sum and product respect spinor transformation.

As above, the transformation $S$ is local and so we have to introduce the spinorial connection defined in terms of the transformation law that guarantees the derivative to be covariant for general spinorial transformations.

Therefore, we have that the coefficients $\mathbf{\Omega}_\nu$ transforming according to

$$\mathbf{\Omega}'_\nu = S\left(\mathbf{\Omega}_\nu - S^{-1}\partial_\nu S\right)S^{-1} \tag{52}$$

are called *spinorial connection*. Once the spinorial connection is assigned, we have that

$$D_\mu\psi = \partial_\mu\psi + \mathbf{\Omega}_\mu\psi \qquad D_\mu\overline{\psi} = \partial_\mu\overline{\psi} - \overline{\psi}\mathbf{\Omega}_\mu \tag{53}$$

are the *covariant derivatives* of the spinorial fields.

We now give a list of properties of the Clifford matrices.

We have that the *Clifford matrices* $\gamma^a$ such that

$$\mathbf{\Lambda}\gamma^b\mathbf{\Lambda}^{-1} \equiv (\Lambda^{-1})^b_a\gamma^a \tag{54}$$

verify the anticommutation relationships

$$\{\gamma_a, \gamma_b\} = 2\eta_{ab}\mathbb{I} \tag{55}$$

so that we can define the matrices $\sigma_{ab}$ as

$$\sigma_{ab} = \tfrac{1}{4}[\gamma_a, \gamma_b] \tag{56}$$

and

$$\sigma_{ab} = -\tfrac{i}{2}\varepsilon_{abcd}\pi\sigma^{cd} \tag{57}$$

for the $\pi$ matrix to be implicitly defined. Then,

$$\gamma_0\gamma_a^\dagger\gamma_0 = \gamma_a \tag{58}$$
$$\gamma_0\sigma_{ab}^\dagger\gamma_0 = -\sigma_{ab} \tag{59}$$
$$\pi^\dagger = \pi \tag{60}$$

alongside the square properties

$$\gamma_a\gamma^a = 4\mathbb{I} \tag{61}$$
$$\sigma_{ab}\sigma^{ab} = -3\mathbb{I} \tag{62}$$
$$\pi^2 = \mathbb{I} \tag{63}$$

together with the anticommutation properties

$$\{\pi, \gamma_a\} = 0 \tag{64}$$
$$\{\gamma_i, \sigma_{jk}\} = i\varepsilon_{ijkq}\pi\gamma^q \tag{65}$$

and the commutation properties

$$[\boldsymbol{\pi}, \sigma_{ab}] = 0 \tag{66}$$

$$[\gamma_a, \sigma_{bc}] = \eta_{ab}\gamma_c - \eta_{ac}\gamma_b \tag{67}$$

$$[\sigma_{ab}, \sigma_{cd}] = \eta_{ad}\sigma_{bc} - \eta_{ac}\sigma_{bd} + \eta_{bc}\sigma_{ad} - \eta_{bd}\sigma_{ac} \tag{68}$$

as well as

$$\gamma_a\gamma_b = \eta_{ab}\mathbb{I} + 2\sigma_{ab} \tag{69}$$

$$\gamma_i\gamma_j\gamma_k = \gamma_i\eta_{jk} - \gamma_j\eta_{ik} + \gamma_k\eta_{ij} + i\varepsilon_{ijkq}\boldsymbol{\pi}\gamma^q \tag{70}$$

all being spinorial identities. Employing $\gamma_0$, we can define

$$\overline{\psi} = \psi^\dagger\gamma_0 \qquad \gamma_0\overline{\psi}^\dagger = \psi \tag{71}$$

as the *spinor conjugation*. In particular, we have

$$\boldsymbol{\pi}_L = \tfrac{1}{2}(\mathbb{I} - \boldsymbol{\pi}) \tag{72}$$

$$\boldsymbol{\pi}_R = \tfrac{1}{2}(\mathbb{I} + \boldsymbol{\pi}) \tag{73}$$

as *left-handed/right-handed chiral projectors*. They verify

$$\boldsymbol{\pi}_L^\dagger = \boldsymbol{\pi}_L \tag{74}$$

$$\boldsymbol{\pi}_R^\dagger = \boldsymbol{\pi}_R \tag{75}$$

alongside

$$\boldsymbol{\pi}_L^2 = \boldsymbol{\pi}_L \tag{76}$$

$$\boldsymbol{\pi}_R^2 = \boldsymbol{\pi}_R \tag{77}$$

together with

$$\boldsymbol{\pi}_L\boldsymbol{\pi}_R = \boldsymbol{\pi}_R\boldsymbol{\pi}_L = 0 \tag{78}$$

and such that

$$\boldsymbol{\pi}_L + \boldsymbol{\pi}_R = \mathbb{I} \tag{79}$$

in general. We can also define

$$\boldsymbol{\pi}_L\psi = \psi_L \qquad \overline{\psi}\boldsymbol{\pi}_R = \overline{\psi}_L \tag{80}$$

$$\boldsymbol{\pi}_R\psi = \psi_R \qquad \overline{\psi}\boldsymbol{\pi}_L = \overline{\psi}_R \tag{81}$$

and

$$\overline{\psi}_L + \overline{\psi}_R = \overline{\psi} \qquad \psi_L + \psi_R = \psi \tag{82}$$

as *left-handed/right-handed chiral parts*. With the pair of conjugate spinors, we define the *bi-linear spinorial quantities* according to

$$2\overline{\psi}\sigma^{ab}\pi\psi = \Sigma^{ab} \tag{83}$$

$$2i\overline{\psi}\sigma^{ab}\psi = M^{ab} \tag{84}$$

$$\overline{\psi}\gamma^a\pi\psi = S^a \tag{85}$$

$$\overline{\psi}\gamma^a\psi = U^a \tag{86}$$

$$i\overline{\psi}\pi\psi = \Theta \tag{87}$$

$$\overline{\psi}\psi = \Phi \tag{88}$$

such that they are all real tensor quantities. From them,

$$\psi\overline{\psi} \equiv \tfrac{1}{4}\Phi\mathbb{I} + \tfrac{1}{4}U_a\gamma^a + \tfrac{i}{8}M_{ab}\sigma^{ab} -$$
$$- \tfrac{1}{8}\Sigma_{ab}\sigma^{ab}\pi - \tfrac{1}{4}S_a\gamma^a\pi - \tfrac{i}{4}\Theta\pi \tag{89}$$

from which we get the relationships

$$2U_\mu S_\nu\sigma^{\mu\nu}\pi\psi + U^2\psi = 0 \tag{90}$$

$$i\Theta S_\mu\gamma^\mu\psi + \Phi S_\mu\gamma^\mu\pi\psi + U^2\psi = 0 \tag{91}$$

and

$$U_a\gamma^a\psi = -S_a\gamma^a\pi\psi = (\Phi\mathbb{I} + i\Theta\pi)\psi \tag{92}$$

as well as the relationships

$$\Sigma^{ab} = -\tfrac{1}{2}\varepsilon^{abij}M_{ij} \tag{93}$$

$$M^{ab} = \tfrac{1}{2}\varepsilon^{abij}\Sigma_{ij} \tag{94}$$

and

$$M_{ab}\Phi - \Sigma_{ab}\Theta = U^j S^k \varepsilon_{jkab} \tag{95}$$

$$M_{ab}\Theta + \Sigma_{ab}\Phi = U_{[a}S_{b]} \tag{96}$$

with

$$M_{ik}U^i = \Theta S_k \tag{97}$$

$$\Sigma_{ik}U^i = \Phi S_k \tag{98}$$

$$M_{ik}S^i = \Theta U_k \tag{99}$$

$$\Sigma_{ik}S^i = \Phi U_k \tag{100}$$

and also

$$\tfrac{1}{2}M_{ab}M^{ab} = -\tfrac{1}{2}\Sigma_{ab}\Sigma^{ab} = \Phi^2 - \Theta^2 \tag{101}$$

$$U_aU^a = -S_aS^a = \Theta^2 + \Phi^2 \tag{102}$$

$$\tfrac{1}{2}M_{ab}\Sigma^{ab} = -2\Theta\Phi \tag{103}$$

$$U_aS^a = 0 \tag{104}$$

called *Fierz re-arrangements* of spinor fields. If both scalars $\Theta$ and $\Phi$ do not vanish identically, we can always find a special frame where the most general spinor is written as

$$\psi = \phi e^{-\frac{i}{2}\beta\pi} e^{-i\alpha} \begin{pmatrix} 1 \\ 0 \\ 1 \\ 0 \end{pmatrix} \qquad (105)$$

up to the reversal of the third axis and up to the discrete transformation $\psi \rightarrow \pi\psi$ and called *polar form*. From this, we can write

$$\Sigma^{ab} = 2\phi^2 (\cos\beta u^{[a} s^{b]} - \sin\beta u_j s_k \varepsilon^{jkab}) \qquad (106)$$

$$M^{ab} = 2\phi^2 (\cos\beta u_j s_k \varepsilon^{jkab} + \sin\beta u^{[a} s^{b]}) \qquad (107)$$

in terms of

$$S^a = 2\phi^2 s^a \qquad (108)$$

$$U^a = 2\phi^2 u^a \qquad (109)$$

and

$$\Theta = 2\phi^2 \sin\beta \qquad (110)$$

$$\Phi = 2\phi^2 \cos\beta \qquad (111)$$

showing that the fields $\phi$ and $\beta$ are a scalar and a pseudo-scalar, respectively. Then,

$$u_a u^a = -s_a s^a = 1 \qquad (112)$$

$$u_a s^a = 0 \qquad (113)$$

showing that the normalized velocity vector $u^a$ and the normalized spin axial-vector $s^a$ possess three independent components each. This means that $\phi$ and $\beta$ are the only true real scalar degrees of freedom and called *module* and *Yvon–Takabayashi angle*. The reader interested in details for all these statements can have a look at [15].

The conditions of compatibility now read $D_\mu \gamma_a = 0$ in general: if the spinorial matrix also has a tensorial index, the covariant derivative is to be completed to the form

$$\boldsymbol{D}_\mu \boldsymbol{B}_a = \partial_\mu \boldsymbol{B}_a - \boldsymbol{B}_b \Omega^b{}_{a\mu} + [\boldsymbol{\Omega}_\mu, \boldsymbol{B}_a]$$

which can be taken for the gamma matrix and hence implementing the above condition, and recalling that these matrices in Lorentz indices are constants, yields

$$-\gamma_b \Omega^b{}_{a\mu} + [\boldsymbol{\Omega}_\mu, \gamma_a] = 0$$

as a relation among connections. By writing a general

$$\boldsymbol{\Omega}_\mu = a\Omega^{ij}{}_\mu \sigma_{ij} + A_\mu$$

and plugging it into the above relation, we obtain that

$$-\gamma_b \Omega^b{}_{k\mu} + a\Omega^{ij}{}_\mu [\sigma_{ij}, \gamma_k] + [A_\mu, \gamma_k] = 0$$

and with $[\sigma_{ij}, \gamma_k] = \eta_{kj}\gamma_i - \eta_{ki}\gamma_j$ we get $a = 1/2$ and

$$[A_\mu, \gamma_5] = 0$$

telling that $A_\mu$ must commute with all gamma matrices, and thus, with all possible matrices, implying that it must be proportional to the identity matrix. Writing it as

$$A_\mu = (pC_\mu + ibA_\mu)\mathbb{I},$$

it is possible to see that, for $b = q$, it is possible to interpret the vector $A_\mu$ as the gauge potential. Because the other term is related to conformal transformations, which are not symmetries in our case, we set $p = 0$ in general. Then, we have that, all considered, we may write the expression

$$\mathbf{\Omega}_\mu = \tfrac{1}{2}\Omega^{ij}{}_\mu \sigma_{ij} + iqA_\mu\mathbb{I}$$

as the most general form of spinorial connection.

To summarize, we have that the most general spinorial connection is given by

$$\mathbf{\Omega}_\mu = \tfrac{1}{2}\Omega_{ab\mu}\sigma^{ab} + iqA_\mu\mathbb{I} \tag{114}$$

in terms of the generator-valued spin connection and the gauge potential, and this is equivalent to the fact that the spinorial covariant derivatives of the gamma matrices are

$$\mathbf{D}_\mu\gamma_a = 0 \tag{115}$$

vanishing identically, as it is quite straightforward to see.

We have that, from the spinorial connection, we define

$$\mathbf{F}_{\alpha\beta} = \partial_\alpha\mathbf{\Omega}_\beta - \partial_\beta\mathbf{\Omega}_\alpha + [\mathbf{\Omega}_\alpha, \mathbf{\Omega}_\beta] \tag{116}$$

as the *spinorial curvature*. With it,

$$[\mathbf{D}_\mu, \mathbf{D}_\nu]\psi = Q^\alpha{}_{\mu\nu}\mathbf{D}_\alpha\psi + \mathbf{F}_{\mu\nu}\psi \tag{117}$$

as *commutator of covariant derivatives* of spinor fields.

Correspondingly, the curvature is decomposable as

$$\mathbf{F}_{\mu\nu} = \tfrac{1}{2}G_{ab\mu\nu}\sigma^{ab} + iqF_{\mu\nu}\mathbb{I} \tag{118}$$

with the curvature tensor and gauge strength.

For a final step, we have

$$\mathbf{D}_\mu\mathbf{F}_{\kappa\rho} + \mathbf{D}_\kappa\mathbf{F}_{\rho\mu} + \mathbf{D}_\rho\mathbf{F}_{\mu\kappa} +$$
$$+ \mathbf{F}_{\beta\mu}Q^\beta{}_{\rho\kappa} + \mathbf{F}_{\beta\kappa}Q^\beta{}_{\mu\rho} + \mathbf{F}_{\beta\rho}Q^\beta{}_{\kappa\mu} \equiv 0 \tag{119}$$

as spinorial geometrical identities holding in general.

We conclude with some fundamental comments: the first and most important one is about the fact that so far we have encountered three types of transformation laws: the first type was the most general coordinate transformation; the second type was the gauge transformation; the third type was the specific Lorentz transformation, which was given in real representation for tensors and complex representation for spinors. The coordinate transformation is known as *passive transformation*; the Lorentz transformation in real representation as well as the Lorentz transformation in complex representation merged with the gauge transformation that is the spinor transformation, are known altogether as *active transformations*. Because they have the very same parameters, we then have that both active transformations have to be performed simultaneously.

Another interesting comment is on the connections and how they are built: the torsion tensor, when the metric tensor is used, gives the connection (16); this connection, when the dual bases of tetrad fields are employed, gives the spin connection (37); this spin connection,

when the gamma matrices and their commutators are considered, with the gauge potential, when multiplied by the identity matrix, give the spinorial connection (114). Remarkably, all fields fit within the most general spinorial connection, with no room for anything else: this circumstance can be seen as a sort of geometric unification of all the physical fields that are involved. On the other hand, however, in order to see it that way, we have to wait until we interpret these geometric quantities.

A final comment regards the structure of the covariant commutator (117), in which, by interpreting the covariant derivative as the covariant generators of translations, one sees that the completely antisymmetric torsion plays the role that in Lie group theory is played by the completely antisymmetric structure coefficients; we also recall to the reader that, in the curvature, there appear sigma matrices which are the generators of the Lorentz transformations and therefore of the space–time rotations. An additional interpretation that can be assigned to the covariant commutator is that, when a field is moved around, it would fail to go back to the starting point and have the initial orientation. A position mismatch is measured by torsion and a directional mismatch is measured by curvature, and this is why torsion is also said to describe the dislocations while curvature is also said to describe the disclinations of a round trip. This shows intuitively that both torsion and curvature have to be accounted for the most general description of space–time.

For some introduction to the general theory of spinors and their classifications, we refer the readers to [16,17].

*Geometry and Matter in Interaction*

Now that, in terms of general symmetry arguments, we have completed the definitions of all geometric quantities for the kinematic background, the next step is to have them coupled to one another in order to assign their dynamics.

2.1.3. Covariant Field Equations

When in 1916 Einstein wanted to construct the theory of gravitation, the idea he wished to follow was inspired geometrically, based on the principle of equivalence.

The principle of equivalence states the equivalence at a local level between inertia and gravitation, in the sense that locally inertial and gravitational forces can simulate one another so well that, when both present, their effects can be made to cancel: it can be stated by saying that one can always find a system of coordinates in which locally the accelerations due to gravitation are negligible.

On the other hand, one can demonstrate a theorem originally due to Weyl whose statement sounds analogous: it states that one can always find a system of coordinates in which in a point the symmetric part of the connection vanishes.

In the previous sections, we have discussed in what way the condition of complete antisymmetry of torsion gives rise to a unique symmetric part of the connection, thus removing any possible ambiguity in the implementation of Weyl theorem: hence, for a completely antisymmetric torsion, the Weyl theorem is the mathematical implementation of the principle of equivalence insofar as the acceleration due to gravitation is encoded within the symmetric connection. A unique symmetric connection corresponds to a uniquely defined gravitational field as our physical intuition would suggest. Furthermore, the single symmetric connection is entirely written in terms of the derivatives of the metric, and therefore, if the gravitational field is encoded within the symmetric connection, then the gravitational potential is encoded within the metric tensor.

The metric tensor is a tensor, but it cannot vanish and none of its derived scalar is non-trivial, and the connection is not a tensor, so they will always depend on the choice of coordinates: hence, the information about gravity will always be intertwined with inertial information, which is not a surprise, since after all we know, they are locally indistinguishable. On the other hand, we wish to have a way to tell gravity apart from inertial information, and, to do that, it is necessary to take a less local level, then considering the Riemann curvature tensor: if gravity is contained in the metric tensor as well as in

the connection, then it is contained in the Riemann curvature tensor too, but the Riemann curvature tensor is a tensor from which non-trivial scalars can be derived or which can be vanished, and this is what makes it able to discriminate gravity from inertial forces. If the metric is Minkowskian and the connection is zero, we cannot know whether this is because gravity is absent or compensated by inertial forces, and, similarly, if the metric is not Minkowskian and the connection is not zero, we cannot know whether this is because gravity is present or simulated by inertial forces as above. However, if the Riemann curvature tensor is zero, we know it is because gravity is absent, and, if the Riemann curvature tensor is not zero, we know gravity is present in general terms. This has to be so, as there can not be any compensation due to inertial forces since there can be no inertial forces, within the Riemann curvature tensor.

Therefore, the principle of equivalence is the manifestation of the interpretative principle telling that gravitation is geometrized, and this is so as a consequence of the fact that gravity alone is contained in the Riemann curvature.

This statement has to be taken into account together with the parallel fact that, in Einstein relativity, the mass is a form of energy, as it is very well known indeed.

Putting the two things together, it becomes clear that the gravitational field equations that were given in terms of a second-order differential operator of the gravitational potential proportional to the mass density have to be considered as an approximated form of a more general set of gravitational field equations given by a certain linear combination of the curvature proportional to the energy.

The energy density is a tensor having two indices and therefore the curvature we are looking for must have two indices as well, which tells that we need the contraction of the Riemann curvature given by the Ricci curvature.

In 1916, all matter forms that were known consisted of macroscopic fluids, scalars, and electro-dynamic fields, all of which have an energy density symmetric in the two indices. This may be a problem as the Ricci curvature is not symmetric.

In addition, this is where Einstein assumption of the vanishing torsion came about: assuming torsion to be equal to zero meant that a specific linear combination of the Ricci curvatures were symmetric, and thus proportional to the energy.

To see this, consider identity (29) in the case in which torsion vanishes. Its full contraction gives, in the most general case, the following identity:

$$\nabla_\mu (R^{\mu\nu} - \tfrac{1}{2}g^{\mu\nu}R - g^{\mu\nu}\Lambda) = 0$$

where the object in parenthesis is symmetric indeed, and so it can be taken to be proportional to the energy density.

Now, Einstein geometrical insight is expressed by the gravitational field equations

$$R^{\mu\nu} - \tfrac{1}{2}g^{\mu\nu}R - g^{\mu\nu}\Lambda = \tfrac{1}{2}kE^{\mu\nu} \tag{120}$$

called Einstein field equations: from them, it follows that the energy density verifies $E^{\mu\nu} = E^{\nu\mu}$ and $\nabla_\mu E^{\mu\nu} = 0$ as is well known.

Therefore, Einstein field equations are the most general linear combination of curvatures for which geometric identities imply the validity of the symmetry and conservation law for the energy of matter. In this sense, the field equations are established on the bases of their conservation laws, themselves obtained from geometric identities, and this is what represents the Einstein spirit of geometrization—at least in the most general case without torsion.

Then, one might wonder what happens if torsion were not neglected.

The first thing we would have to keep in mind is that, in this case, geometry would provide both a curvature and a torsion tensor. The second point to be retained is that the Einsteinian gravitational theory is based on the fact that the curvature tensor is sourced by the energy. Putting things together, we should expect in the presence of torsion that

there be another conserved quantity in parallel to the energy and another field equation coupling such a conserved quantity to the torsion tensor itself.

Such a quantity, however, is already at hand.

In 1928, Dirac was the first to describe a system of matter fields, named spinors, which possessed an energy together with a spin, and this is the quantity we are seeking.

In a torsional completion of the theory of gravitation, matter fields described by both an energy and a spin can naturally find a place when the spin is coupled to torsion much in the same way in which the energy is coupled to curvature. For such a theory, the full system of field equations is given by the spin–torsion field equations, which simply spell the proportionality between torsion and spin, called Sciama–Kibble field equations, alongside the curvature–energy field equations, which are formally the same as in Einstein gravity, and therefore still called Einstein field equations. Altogether, they are known under the name of Einstein–Sciama–Kibble ESK field equations [18–20].

However, contrary to what is believed, the ESK field equations are actually not the most general either because, while torsion and gravitation are independent, their field equations have the same coupling constant, and this accounts for an arbitrary restriction.

If we want independent fields to have independent coupling to their independent sources, we must find a way to obtain the ESK field equations generalized so that the two coupling constants are different.

We will not spend time on the mathematical details of this generalization, but the interested reader can find such generalized system of field equations in the case of two different coupling constants in [21].

However, then again, this is not still the most general system of field equations because the torsion tensor enters algebraically in its coupling to the spin density tensor.

As mentioned, the above system of field equations has the feature that torsion and spin are algebraically related and this constitutes a conceptual problem because in the case in which the spin density were to vanish, then torsion would vanish too, with no possibility to propagate, and hence the torsion tensor would be unphysical.

That the torsion–spin coupling is algebraic might not be seen as a problem because also the curvature–energy coupling is algebraic, but there are reasons for this situation not be to entirely analogous: the most important is that the torsion that enters in the field equations is the general Cartan torsion, with the consequence that, if the spin density were to be vanishing everywhere the Cartan torsion would be vanishing as well, but the curvature that enters the field equations is the Ricci curvature and not the Riemann curvature, with the consequence that, even if the energy density were to be vanishing everywhere, the Ricci curvature would also be vanishing, but this would not imply that the Riemann curvature would be equal to zero, and gravity may still be present.

In addition, the curvature has an internal structure given in terms of first-order derivatives of the connection and thus in terms of second-order derivatives of the metric tensor, so that there exists a dynamics for the gravitational field, unlike for torsion.

If we desire that the torsion dynamics be implemented in the theory, then we have to look for dynamical terms in the torsion–spin field equations, and also for torsional contribution in all of the other field equations as well.

We specify that our main goal is following the Einstein spirit of geometrization, and, in order to do so, we are going to obtain the field equations for the theory in a genuinely geometric way by finding the most general form of the field equations that is compatible with the constraints given by underlying geometric identities.

In order to construct the most general system of field equations, we are going to start by distinguishing them into two different types: the field equations for the geometry–matter coupling, which shall be written in the form of second-order derivatives of the metric and torsion and also gauge potentials equal to sources given by the energy and spin and also the current of fields; and the matter field equations, which will be written in the form of a first-order differential operator containing metric and torsion and gauge potentials acting on the spinor field and equalling the spinor field itself. This discrimination

comes from the fact that, on the one hand, it is possible to employ spinors to construct sources for the tensor and gauge field equations, but, on the other hand, it is not possible to use tensor and gauge fields to build sources of the spinorial field equations. In the spinorial field equations, the derivatives of the spinor field must be proportional to the spinor field itself. This discrimination between the form of geometric and matter field equations is therefore intrinsic to the structure of the fields we use.

We start by considering the fact that field equations for the metric have to be in the form of some derivative of the metric equal to some source: because the covariant derivative of the metric tensor vanishes identically, then any dynamics of the metric can only be described in terms of the partial derivatives of the metric, or, equivalently, by the metric connection (15). Again, the metric connection is not a tensor, and the only way we have from the symmetric connection to form a tensor is to take another partial derivative, therefore forming the metric curvature tensor as given by (22). As Equation (27) shows, the metric curvature tensor is one peculiar combination of second-order partial derivatives of the metric, that is, arguments of symmetry under the most general coordinate transformations force at least second-order derivatives of the metric in the differential field equations. Then, arguments of simplicity would require that we do not take any further differential structure. In the following, we will see that second-order derivatives in the metric field equations endow them with a character that no other field equation will have, rendering them somewhat peculiar indeed.

For the moment, what we have established is that the metric field equations will have to be given in the form of some combination of the metric curvature tensor, and to see what combination, we start from considering that, if the leading term were to be given by the Riemann metric curvature tensor $R^{\alpha\tau\sigma\nu}$, then the vacuum equations would reduce to the condition of vanishing of Riemann metric curvature tensor, so that they would imply that there only be the trivial metric. Hence, if we want non-trivial metrics to be possible in vacuum, then the Riemann metric curvature tensor must appear contracted as the Ricci metric curvature tensor $R^{\alpha\mu}$ for leading term, and of course we may have contractions such as the Ricci metric curvature scalar $Rg_{\alpha\mu}$ or even $\Lambda g_{\alpha\mu}$ as sub-leading terms in general: as we have already seen above, the most general form of linear combination of curvatures in the field equations is given by (120), in which the only constant $\Lambda$ is still undetermined, and it will remain undetermined since there is no way to fix it on geometrical grounds. Thus, we might well think of it as a generic integration constant, which can always be added and whose value cannot be fixed.

We now turn our attention to the other field equations, for which the covariant derivatives of the fields will not be identically zero.

The field equations for the torsion have to be in the form of covariant derivatives of the torsion axial-vector equal to some source: taking covariant derivatives of the torsion axial-vector implies that we will have to write the field equation in the form of the covariant divergence of the torsion axial-vector equal to a source constituted by a pseudo-scalar field, but the temporal derivative will be specified for the temporal component of the torsion axial-vector solely. In addition, thus, we must take two covariant derivatives of the torsion axial-vector as a leading term.

To assess what are the most general field equations for the torsion axial-vector, we consider that the leading term given in the form of two covariant derivatives of the torsion axial-vector $\nabla_\sigma\nabla_\alpha W_\rho$ is to be such that one of the indices of the derivatives has to be contracted yielding the two forms $\nabla_\sigma\nabla^\sigma W_\rho$ and $\nabla_\rho\nabla_\sigma W^\sigma$ as leading terms: sub-leading terms may be added eventually and so we may establish the most general form of field equations as

$$
\begin{aligned}
&2\Pi\nabla_\sigma\nabla^\sigma W^\eta - 2H\nabla^\eta\nabla_\rho W^\rho - \\
&-V\nabla_\alpha W_\nu W_\rho \varepsilon^{\alpha\nu\rho\eta} - UW^\alpha W_\alpha W^\eta - \\
&-2LR^{\eta\rho}W_\rho + 2NRW^\eta + PW^\eta = \kappa S^\eta
\end{aligned}
$$

where $S^\alpha$ will have to be fixed on general grounds.

This general field equation can be restricted with the Velo–Zwanziger method [22,23]. Thus, taking its divergence

$$2(\Pi-H)\nabla_\eta\nabla^\eta\nabla_\rho W^\rho +$$
$$+V\nabla_\eta\nabla_\alpha W_\nu W_\rho\varepsilon^{\eta\alpha\nu\rho} +$$
$$+V\nabla_\alpha W_\nu\nabla_\eta W_\rho\varepsilon^{\eta\alpha\nu\rho} -$$
$$-2[UW^\rho W^\eta+(L-\Pi)R^{\eta\rho}]\nabla_\eta W_\rho +$$
$$+(2N-L+\Pi)\nabla_\eta RW^\eta -$$
$$-(UW^\alpha W_\alpha-2NR-P)\nabla_\eta W^\eta=\kappa\nabla_\eta S^\eta,$$

it becomes possible to see that there appears a third-order time derivative for the temporal component of the torsion axial-vector implying that the constraint obtained from the field equations would actually determine the time evolution of some components of the torsion axial-vector field. Since this would spoil a balance between the number of independent field equations and the amount of degrees of freedom of a given field, then no higher-order derivative terms must be produced in the constraints and thus we set $\Pi=H$ identically. Once this is done, there is no second-order derivative in time for any components of the field in the constraint, which is thus a true constraint, which substituted back into the field equations gives

$$2H\nabla_\sigma\nabla^\sigma W^\eta-2H(UW^\alpha W_\alpha-2NR-P)^{-1}\cdot$$
$$\cdot\nabla^\eta[V\nabla_\tau\nabla_\alpha W_\nu W_\rho\varepsilon^{\tau\alpha\nu\rho}+V\nabla_\alpha W_\nu\nabla_\tau W_\rho\varepsilon^{\tau\alpha\nu\rho} -$$
$$-2[UW^\rho W^\tau+(L-H)R^{\tau\rho}]\nabla_\tau W_\rho +$$
$$+(2N-L+H)\nabla_\tau RW^\tau-\kappa\nabla_\tau S^\tau] +$$
$$+2H\nabla^\eta(UW^\alpha W_\alpha-2NR)(UW^\alpha W_\alpha-2NR-P)^{-2}\cdot$$
$$\cdot[V\nabla_\tau\nabla_\alpha W_\nu W_\rho\varepsilon^{\tau\alpha\nu\rho}+V\nabla_\alpha W_\nu\nabla_\tau W_\rho\varepsilon^{\tau\alpha\nu\rho} -$$
$$-2[UW^\rho W^\tau+(L-H)R^{\tau\rho}]\nabla_\tau W_\rho +$$
$$+(2N-L+H)\nabla_\tau RW^\tau-\kappa\nabla_\tau S^\tau] -$$
$$-V\nabla_\alpha W_\nu W_\rho\varepsilon^{\alpha\nu\rho\eta}-UW^\alpha W_\alpha W^\eta -$$
$$-2LR^{\eta\rho}W_\rho+2NRW^\eta+PW^\eta=\kappa S^\eta$$

which contains second-order time derivatives of all components of the torsion axial-vector, and therefore this is a true field equation. To check the propagation properties of the field, we consider its characteristic determinant

$$(UW^\alpha W_\alpha-2NR-P)n^2+2[UW^\tau W^\nu+(L-H)R^{\tau\nu}]n_\tau n_\nu=0$$

and, by following the general discussion of Velo and Zwanziger, one can see that, in general, acausality may be possible unless we have $L=H$ and $N=U=0$ identically, in which case $n^2=0$ and thus causality is ensured. Notice that there are no constraints on $V$, which remains a free parameter.

Placing all constraints together gives field equations

$$4\nabla_\rho(\partial W)^{\rho\eta}-VW_\rho(\partial W)_{\alpha\nu}\varepsilon^{\rho\alpha\nu\eta}+2PW^\eta=2\kappa S^\eta$$

because $H$ can be reabsorbed within a redefinition of all the other constants.

To proceed, we notice that, for the metric field equations, the source contribution from the torsion axial-vector field has to be built with no quartic torsion term because they would correspond to what in the torsion field equations are cubic torsion terms, which are absent, and no second derivatives of torsion because they would give rise to curvatures,

which cannot be present since they are already addressed. Thus, it is possible to come to the most general form of this contribution as the one given by

$$E^{\mu\nu} = aW^{\mu}W^{\nu} + bW^2 g^{\mu\nu} + z(W^{\nu}W_{\rho}(\partial W)_{\alpha\sigma}\varepsilon^{\rho\alpha\sigma\mu} +$$
$$+ W^{\mu}W_{\rho}(\partial W)_{\alpha\sigma}\varepsilon^{\rho\alpha\sigma\nu}) + y(\nabla_{\sigma}W^{\mu}(\partial W)^{\sigma\nu} + \nabla_{\sigma}W^{\nu}(\partial W)^{\sigma\mu}) + x\nabla^{\mu}W_{\sigma}\nabla^{\nu}W^{\sigma} +$$
$$+ w\nabla_{\sigma}W^{\mu}\nabla^{\sigma}W^{\nu} + v\nabla_{\alpha}W_{\sigma}\nabla^{\alpha}W^{\sigma}g^{\mu\nu} + u(\partial W)^{\nu\sigma}(\partial W)^{\mu}{}_{\sigma} + t(\partial W)^2 g^{\mu\nu}$$

in terms of ten constants: because we know that $\nabla_{\nu}E^{\nu\mu} = 0$ and because in vacuum the divergence of the torsion field equations gives

$$4P\nabla \cdot W + V(\partial W)_{\eta\rho}(\partial W)_{\alpha\nu}\varepsilon^{\eta\rho\alpha\nu} = 0,$$

then one can easily see that it must be $V = z = v = 0$ with $x = y = -w$ and $x + u = -4t$ and together with $a = -2b = 2tP$ which must hold identically.

We also notice that we must have $P = 2M^2$ because this is just the mass term of the torsion axial-vector field as it is well known.

The field equations for the gauge field are also in the form of covariant derivatives of the gauge potential equal to some source: nevertheless, taking derivatives of the gauge potential means that that we have to consider the gauge strength because this is the only term that is differential in the potential and which is still gauge invariant, but, since this is irreducible, any contraction of the gauge strength vanishes and therefore these terms alone cannot be not enough. Hence, we have to take one more covariant derivative of the gauge strength as a leading term.

The most general field equations for the gauge fields have a leading term in the form $\nabla_{\sigma}F_{\alpha\rho}$ and, after contraction, we get $\nabla_{\sigma}F^{\sigma\rho}$ as the leading term: then, we get

$$\nabla_{\sigma}F^{\sigma\eta} - \tfrac{1}{12}BF_{\alpha\nu}W_{\rho}\varepsilon^{\alpha\nu\rho\eta} = qJ^{\eta}$$

in which the source $J^{\alpha}$ will have to be fixed as well.

The contribution from the gauge field is similarly built in terms of squares of the gauge strength strength, since any other term would violate gauge symmetry. Thus,

$$E^{\mu\nu} = \alpha F^{\mu\rho}F^{\nu}{}_{\rho} + \beta F^{\alpha\pi}F_{\alpha\pi}g^{\mu\nu}$$

in terms of two constants: again, because $\nabla_{\nu}E^{\nu\mu} = 0$ and using the form of the electrodynamic field equations, we can see that $B = 0$ and $\alpha = -4\beta$ identically.

In the metric field equation, the contributions due to torsion and gauge fields are analogous, and torsion and gauge fields are independent, so we may normalize torsion and gauge fields with no loss of generality in order to have the two constants $t$ and $\beta$ with the same value, and it is still without losing generality that they can be reabsorbed in the $k$ constant. We notice that, in reabsorbing within a renaming of the constant $k$ the values of the constants $t$ and $\beta$, we did not lose any generality in their absolute value, but, in order not to lose any generality also for the sign, all constants would have to be positive, and this in general may not be the case: the reason why we did it anyway is that those constants are in front of torsion and gauge fields' energy contributions, which are positive defined. Of course, we might have assumed those constants to possess a generic sign, but, in the final form of the field equations, we would have discovered that those signs were positive, and thus we can assume this immediately with no loss of generality.

To proceed with the inclusion of matter fields, it is fundamental to notice that spinor fields are defined in terms of gamma matrices that can also be used in building fundamental quantities, whose employment allows for lowering the order of derivatives in all such quantities because every time covariance demands for a single covariant index to be present, and one gamma matrix can be used instead of one spinorial derivative.

The most general field equations for the spinor field have a leading term containing $\nabla_\mu\psi$ so that, after multiplying by the matrix $\gamma^\nu$, it is possible to contract the indices getting $\gamma^\mu\nabla_\mu\psi$ as a leading term: therefore, we may establish the most general form of field equations as

$$i\gamma^\mu\nabla_\mu\psi - XW_\sigma\gamma^\sigma\boldsymbol{\pi}\psi - m\psi = 0$$

where the imaginary unit has been placed because, in free cases, $i\gamma^\mu\nabla_\mu\psi - m\psi = 0$ so that taking the square of the derivative gives $\nabla^2\psi + m^2\psi = 0$, and $m$ can be interpreted as the mass term, which is what is expected. That is, the imaginary unit has to be interpreted as what ensures the mass of the field will behave as to provide non-imaginary contribution to the dynamics of the free field equations.

Then, we have to write the general form of their contribution in the metric field equations, and this can be constructed by employing no more than one spinorial derivative of the spinor field, since gamma matrices can be used to saturate indices: eventually,

$$\begin{aligned}
E^{\rho\sigma} = {}&\zeta[\nabla^\rho(\overline{\psi}\gamma^\sigma\psi) + \nabla^\sigma(\overline{\psi}\gamma^\rho\psi)] + \\
&+ i\xi(\overline{\psi}\gamma^\rho\nabla^\sigma\psi - \nabla^\sigma\overline{\psi}\gamma^\rho\psi + \overline{\psi}\gamma^\sigma\nabla^\rho\psi - \nabla^\rho\overline{\psi}\gamma^\sigma\psi) + \\
&+ \chi\nabla_\alpha(\overline{\psi}\gamma^\alpha\psi)g^{\rho\sigma} + \lambda i(\overline{\psi}\gamma^\alpha\nabla_\alpha\psi - \nabla_\alpha\overline{\psi}\gamma^\alpha\psi)g^{\rho\sigma} + \\
&+ \tau(W^\sigma\overline{\psi}\gamma^\rho\boldsymbol{\pi}\psi + W^\rho\overline{\psi}\gamma^\sigma\boldsymbol{\pi}\psi) + vW_\alpha\overline{\psi}\gamma^\alpha\boldsymbol{\pi}\psi g^{\sigma\rho} + \mu\overline{\psi}\psi g^{\rho\sigma}
\end{aligned}$$

in general; the contribution as a source of the torsion field equations is the spin density of the material field, and it can be taken without any spinorial derivative at all when gamma matrices are considered, therefore obtaining that

$$S^\mu = \omega\overline{\psi}\gamma^\mu\boldsymbol{\pi}\psi$$

also in general; the contribution as source of the gauge field equations is the current density of the material field, and similarly it is given according to

$$J^\rho = p\overline{\psi}\gamma^\rho\psi$$

again in the most general case: by considering again the divergences of all field equations and with the same reasoning as before, one can eventually see that $\zeta = 0$ as well as $\mu = -2\lambda m$ and $p = 4\xi$ with $\tau = -2\xi X$ and $v = -2\lambda X$ and also $\kappa\omega = 2\xi X$ identically.

Finally, we notice that, without affecting the metric or the torsion or the gauge fields, the spinor field may be renormalized in such a way that, without losing generality, we can always set $4\xi = 1$ and, as a consequence, it is possible to see that the full system of field equations has been completely determined.

It is constituted by the metric field equations given according to the expression

$$\begin{aligned}
R^{\rho\sigma} - \tfrac{1}{2}Rg^{\rho\sigma} &- \tfrac{k}{2}[\tfrac{1}{4}(\partial W)^2 g^{\rho\sigma} - (\partial W)^{\sigma\alpha}(\partial W)^\rho{}_\alpha] - \\
&- \tfrac{k}{2}(\tfrac{1}{4}F^2 g^{\rho\sigma} - F^{\rho\alpha}F^\sigma{}_\alpha) - \tfrac{k}{2}M^2(W^\rho W^\sigma - \tfrac{1}{2}W^2 g^{\rho\sigma}) - \\
- \Lambda g^{\rho\sigma} &= \tfrac{1}{2}k[\tfrac{i}{4}(\overline{\psi}\gamma^\rho\nabla^\sigma\psi - \nabla^\sigma\overline{\psi}\gamma^\rho\psi + \overline{\psi}\gamma^\sigma\nabla^\rho\psi - \nabla^\rho\overline{\psi}\gamma^\sigma\psi) - \\
&- \tfrac{1}{2}X(W^\sigma\overline{\psi}\gamma^\rho\boldsymbol{\pi}\psi + W^\rho\overline{\psi}\gamma^\sigma\boldsymbol{\pi}\psi)]
\end{aligned}$$

and the torsion field equations given according to

$$\nabla_\rho(\partial W)^{\rho\mu} + M^2 W^\mu = X\overline{\psi}\gamma^\mu\boldsymbol{\pi}\psi$$

with gauge field equations given by

$$\nabla_\sigma F^{\sigma\mu} = q\overline{\psi}\gamma^\mu\psi$$

as the form that is usually known, while the matter field equations are

$$i\gamma^\mu\nabla_\mu\psi - XW_\sigma\gamma^\sigma\boldsymbol{\pi}\psi - m\psi = 0$$

with parameters $\Lambda$ and $M$ and also $m$ describing intrinsic properties of metric and torsion and also spinor fields, while parameters $k$, $X$, and $q$ are the constants that measure the strength with which metric, torsion, and gauge fields couple to energy, spin, and current.

It is possible to write the above system of coupled field equations into the system of coupled field equations with respect to which all the torsionless derivatives and curvatures are the torsionful derivatives and curvatures.

Thus, we can give the full system of field equations as the torsion–spin and the curvature–energy field equations as

$$D_{[\rho}D^\sigma Q_{\mu\nu]\sigma} + Q_{\eta[\mu\nu}G_{\rho]\sigma}g^{\sigma\eta} - G_{\sigma[\rho}Q_{\mu\nu]\eta}g^{\sigma\eta} + M^2 Q_{\rho\mu\nu} = \tfrac{1}{12}S_{\rho\mu\nu} \tag{121}$$

and

$$\begin{aligned}
&G^{\rho\sigma} - \tfrac{1}{2}Gg^{\rho\sigma} - 18k\big[\tfrac{1}{3}D_\alpha D^{[\alpha}D_\pi Q^{\rho\sigma]\pi} - \tfrac{1}{3}D_\alpha D_\eta Q^{\eta\pi[\alpha}Q^{\rho\sigma]\nu}g_{\nu\pi} - \\
&\quad -\tfrac{1}{3}Q^{\rho\eta\varphi}D^{[\sigma}D_\pi Q^{\eta\varphi]\pi} - \tfrac{1}{3}Q^{\sigma\eta\varphi}D^{[\rho}D_\pi Q^{\eta\varphi]\pi} + \tfrac{1}{2}D^\pi D_\tau Q^{\tau\mu\nu}Q_{\pi\mu\nu}g^{\rho\sigma} + \\
&\quad +\tfrac{1}{4}D_\pi Q^{\tau\mu\nu}D^\tau Q_{\tau\mu\nu}g^{\rho\sigma} - D_\pi Q^{\tau\mu\rho}D^\tau Q_{\tau\mu}{}^\sigma - \tfrac{1}{3}D_\eta Q^{\eta\pi\alpha}D_\alpha Q^{\rho\sigma}{}_\pi + \\
&\quad +\tfrac{1}{3}(Q^{\rho\eta\varphi}D_\tau Q^{\tau\pi\sigma} + Q^{\sigma\eta\varphi}D_\tau Q^{\tau\pi\rho})Q^{\eta\varphi}{}_\pi\big] - \tfrac{1}{2}k(\tfrac{1}{4}F^2 g^{\rho\sigma} - F^{\rho\alpha}F^\sigma{}_\alpha) - \\
&\quad -(12kM^2 + 1)(\tfrac{1}{2}D_\alpha Q^{\alpha\rho\sigma} - \tfrac{1}{4}Q^{\rho\alpha\pi}Q^\sigma{}_{\alpha\pi} + \tfrac{1}{8}Q^2 g^{\rho\sigma}) - \Lambda g^{\rho\sigma} = \tfrac{1}{2}kT^{\rho\sigma}
\end{aligned} \tag{122}$$

called *Sciama–Kibble field equations* and *Einstein field equations* and they come alongside the gauge-current field equations

$$D_\sigma F^{\sigma\mu} + \tfrac{1}{2}F_{\alpha\nu}Q^{\alpha\nu\mu} = J^\mu \tag{123}$$

called *Maxwell field equations*, where the sources are given by the spin and the energy

$$S^{\rho\mu\nu} = -8X\tfrac{i}{4}\overline{\psi}\{\gamma^\rho, \sigma^{\mu\nu}\}\psi \tag{124}$$

and

$$\begin{aligned}
T^{\rho\sigma} &= \tfrac{i}{2}(\overline{\psi}\gamma^\rho\boldsymbol{D}^\sigma\psi - \boldsymbol{D}^\sigma\overline{\psi}\gamma^\rho\psi) + (8X+1)D_\alpha(\tfrac{i}{4}\overline{\psi}\{\gamma^\alpha, \sigma^{\rho\sigma}\}\psi) + \\
&\quad +\tfrac{1}{2}(8X+1)Q^{\rho\mu\nu}\tfrac{i}{4}\overline{\psi}\{\gamma^\sigma, \sigma_{\mu\nu}\}\psi - (8X+1)Q^{\sigma\mu\nu}\tfrac{i}{4}\overline{\psi}\{\gamma^\rho, \sigma_{\mu\nu}\}\psi
\end{aligned} \tag{125}$$

alongside the current

$$J^\mu = q\overline{\psi}\gamma^\mu\psi \tag{126}$$

given in terms of the matter field. They come alongside the spinorial field equation

$$i\gamma^\mu\boldsymbol{D}_\mu\psi - i(X + \tfrac{1}{8})Q_{\nu\tau\alpha}\gamma^\nu\gamma^\tau\gamma^\alpha\psi - m\psi = 0 \tag{127}$$

called *Dirac spinorial field equations*, which decompose according to

$$\tfrac{i}{2}(\overline{\psi}\gamma^\mu\boldsymbol{D}_\mu\psi - \boldsymbol{D}_\mu\overline{\psi}\gamma^\mu\psi) - (X + \tfrac{1}{8})Q^{\pi\tau\eta}S^\sigma\varepsilon_{\pi\tau\eta\sigma} - m\Phi = 0 \tag{128}$$

$$D_\mu U^\mu = 0 \tag{129}$$

$$\tfrac{i}{2}(\overline{\psi}\gamma^\mu\boldsymbol{\pi}\boldsymbol{D}_\mu\psi - \boldsymbol{D}_\mu\overline{\psi}\gamma^\mu\boldsymbol{\pi}\psi) - (X + \tfrac{1}{8})Q^{\pi\tau\eta}U^\sigma\varepsilon_{\pi\tau\eta\sigma} = 0 \tag{130}$$

$$D_\mu S^\mu - 2m\Theta = 0 \tag{131}$$

$$i(\overline{\psi}\boldsymbol{D}^\alpha\psi - \boldsymbol{D}^\alpha\overline{\psi}\psi) - D_\mu M^{\mu\alpha} + (2X + \tfrac{1}{4})\varepsilon_{\pi\tau\eta\sigma}Q^{\pi\tau\eta}\Sigma^{\sigma\alpha} - 2mU^\alpha = 0 \tag{132}$$

$$D_\alpha\Phi - 2(\overline{\psi}\sigma_{\mu\alpha}\boldsymbol{D}^\mu\psi - \boldsymbol{D}^\mu\overline{\psi}\sigma_{\mu\alpha}\psi) + (2X + \tfrac{1}{4})\Theta Q^{\pi\tau\eta}\varepsilon_{\pi\tau\eta\alpha} = 0 \tag{133}$$

$$D_\nu\Theta - 2i(\overline{\psi}\sigma_{\mu\nu}\boldsymbol{\pi}\boldsymbol{D}^\mu\psi - \boldsymbol{D}^\mu\overline{\psi}\sigma_{\mu\nu}\boldsymbol{\pi}\psi) - (2X + \tfrac{1}{4})\Phi Q^{\pi\tau\eta}\varepsilon_{\pi\tau\eta\nu} + 2mS_\nu = 0 \tag{134}$$

$$(\boldsymbol{D}_\alpha\overline{\psi}\boldsymbol{\pi}\psi - \overline{\psi}\boldsymbol{\pi}\boldsymbol{D}_\alpha\psi) + D^\mu\Sigma_{\mu\alpha} + (2X + \tfrac{1}{4})\varepsilon^{\pi\tau\eta\mu}Q_{\pi\tau\eta}M_{\mu\alpha} = 0 \tag{135}$$

$$D^\mu S^\rho\varepsilon_{\mu\rho\alpha\nu} + i(\overline{\psi}\gamma_{[\alpha}\boldsymbol{D}_{\nu]}\psi - \boldsymbol{D}_{[\nu}\overline{\psi}\gamma_{\alpha]}\psi) + (2X + \tfrac{1}{4})Q^{\pi\tau\eta}\varepsilon_{\pi\tau\eta[\alpha}S_{\nu]} = 0 \tag{136}$$

$$D^{[\alpha}U^{\nu]} - i\varepsilon^{\alpha\nu\mu\rho}(\boldsymbol{D}_\mu\overline{\psi}\gamma_\rho\boldsymbol{\pi}\psi - \overline{\psi}\gamma_\rho\boldsymbol{\pi}\boldsymbol{D}_\mu\psi) - (12X + \tfrac{3}{2})Q^{\alpha\nu\rho}U_\rho - 2mM^{\alpha\nu} = 0 \tag{137}$$

which are altogether equivalent to the Dirac spinor field equations and called *Gordon decompositions*. Spin, energy, and current verify

$$D_\rho S^{\rho\mu\nu} + \tfrac{1}{2}T^{[\mu\nu]} \equiv 0 \tag{138}$$

and

$$D_\mu T^{\mu\nu} + T_{\rho\beta}Q^{\rho\beta\nu} - S_{\mu\rho\beta}G^{\mu\rho\beta\nu} + J_\rho F^{\rho\nu} \equiv 0 \tag{139}$$

alongside

$$D_\rho J^\rho = 0 \tag{140}$$

satisfied in the most general case.

Intriguingly, we notice that, in the spinor field equations, the mass appears linearly, and thus it may be positive as well as negative, and therefore it is possible to have two different types of spinor field equations. Such possibility is clear because, if $m \to -m$ is accompanied by the discrete transformation $\psi \to \boldsymbol{\pi}\psi$, then the system of field equations is invariant, and, consequently, any solutions of the first are also a solution of the second. The fact that we may have two different spinor field equations is translated into the fact that we may have two different solutions linked by $\psi \to \boldsymbol{\pi}\psi$ in general.

The full system of field equations is invariant under the transformation of parity reflection [24], and it is the most general under the restriction of being at the least-order differential form [25]. What this means is that arguments of compatibility with covariance, generality, and having field equations at their least-order derivative are enough to lead to the above physical field equations. In addition, this is true regardless of the principle of equivalence. The principle of equivalence might have been a guide for Einstein from a historical perspective, but mathematically there is no need for it. Its role is reduced to that of an interpretative principle telling us that the metric is what encodes the information about gravitation. Moreover, it is common knowledge of Einsteinian gravity that, when Einstein field equations are linearized and taken in the static case and for small velocities, they reduce to Newton equations, in which the time–time component of the metric is witnessed to be the Newtonian gravitational potential. Henceforth, the principle of equivalence can be abandoned. Certainly, this principle may give important insights, but it can be equally well disregarded, as the interpretation of gravity within the metric tensor naturally emerges from specific limits of the Einstein field equations and these come from arguments of simplicity, generality, and compatibility with identities proper to the underlying geometric structure.

### 3. Torsion-Spin Interactions

In this second section, we will consider the above physical field equations in order to investigate their properties: the idea will be to write them in an equivalent but somewhat clearer manner. We will end with general remarks about the interaction of geometry with its material content.

*3.1. Torsion and Spinor Decomposition*

To have the physical field equations converted in more manageable forms, we will decompose all quantities that can be decomposed into more fundamental ones: we will separate torsion from all torsionless quantities in all the covariant derivatives and curvature tensors. Finally, the spinor field will also be decomposed into its two irreducible chiral projections and elementary degrees of freedom.

3.1.1. Torsion as Axial-Vector Massive Field

Among all geometric fields, torsion has a special property indeed. The gauge potential is a gauge field for phase transformations, and the metric tensor can be considered a gauge field for coordinate transformations, so both are always depending on the phase or the coordinate system, while torsion is a tensor that does not have any relation with such properties. Thus, torsion can be split from gauge and metric connections, with all the covariant derivatives and curvatures being written as covariant derivatives and curvatures with no torsion but with all the torsion terms appearing as independent.

To have the most general connection decomposed into the simplest symmetric connection plus torsion terms, we only need to substitute (16) in (37), and this in (114).

Thus, the system of field equations reduces to

$$\nabla_\rho (\partial W)^{\rho\mu} + M^2 W^\mu = X\overline{\psi}\gamma^\mu \boldsymbol{\pi}\psi \tag{141}$$

and

$$
\begin{aligned}
R^{\rho\sigma} - \tfrac{1}{2}Rg^{\rho\sigma} - \Lambda g^{\rho\sigma} = \tfrac{k}{2}\big[ &\tfrac{1}{4}F^2 g^{\rho\sigma} - F^{\rho\alpha}F^\sigma{}_\alpha + \\
&+ \tfrac{1}{4}(\partial W)^2 g^{\rho\sigma} - (\partial W)^{\sigma\alpha}(\partial W)^\rho{}_\alpha + M^2(W^\rho W^\sigma - \tfrac{1}{2}W^2 g^{\rho\sigma}) + \\
&+ \tfrac{i}{4}(\overline{\psi}\gamma^\rho\nabla^\sigma\psi - \nabla^\sigma\overline{\psi}\gamma^\rho\psi + \overline{\psi}\gamma^\sigma\nabla^\rho\psi - \nabla^\rho\overline{\psi}\gamma^\sigma\psi) - \\
&- \tfrac{1}{2}X(W^\sigma\overline{\psi}\gamma^\rho\boldsymbol{\pi}\psi + W^\rho\overline{\psi}\gamma^\sigma\boldsymbol{\pi}\psi)\big]
\end{aligned} \tag{142}
$$

for the torsion–spin and curvature–energy coupling, and

$$\nabla_\sigma F^{\sigma\mu} = q\overline{\psi}\gamma^\mu\psi \tag{143}$$

for the gauge–current coupling. Then, we also have that

$$i\gamma^\mu\nabla_\mu\psi - XW_\sigma\gamma^\sigma\boldsymbol{\pi}\psi - m\psi = 0 \tag{144}$$

for the spinor field equations.

If we take the divergence of (141) and contract (142), we obtain the constraints

$$M^2\nabla_\mu W^\mu = 2Xmi\overline{\psi}\boldsymbol{\pi}\psi \tag{145}$$

and

$$\tfrac{2}{k}R + \tfrac{8}{k}\Lambda - M^2 W^2 = -m\overline{\psi}\psi \tag{146}$$

where (144) has been used.

It is now possible to interpret torsion: just a quick look at the torsion–spin and curvature–energy field equations simply reveals that *torsion is an axial-vector massive field* verifying Proca field equations with corresponding energy and torsional-spin coupling

within the gravitational field equations [26]. With this insight, one might now wonder if there really was the necessity to go through the trouble of insisting on the presence of torsion if all comes to the presence of an axial-vector massive field, asking why we could not simply impose torsion equal to zero and then allowing an axial-vector massive field to be included into the theory. The answer is that, although mathematically it is equivalent to follow both approaches, conceptually the former approach is the most straightforward construction in which all quantities are defined and all relationships are built in the most general manner, while, on the other hand, the latter approach would be afflicted by a number of arbitrary assumptions. If this latter approach were the one to be followed, we would have to justify why torsion albeit in general present should be removed, why, among all fields that could be included, we pick precisely a vector field with pseudo-tensorial properties, and why it would have to be massive, and hence resulting into an approach having three unjustified assumptions in alternative to the other approach in which assumptions are either justified or not assumed at all. In order to avoid this high degree of arbitrariness, we prefer to follow the approach that we actually followed here. This leads after all to the presence of an axial-vector massive field. Then, if in some part of the theory, there were to appear new physics that could somehow be reconducted to an axial-vector massive field, we would know that these effects would emerge from the existence of torsion. In fact, such effects might be something that we have already observed, even if we ignored that they could come from the torsion tensor.

### Spinors as Sum of Chiral Parts

Analogously to the covariant decomposition of torsion, there is also a perfectly covariant split of the spinor field into its two chiral parts according to (80) and (81) and therefore in its degrees of freedom as expressed by (105).

When (105) is plugged into the Gordon decompositions, we obtain the polar forms of the Gordon decompositions, among which we find the following two equations:

$$-XW_\mu - \tfrac{1}{4}g_{\mu\nu}\varepsilon^{\nu\rho\sigma\alpha}\partial_\rho\xi^k_\sigma\xi^j_\alpha\eta_{jk} - (\nabla\alpha - qA)^\iota u_{[\iota}s_{\mu]} + s_\mu m\cos\beta + \tfrac{1}{2}\nabla_\mu\beta = 0$$

and

$$s_\mu m\sin\beta - (\nabla\alpha - qA)^\rho u^\nu s^\alpha\varepsilon_{\mu\rho\nu\alpha} + \tfrac{1}{2}|\xi|^{-1}\xi^k_\mu\partial_\alpha(|\xi|\xi^\alpha_k) + \nabla_\mu\ln\phi = 0$$

which are very special since we can show that these two expressions imply the spinor field Equations (144): in fact, by employing the above pair of equations, we have that

$$i\gamma^\mu\nabla_\mu\psi - XW_\sigma\gamma^\sigma\boldsymbol{\pi}\psi - m\psi =$$
$$= (\nabla\alpha - qA)^\iota(i\gamma^\mu u^\nu s^\alpha\varepsilon_{\mu\iota\nu\alpha} + u_{[\iota}s_{\mu]}\gamma^\mu\boldsymbol{\pi} + \gamma_\iota)\psi -$$
$$- m(is_\mu\gamma^\mu\sin\beta + s_\mu\gamma^\mu\boldsymbol{\pi}\cos\beta + \mathbb{I})\psi$$

and then, using (90, 91), we get

$$i\gamma^\mu\nabla_\mu\psi - XW_\sigma\gamma^\sigma\boldsymbol{\pi}\psi - m\psi = 0$$

which is the Dirac spinor field Equation (144) as expected.

Therefore, we may summarize by saying that the Dirac spinorial field equation are equivalent to the equations

$$\nabla_\mu\beta - 2XW_\mu - \tfrac{1}{2}g_{\mu\nu}\varepsilon^{\nu\rho\sigma\alpha}\partial_\rho\xi^k_\sigma\xi^j_\alpha\eta_{jk} - 2(\nabla\alpha - qA)^\iota u_{[\iota}s_{\mu]} + 2ms_\mu\cos\beta = 0 \tag{147}$$

and

$$\nabla_\mu\ln\phi^2 + |\xi|^{-1}\xi^k_\mu\partial_\alpha(|\xi|\xi^\alpha_k) - 2(\nabla\alpha - qA)^\rho u^\nu s^\alpha\varepsilon_{\mu\rho\nu\alpha} + 2ms_\mu\sin\beta = 0 \tag{148}$$

in the most general case that is possible.

Thus, we interpret spinor fields in this way: despite being fundamental, *spinor fields are reducible, constituted from two chiral parts. Their independence is measured by the Yvon–Takabayashi angle describing internal dynamics of the spinor field. The module describes the overall matter distribution.* These are the two degrees of freedom of the spinor field, with all space–time derivatives of these two degrees of freedom specified by (147) and (148) [27].

### 3.2. Torsion–Spinor Interactions

We now have all elements to deepen the investigation about the interaction between geometry and matter.

### 3.2.1. Torsion–Spinor Binding

In the recent parts, we have seen that (145) and (146) provide very simply links between geometrical structures and the bi-linear spinorial scalars. In addition, (147) and (148) constitute some form of dynamical conditions upon such spinorial scalars.

We have in fact that (145) and (146) can be written as

$$M^2 \nabla_\mu W^\mu = 4Xm\phi^2 \sin\beta \tag{149}$$

and

$$\tfrac{2}{k}R + \tfrac{8}{k}\Lambda - M^2 W^2 = -2m\phi^2 \cos\beta \tag{150}$$

linking torsion and curvature to Yvon–Takabayashi angle and module, these last being subject to

$$\nabla_\mu \beta - 2XW_\mu - \tfrac{1}{2}g_{\mu\nu}\varepsilon^{\nu\rho\sigma\alpha}\partial_\rho\xi^k_\sigma\xi^j_\alpha\eta_{jk} - 2(\nabla\alpha - qA)^\iota u_{[\iota}s_{\mu]} + 2ms_\mu \cos\beta = 0 \tag{151}$$

and

$$\nabla_\mu \ln\phi^2 + |\xi|^{-1}\xi^k_\mu\partial_\alpha(|\xi|\xi^\alpha_k) - 2(\nabla\alpha - qA)^\rho u^\nu s^\alpha \varepsilon_{\mu\rho\nu\alpha} + 2ms_\mu \sin\beta = 0 \tag{152}$$

as dynamical conditions. Therefore, by solving these last equations, *we can always integrate spinor fields as all their degrees of freedom can be tied to torsion and curvature.*

This is remarkable because it shows that formally the spinorial degrees of freedom can always be replaced by quantities related to the underlying geometric structure.

To conclude this part, we will give a few more results starting from the introduction of the potentials

$$K_\mu = 2XW_\mu + \tfrac{1}{2}g_{\mu\nu}\varepsilon^{\nu\rho\sigma\alpha}\partial_\rho\xi^k_\sigma\xi^j_\alpha\eta_{jk} + 2(\nabla\alpha - qA)^\iota u_{[\iota}s_{\mu]} \tag{153}$$

and

$$G_\mu = -|\xi|^{-1}\xi^k_\mu\partial_\alpha(|\xi|\xi^\alpha_k) + 2(\nabla\alpha - qA)^\rho u^\nu s^\alpha \varepsilon_{\mu\rho\nu\alpha}, \tag{154}$$

in terms of which we have

$$\nabla_\mu \beta - K_\mu + s_\mu 2m\cos\beta = 0 \tag{155}$$

and

$$\nabla_\mu \ln\phi^2 - G_\mu + s_\mu 2m\sin\beta = 0 \tag{156}$$

as the Dirac equations in polar form. From these, we get

$$\left|\nabla\tfrac{\beta}{2}\right|^2 - m^2 - \phi^{-1}\nabla^2\phi + \tfrac{1}{2}(\nabla G + \tfrac{1}{2}G^2 - \tfrac{1}{2}K^2) = 0 \tag{157}$$

and

$$\nabla_\mu(\phi^2\nabla^\mu \tfrac{\beta}{2})-\tfrac{1}{2}(\nabla K+KG)\phi^2=0 \tag{158}$$

as a Hamilton–Jacobi equation and a continuity equation.

Alternatively, we may define

$$\tfrac{1}{2}(\nabla_\mu\beta-2XW_\mu-\tfrac{1}{2}g_{\mu\nu}\varepsilon^{\nu\rho\sigma\alpha}\partial_\rho\xi^k_\sigma\xi^j_\alpha\eta_{jk})=Y_\mu \tag{159}$$

and

$$-\tfrac{1}{2}[\nabla_\mu\ln\phi^2+|\xi|^{-1}\xi^k_\mu\partial_\alpha(|\xi|\xi^\alpha_k)]=Z_\mu \tag{160}$$

in terms of which

$$Y_\mu-(\nabla\alpha-qA)^\iota u_{[\iota}s_{\mu]}+ms_\mu\cos\beta=0 \tag{161}$$

and

$$Z_\mu-(\nabla\alpha-qA)^\rho u^\nu s^\alpha\varepsilon_{\mu\rho\nu\alpha}+ms_\mu\sin\beta=0 \tag{162}$$

as Dirac equations in polar form. Then, defining

$$P_\nu=\nabla_\nu\alpha-qA_\nu \tag{163}$$

as the momentum, we have that

$$P^\nu=m\cos\beta u^\nu+Y_\mu u^{[\mu}s^{\nu]}+Z_\mu s_\rho u_\sigma\varepsilon^{\mu\rho\sigma\nu} \tag{164}$$

giving its explicit form in terms of mass and velocity but also in terms of the Yvon–Takabayashi angle and spin as well as the potentials given in the (159) and (160) above.

There is a very important point to be clarified regarding the spinorial active transformations acting on spinorial fields. Consider the rotations around the third axis and the spinors in polar form (105): despite the fact that these spinors are aligned along the axis around which we perform the above rotation, that rotation does not leave them unchanged (as we have for vectors). This might already sound problematic, but, in addition, we also have that, when such a rotation is given for an angle $\theta=2\pi$, it is $\Lambda=-\mathbb{I}$ implying that the spinor would not go back to the initial configuration (as we have when we perform a passive rotation). This too sounds peculiar. Thus, we might ask, is there any intuitive way to see things under which these odd behaviors would look natural? First of all, we have to take into account the fact that the rotation is an active rotation, and therefore an operation that, keeping fixed the space–time, moves the spinor. Then, we have to keep in mind that spinors are more sensitive than vectors to the structure of the space–time, as if anchored instead of being free to slide in it. Thus, for a given active rotation around a certain axis, a vector behaves like a pole, and, if aligned to the axis of rotation, it would be left unchanged as a whole. For the same active rotation, however, spinors would behave as a pole with a flag, so, even if aligned to the axis of rotation, they would be left unchanged almost fully but not quite entirely. They would indeed behave as if the rotation was taking place on a Möbius band. One way to picture them would be that of the belt trick, or the spinning plates, as described in [28].

We have shown that there is a duplicity in the spinorial structure made clear from the fact that spinors were defined up to the $\psi\to\pi\psi$ discrete transformation, or from the fact that the combined $\psi\to\pi\psi$ and $m\to-m$ is one symmetry of the physical field equations. Such duplicity may suggest a form of matter/antimatter duality [29,30].

Physical effects and phenomenological implications provided by a torsion tensor with a dynamical axial-vector field have also been recently presented in [31].

## 4. Limiting Situations

In this third section, we will consider the above theory in some specific cases so to deepen their examination: we will first consider what happens as a consequence of the fact that torsion is an axial-vector field with mass and in addition we will discuss what happens as a consequence of the fact that also the spinor field is massive. Eventually, we will see what happens in the complementary situation in which masslessness will allow another symmetry.

### 4.1. Massive Cases

We start from the analysis of the consequences of massive torsion: assuming also that the torsion mass is quite large, we will study the effective approximation. Finally, some comments about low-speed conditions will be given from the perspective of the non-relativistic limit.

### 4.1.1. Effective Approximation

To begin our investigation, we remark that torsion had a first property that was unlike what any other space–time or gauge fields had, and that it comes as a general feature of the geometry and not from a symmetry principle, with the consequence that there is no symmetry protecting it from being massive. Thus, the torsional field Equations (141) are such that, in the presence of a massive field, they can be taken in the approximation in which the dynamical term is negligible compared to the mass term. Thus, we may write

$$M^2 W^\mu \approx X\overline{\psi}\gamma^\mu\boldsymbol{\pi}\psi \tag{165}$$

yielding an algebraic equation that can be used to have torsion substituted in all other field equations in terms of the spin of the spinor, so that all torsional contributions can effectively be converted into spin–spin interactions.

This is the so-called effective approximation.

Let us now move back to the physical field equations, which consist of expressions (141)–(144). These equations, by employing the variational formalism, can be derived from a dynamical action whose Lagrangian is

$$\mathscr{L} = -\tfrac{1}{4}(\partial W)^2 + \tfrac{1}{2}M^2 W^2 - \tfrac{1}{k}R - \tfrac{2}{k}\Lambda - \tfrac{1}{4}F^2 +$$
$$+ i\overline{\psi}\gamma^\mu\nabla_\mu\psi - X\overline{\psi}\gamma^\mu\boldsymbol{\pi}\psi W_\mu - m\overline{\psi}\psi \tag{166}$$

where torsion is already decomposed. Equivalently,

$$\mathscr{L} = -\tfrac{1}{4}(\partial W)^2 + \tfrac{1}{2}M^2 W^2 - \tfrac{1}{k}R - \tfrac{2}{k}\Lambda - \tfrac{1}{4}F^2 +$$
$$+ i\overline{\psi}_L\gamma^\mu\nabla_\mu\psi_L + i\overline{\psi}_R\gamma^\mu\nabla_\mu\psi_R +$$
$$+ X\overline{\psi}_L\gamma^\mu\psi_L W_\mu - X\overline{\psi}_R\gamma^\mu\psi_R W_\mu -$$
$$- m\overline{\psi}_R\psi_L - m\overline{\psi}_L\psi_R \tag{167}$$

in which the chiral split is already done.

In effective approximation, the Lagrangian becomes

$$\mathscr{L}_{\text{effective}} = -\tfrac{1}{k}R - \tfrac{2}{k}\Lambda - \tfrac{1}{4}F^2 +$$
$$+ i\overline{\psi}\gamma^\mu\nabla_\mu\psi + \tfrac{1}{2}\tfrac{X^2}{M^2}\overline{\psi}\gamma^\mu\psi\overline{\psi}\gamma_\mu\psi - m\overline{\psi}\psi \tag{168}$$

where (102) was used. Equivalently,

$$\begin{aligned}
\mathscr{L}_{\text{effective}} = &-\tfrac{1}{k}R - \tfrac{2}{k}\Lambda - \tfrac{1}{4}F^2 + \\
&+ i\overline{\psi}_L\gamma^\mu\nabla_\mu\psi_L + i\overline{\psi}_R\gamma^\mu\nabla_\mu\psi_R + \\
&+ \tfrac{X^2}{M^2}\overline{\psi}_L\gamma^\mu\psi_L\overline{\psi}_R\gamma_\mu\psi_R - \\
&- m\overline{\psi}_R\psi_L - m\overline{\psi}_L\psi_R
\end{aligned} \tag{169}$$

which is exactly the Lagrangian of the Nambu–Jona–Lasinio model [32,33].

As (102) shows, it is precisely the axial-vector nature of the field that produces the inversion of the sign of the potential, making the contact interaction attractive.

In addition, as it is clear, such an interaction takes place between two chiral projections.

In fact, general knowledge of the NJL model shows that the torsionally-induced spin–spin contact interaction is an attraction between the two chiral parts of the spinor.

We recall that the role of the Higgs boson is analogous.

This is not surprising since the torsion–spin coupling is the axial-vector analog of the scalar Yukawa coupling. In fact, if the effective Lagrangian (168) is further re-arranged in terms of (102), it can be put in the form

$$\begin{aligned}
\mathscr{L}_{\text{effective}}^{\text{spinor}} = &\, i\overline{\psi}\gamma^\mu\nabla_\mu\psi + \\
&+ \tfrac{1}{2}\tfrac{X^2}{M^2}(|\overline{\psi}\psi|^2 + |i\overline{\psi}\boldsymbol{\pi}\psi|^2) - m\overline{\psi}\psi
\end{aligned} \tag{170}$$

as the Lagrangian of the spinor field complemented with the torsionally-induced spin-contact interactions. On the other hand, in the standard model of particle physics [34], we might take into account the Lagrangian for the electron in the presence of the Higgs interaction alone. If the Higgs is taken in effective approximation, we have

$$M^2 H \approx -\tfrac{Y}{2}\overline{e}e \tag{171}$$

which is analogous to (165) in scalar form. Plugging it into the standard model Lagrangian gives

$$\mathscr{L}_{\text{effective}}^{\text{electron}} = i\overline{e}\gamma^\mu\nabla_\mu e + \tfrac{Y^2}{4M^2}|\overline{e}e|^2 - m\overline{e}e \tag{172}$$

for the electronic field with the Higgs-induced interaction. The comparison between (170) and (172) shows that

$$\mathscr{V}_{\text{effective}}^{\text{spinor}} = -\tfrac{1}{2}\tfrac{X^2}{M^2}(|\overline{\psi}\psi|^2 + |i\overline{\psi}\boldsymbol{\pi}\psi|^2) \tag{173}$$

$$\mathscr{V}_{\text{effective}}^{\text{electron}} = -\tfrac{Y^2}{4M^2}|\overline{e}e|^2 \tag{174}$$

meaning that torsion gives a self-interaction with a scalar part and a pseudo-scalar part, so spin dependent, while the Higgs gives rise to a scalar self-interaction only. Apart from this, they are both attractive and occur between the chiral parts.

From the Lagrangian (170), we extract the potential

$$\mathscr{V} = -\tfrac{X^2}{2M^2}\left(|\overline{\psi}\psi|^2 + |i\overline{\psi}\boldsymbol{\pi}\psi|^2\right) \tag{175}$$

which is negative, as expected for attractive interactions, and so the energy is the kinetic energy plus the potential energy, given by the general expression according to

$$\mathscr{E} = \mathscr{K} - \tfrac{X^2}{2M^2}\left(|\overline{\psi}\psi|^2 + |i\overline{\psi}\boldsymbol{\pi}\psi|^2\right) \tag{176}$$

and we recall that all quantities are densities. In fact, a straightforward dimensional analysis shows that we have

$$E = K - \tfrac{X^2}{2M^2}\tfrac{1}{V} \tag{177}$$

having interpreted $|\overline{\psi}\psi|^2+|i\overline{\psi}\pi\psi|^2 = V^{-2}$ as inverse volume, which is reasonable at least on dimensional grounds. On the other hand, it is possible to compute what turns out to be the expression for the internal energy of a van der Waals gas with negative pressure, given by

$$U = T - C^2 \frac{1}{V} \tag{178}$$

in terms of a generic constant $C$, as it is known from general thermodynamic arguments.

Because thermodynamically the kinetic energy can be interpreted as the temperature, and of course the energy is the internal energy, then the formal similarities of these two apparently unrelated expressions are striking.

In this thermodynamic analogy, we have that the single spinor field can be seen as a matter distribution behaving in the same way in which a van der Waals attractive gas with attractive intermolecular forces would [35].

Consider now the pair of second-order derivative Equations (157) and (158) with $K_\mu \approx 2XW_\mu$ and $G_\mu \approx 0$ and implement the torsion effective approximation: (157) becomes

$$\nabla^2\phi - 4X^4 M^{-4}\phi^5 + 2X^2 M^{-2}K\cdot s\phi^3 -$$
$$-|\nabla\beta/2|^2\phi + m^2\phi = 0 \tag{179}$$

with a quintic potential. We see that such a nonlinear potential is attractive.

Summarizing, in the effective approximation, torsional interactions give rise to a contact force much in the same way in which the Higgs field would, with these two forces being similarly attractive and chiral. In addition, we have seen that the torsional potential would also be analogous to the internal energy of an attractive van der Waals gas.

Consequently, insofar as this effective approximation holds, there is a clear indication that torsion is a sort of internal binding force, a tension, localizing the spinor.

### 4.1.2. Non-Relativistic Limit

In the initial section in which we introduced kinematic quantities, it was clear that tensors and gauge fields were characterized by general definition while spinors were defined in a way that was strongly dependent on the background being a (1 + 3)-dimensional space–time. Therefore, in such a space–time, the spinorial transformation law has a total of six parameters while spinor fields defined in terms of this transformation have a total of eight real components, and we have seen how to remove six components from the spinor field leaving it with two physical degrees of freedom.

However, now one might wonder what would happen when we consider the non-relativistic limit. In such a limit of small velocities, boosts can no longer be viable transformation laws and so time gets frozen, reducing the background to effectively be a three-dimensional space. In this case, spinorial transformation laws would possess a total of three parameters while spinor fields defined by this transformation would have four real components, so that we could remove three of the components from the spinor, hence leaving it with only one physical degree of freedom and nothing more.

To be mathematically precise, in the (1 + 3)-dimensional space–time, the spinor can always be written as (105) like

$$\psi = \phi e^{-i\alpha} \begin{pmatrix} e^{\frac{i}{2}\beta} \\ 0 \\ e^{-\frac{i}{2}\beta} \\ 0 \end{pmatrix} \tag{180}$$

in the representation we used throughout this presentation, called chiral representation, with Yvon–Takabayashi angle expressed in terms of imaginary exponentials. It is, however, possible to introduce another representation in which the Yvon–Takabayashi is expressed

in terms of real circular functions, called standard representation, obtained via the unitary transformation

$$U = \frac{1}{\sqrt{2}} \begin{pmatrix} \mathbb{I} & \mathbb{I} \\ -\mathbb{I} & \mathbb{I} \end{pmatrix} \tag{181}$$

which operates on gamma matrices to give

$$\gamma^0 = \begin{pmatrix} \mathbb{I} & 0 \\ 0 & -\mathbb{I} \end{pmatrix} \tag{182}$$

$$\gamma^K = \begin{pmatrix} 0 & \sigma^K \\ -\sigma^K & 0 \end{pmatrix} \tag{183}$$

so that

$$\sigma^{0A} = \frac{1}{2} \begin{pmatrix} 0 & \sigma^A \\ \sigma^A & 0 \end{pmatrix} \tag{184}$$

$$\sigma_{AB} = -\frac{i}{2} \varepsilon_{ABC} \begin{pmatrix} \sigma^C & 0 \\ 0 & \sigma^C \end{pmatrix} \tag{185}$$

and on spinors to give

$$\psi = \sqrt{2} \phi e^{-i\alpha} \begin{pmatrix} \cos \frac{\beta}{2} \\ 0 \\ -i \sin \frac{\beta}{2} \\ 0 \end{pmatrix} \tag{186}$$

in general. In (1 + 3)-dimensional space–times, spinors can always be written as above.

In three-dimensional space, the spinor can always be written according to

$$\psi = \phi e^{-i\alpha} \begin{pmatrix} 1 \\ 0 \end{pmatrix} \tag{187}$$

and the representation is unique.

Upon comparison, it becomes easy to see that the non-relativistic limit requires a small spatial part of the velocity $u^a$ but also a small Yvon–Takabayashi angle $\beta$ and, when this is accomplished, we have that, in standard representation, the spinor reduces to the form

$$\psi = \sqrt{2} \phi e^{-i\alpha} \begin{pmatrix} 1 \\ 0 \\ 0 \\ 0 \end{pmatrix} \tag{188}$$

where the lower component has vanished, and the upper component has reduced to

$$\psi = \phi e^{-i\alpha} \begin{pmatrix} 1 \\ 0 \end{pmatrix} \tag{189}$$

up to an overall constant, which is irrelevant.

It is also worth noticing that, so far, we have been able to obtain a procedure of non-relativistic limit that involves no definition of momentum. However, if the momentum in (163) is considered, we would see that only in the case in which all spin contributions are negligible can the explicit form of the momentum (164) reduce to

$$P^\nu \approx m \cos \beta u^\nu \tag{190}$$

so that the non-relativistic limit is given as a small spatial part of $P^a$ as commonly used.

Therefore, we have that the non-relativistic limit is implemented by the requirement that, when written in standard representation, the spinor loses its lower component

$$\psi \to \sqrt{2}\phi e^{-i\alpha} \begin{pmatrix} 1 \\ 0 \\ 0 \\ 0 \end{pmatrix} \tag{191}$$

and this is why this component is called small component.

Equivalently, we have that the conditions

$$u^a \to \begin{pmatrix} 1 \\ 0 \\ 0 \\ 0 \end{pmatrix} \tag{192}$$

$$\beta \to 0 \tag{193}$$

are what implements the non-relativistic limit.

In addition, additionally, if the spin is negligible, then

$$P^a \to \begin{pmatrix} m \\ 0 \\ 0 \\ 0 \end{pmatrix} \tag{194}$$

is the final form of non-relativistic limit, and the one that is normally employed.

We notice that, because $u^a$ is the velocity and, as we said, $\beta$ is already linked to the internal dynamics, then, in a non-relativistic limit, the spinor loses both the overall and the internal motions, which is intuitive. In addition, it is remarkable that the spinorial lower component is connected to *Zitterbewegung* effects which are yet another signature of internal dynamics [36]. There seems to be a very tight relation linking the Yvon–Takabayashi angle with effects of *Zitterbewegung* as manifestations of internal dynamics for the Dirac spinorial matter fields in general [37].

### 4.2. Massless Case

In this part, we will study the complementary situation given when both torsion and spinors are massless.

Ultra-Relativistic Limit

Let us now consider what happens when torsion as well as the Dirac spinor are both massless. The torsional field Equations (141) become

$$\nabla_\rho (\partial W)^{\rho\mu} = X\overline{\psi}\gamma^\mu \pi \psi \tag{195}$$

which are analogous to the electro-dynamic field equations apart from the fact that these above are parity-odd. This aside, both are vector field equations in a massless case, and, as such, we should expect some symmetry to be present. The full Lagrangian in the case of masslessness also for the spinor field is given by the following:

$$\begin{aligned} \mathscr{L}_{\text{massless}} = &-\tfrac{1}{4}(\partial W)^2 - \tfrac{1}{k}R - \tfrac{2}{k}\Lambda - \tfrac{1}{4}F^2 + \\ &+ i\overline{\psi}_L\gamma^\mu\nabla_\mu\psi_L + i\overline{\psi}_R\gamma^\mu\nabla_\mu\psi_R + \\ &+ X\overline{\psi}_L\gamma^\mu\psi_L W_\mu - X\overline{\psi}_R\gamma^\mu\psi_R W_\mu \end{aligned} \tag{196}$$

as it is straightforward to see.

This is invariant for the transformation

$$W'_\nu = W_\nu - \partial_\nu \omega \tag{197}$$

with

$$\psi'_L = e^{-iX\omega}\psi_L \qquad \psi'_R = e^{iX\omega}\psi_R \tag{198}$$

or in compact form

$$\psi' = e^{iX\omega\pi}\psi \tag{199}$$

known as chiral gauge transformation.

Additionally, expression (105) can also be written as

$$\psi = \phi e^{-i\alpha} e^{-\frac{i}{2}\beta\pi} \begin{pmatrix} 1 \\ 0 \\ 1 \\ 0 \end{pmatrix} \tag{200}$$

and, from this expression, it is clear that it is always possible to perform a chiral gauge transformation taking the local parameter to be $\beta = 2X\omega$ and leaving

$$\psi' = \phi e^{-i\alpha} \begin{pmatrix} 1 \\ 0 \\ 1 \\ 0 \end{pmatrix} \tag{201}$$

in terms of the module alone: this has to be expected, as symmetries come with redundant information that can be removed by reducing the fields, and because in this case the chiral symmetry is an additional symmetry with one parameter, we have to expect that one degree of freedom be removed. It is clear that the only degree of freedom to remain is the one that cannot be removed in any way whatsoever that is the module.

Because in the massless approximation the two chiral Lagrangians become separable, the two chiral projections are independent, and therefore the Yvon–Takabayashi angle can be vanished, since it carries no information.

This is yet another fact that supports the evidence for which the Yvon–Takabayashi angle can be related to internal dynamics and *Zitterbewegung* for spinor fields.

In addition, this is possible because of the attractiveness that characterizes the axial-vector massive torsion mediation of the chiral mutual interaction within the spinor field.

If torsion were not an axial-vector, the chiral interaction would not be attractive, and, if such an attraction were not massive enough, it would not be sufficiently strong to grant stability for the bound-state spinorial field itself.

### *4.3. Two: Basic Applications*

This second chapter will be about applying the above theory to solve or discuss fundamental problems in modern physics: in the first section, we will tackle the problem of gravitational singularity formation. In the second section, we will discuss the problem of positivity of energy.

## 5. Consequences of Spin

In this first section, our main goal is to take into account the problem of the formation of gravitational singularities and face it in terms of the modifications brought by the presence of torsion interacting with spinors. Some comments on the Pauli exclusion principle will be made.

### 5.1. Singularity Avoidance

If we consider Einstein gravity on its own, it is remarkably difficult to overestimate its success. From planetary precession, through gravitational waves, to black holes, there is not a single prediction that has not been corroborated yet. In fact, if Dark Matter is just another form of matter, there is not a single effect, whether predicted or not, that has never been confirmed so far. Nevertheless, there is a black spot, theoretically.

The Hawking–Penrose theorem is a very general result showing how, under very general conditions on energy, gravitationally-induced singularities form. If true, such a result would constitute an indication that Einstein gravitation has to be generalized, or at least included in an extended framework. There are, in fact, several attempts at extended models, whether they are simple extensions of Einstein gravity, or major revisions of all Einstein concepts of a geometric theory in itself. All these models and theories are certainly worth our attention. However, at times, the solution to a given puzzle might well be much closer than expected. If we wish to try a solution that is based on the physics we already have, the most straightforward possibility is to use the torsion tensor.

Employing torsion to solve this problem has already been done [38]. However, contrary to the expectation that torsion could solve or at least alleviate this issue, Kerlick found that the issue was actually worsened. This way was then abandoned.

Nevertheless, to a more attentive examination, we may find a possible way out. A closer look at the reasons why torsion would enhance the formation of singularities will reveal that the gravitational field is increased because, in the energy density, there are positive contributions coming from the fact that torsional effects for the spin contact interaction of spinors are taken to be repulsive.

This happens to be the case because Kerlick considers the simplest generalization of Einstein gravity, the original Einstein–Sciama–Kibble theory, where torsion is tied to the spin in terms of the Newton gravitational constant.

However, as discussed above, a more general theory of torsion would, first of all, involve a torsion–spin coupling that is not the Newton gravitational constant, but which can be any possible constant and in particular a constant with the opposite sign. In addition, secondly, in the most general case in which torsion propagates, in the effective approximation, the torsion–spin coupling constant has an opposite sign necessarily.

In fact, in this case, in effective approximation, we found that we do have an attractive torsion effect, resulting in a negative potential in the energy density, decreasing gravitation and making the singularity formation avoidable.

Indeed, the torsional contribution could provide such a negative potential that the whole energy may turn negative, the gravitational field may turn repulsive, and singularity formation would be avoided necessarily.

To put words into expressions, take (142) contracted as

$$-R-4\Lambda = \tfrac{k}{2}(-M^2W^2+m\Phi) \tag{202}$$

and plug this back into the original equations to get

$$\begin{aligned} R^{\rho\sigma}+\Lambda g^{\rho\sigma} = \tfrac{k}{2}\big[ & \tfrac{1}{4}F^2g^{\rho\sigma}-F^{\rho\alpha}F^{\sigma}{}_{\alpha}+\tfrac{1}{4}(\partial W)^2g^{\rho\sigma}-(\partial W)^{\sigma\alpha}(\partial W)^{\rho}{}_{\alpha}+M^2W^{\rho}W^{\sigma}+ \\ & +\tfrac{i}{4}(\overline{\psi}\gamma^{\rho}\nabla^{\sigma}\psi-\nabla^{\sigma}\overline{\psi}\gamma^{\rho}\psi+\overline{\psi}\gamma^{\sigma}\nabla^{\rho}\psi-\nabla^{\rho}\overline{\psi}\gamma^{\sigma}\psi)- \\ & -\tfrac{1}{2}X(W^{\sigma}S^{\rho}+W^{\rho}S^{\sigma})-\tfrac{1}{2}m\Phi g^{\rho\sigma}\big] \end{aligned} \tag{203}$$

equivalent to those in the original form. For the singularity theorem in Einstein gravity, we have that the condition

$$R^{\rho\sigma}u_{\rho}u_{\sigma}\geqslant 0 \tag{204}$$

must be verified, and, when this is the case, then singularity formation will become inevitable. With no cosmological constant and neglecting electro-dynamics, we obtain that the condition to have singularity formation reads

$$[\tfrac{1}{4}(\partial W)^2 g^{\rho\sigma} - (\partial W)^{\sigma\alpha}(\partial W)^{\rho}{}_{\alpha} + \tfrac{i}{2}(\overline{\psi}\gamma^{\rho}\nabla^{\sigma}\psi - \nabla^{\sigma}\overline{\psi}\gamma^{\rho}\psi) +$$
$$+ M^2 W^{\rho}W^{\sigma} - XW^{\sigma}S^{\rho} - \tfrac{1}{2}m\Phi g^{\rho\sigma}]u_{\rho}u_{\sigma} \geqslant 0 \qquad (205)$$

and this is what we have to study.

In effective approximation, it becomes

$$\tfrac{i}{2}(\overline{\psi}\gamma^0\nabla_0\psi - \nabla_0\overline{\psi}\gamma^0\psi) - \tfrac{1}{2}m\Phi \geqslant 0 \qquad (206)$$

and because (144) in the effective approximation is

$$i\gamma^0\nabla_0\psi + i\vec{\gamma}\cdot\vec{\nabla}\psi - \tfrac{X^2}{M^2}S_{\sigma}\gamma^{\sigma}\boldsymbol{\pi}\psi - m\psi = 0 \qquad (207)$$

we may use this in the above to get

$$\tfrac{i}{2}(\vec{\nabla}\overline{\psi}\cdot\vec{\gamma}\psi - \overline{\psi}\vec{\gamma}\cdot\vec{\nabla}\psi) + \tfrac{X^2}{M^2}S_{\sigma}S^{\sigma} + \tfrac{1}{2}m\Phi \geqslant 0 \qquad (208)$$

whose structure is similar to the condition of Kerlick but with the sign of the nonlinear interaction inverted. We may now follow Kerlick argument by neglecting the derivative term, and, by employing (102), we get that

$$-4\tfrac{X^2}{M^2}\phi^4 + m\phi^2\cos\beta \geqslant 0 \qquad (209)$$

which for for large densities are be violated, and quite easily too.

Therefore, because of the torsion–spin coupling, the energy condition is not verified and gravitational singularity formation is no longer a necessity [39].

We already said that torsion in effective approximation generates interactions which, without the spin-dependent part, are similar to what we would get by using the Higgs potential. Therefore, it is not surprising that singularity avoidance could be achieved also by the Higgs [40]. The difference is in the mass scale: the Higgs potential can only be used to avoid singularities in black holes, as it does not work before symmetry breaking, while torsion can be used to avoid singularities for black holes and the big bang, since torsion is always a massive field even prior to any mass generation mechanism.

Notice that this mechanism is proper to the Einsteinian gravitation. In fact, in order for this mechanism to work, one must have a theory in which gravitation can become repulsive if the energy density switches sign and in which the energy density is allowed to switch its overall sign. None of this would ever be possible in a theory of gravitation in which the source is not the energy but the mass, since the mass can never be negative.

### 5.2. Pauli Exclusion

The above-commented mechanism with which one may avoid the formation of singularity at a gravitational level is reminiscent of the degeneracy pressure encountered in the usual treatment of neutron stars. Consequently, the correlated Pauli exclusion principle comes to the mind. Such a principle stems from the fact that, in the construction of electronic levels, obtained by solving the non-relativistic matter field equations in a Coulomb potential, the solutions are given in terms of a quantum number $n$ giving the energy level of the external shell, accounting for a total of $n^2$ electrons. However, the number of observed electrons $2n^2$ and hence there must be a two-fold degeneracy. This means that solutions of the matter field equation come in pairs of two, so that each electronic shell can be filled twice by the same state. The exclusion principle presented in this way is the original form by Pauli. Pauli's initial idea to assign a two-fold degeneracy was most straightforwardly that of introducing the concept of spin: the connection is very simple, based on the fact

that irreducible representations of particles of spin $s$ have exactly $d = 2s + 1$ independent components. For particles of spin $s = 1/2$, this means $d = 2/2 + 1 = 2$ components, so that it is possible to think that these two components be precisely the two states that account for the double state of multiplicity. Mathematically, this can be seen from the fact that the spinor field has, for each chiral part, two components. Indeed, recalling (105), we have that spinor fields can be written as

$$
\psi = \phi e^{-i\alpha} e^{-\frac{i}{2}\beta\pi}
\begin{pmatrix} 1 \\ 0 \\ 1 \\ 0 \end{pmatrix}
\quad \text{and} \quad
\psi = \phi e^{-i\alpha} e^{-\frac{i}{2}\beta\pi}
\begin{pmatrix} 0 \\ 1 \\ 0 \\ 1 \end{pmatrix}
\tag{210}
$$

where the first is a spin-up (spin $+1/2$) eigenstate while the second is a spin-down (spin $-1/2$) eigenstate. These are the two opposite-spin eigenvalues of the same eigenspinor. As a consequence of this structure, superposition of two opposite-spin spinors is allowed and thus, if the two initial spinors are solutions, then also their superposition is another solution. This mechanism is indeed what happens in the hydrogen atom.

Nevertheless, the Pauli exclusion principle is not only this. Such a principle must also include a mechanism for which no more than two states can superpose. Quantum mechanics does not solve this problem. In quantum field theory, however, a solution is proposed, and the commonly accepted paradigm is described by the spin-statistic theorem: this theorem says that in a theory that is Lorentz covariance and causal, with positive norms and energies, half-integer spin particles cannot occupy more than one state at a time (while integer spin particles can). However, for this result to take place, the theorem must engage, and this is subject to the conditions granted by its hypotheses. In a classical theory of fields, Lorentz covariance and causality are ensured, but positive norms and energies are not. In fact, we have seen that negative energies are not only possible but also needed to ensure the mechanism to avoid the formation of singularities.

It so appears that the exclusion principle and singularity avoidance can not both be implemented in the same framework. In addition, usually, the common behavior is that of implementing the spin-statistic theorem and leaving the singularity formation unsolved. However, one can instead consider the complementary position of ensuring singularity avoidance and leave the Pauli exclusion open.

However, in a theory where spinors interact with torsion, we have seen that, in effective approximation, the torsionally-induced spin–contact interactions of the spinor give rise to self-interactions for the spinor field. These nonlinear contributions in the matter field equations are enough to ensure that no superposition of two identical solutions can also be a solution. This entails the exclusion principle.

Notice that, in case the two solutions are not identical, that is, if the two solutions correspond to opposite spins, their superposition is allowed by the double-valuedness that characterizes the spinorial fields in general cases.

## 6. Conditions on Energy

In this second section, we intend to deepen the investigation of the problem of the positive energies. We conclude with comments on the macroscopic approximation.

### 6.1. Positive Energy

In the development of field theories, it is not uncommon for some properties to be present in a given approximation but not in the full theory. Thus, particles behave in a certain manner in classical mechanics and very differently in quantum mechanics, and quantum particles behave in a given way in quantum mechanics and rather differently in the relativistic version of quantum mechanics.

Following a bottom-up approach in terms of successive generalizations, fewer and fewer properties will be found within the most general theory that is possible. There is, however, a property that does not appear to follow such a pattern, which is the energy.

From classical mechanics to quantum mechanics, to relativistic quantum mechanics, to relativistic quantum mechanics of spinning fields, the energy of a particle is always taken to be positive, either because it is proven positive, or because we force it to be positive by correcting the theory in an appropriate way.

Forcing the energy to be positive does have a number of consequences, not only for the interpretation, but also to obtain results like the spin-statistic theorem as discussed above. However, we have seen that the exclusion principle can also be entailed in a different manner, and there should be no surprise in finding a generalization of field the theory in which some energies happen to be negative after all.

Allowing negative energies has considerable advantages too, not only for the fact that the mathematics tells us that they are possible, but also to obtain results like the avoidance of singularities as we had discussed before.

Just the same, even assuming that energies can be negative, we have that they will have to turn out to be positive in those approximations in which we know they are.

To see this is in fact the case, let us consider the energy given as the right-hand side of (142), and that is

$$
\begin{aligned}
T^{\rho\sigma} = {} & \tfrac{1}{4}F^2 g^{\rho\sigma} - F^{\rho\alpha}F^{\sigma}{}_{\alpha} + \\
& + \tfrac{1}{4}(\partial W)^2 g^{\rho\sigma} - (\partial W)^{\sigma\alpha}(\partial W)^{\rho}{}_{\alpha} + \\
& + M^2(W^{\rho}W^{\sigma} - \tfrac{1}{2}W^2 g^{\rho\sigma}) + \\
& + \tfrac{i}{4}(\overline{\psi}\gamma^{\rho}\nabla^{\sigma}\psi - \nabla^{\sigma}\overline{\psi}\gamma^{\rho}\psi + \overline{\psi}\gamma^{\sigma}\nabla^{\rho}\psi - \nabla^{\rho}\overline{\psi}\gamma^{\sigma}\psi) - \\
& - \tfrac{1}{2}X(W^{\sigma}\overline{\psi}\gamma^{\rho}\boldsymbol{\pi}\psi + W^{\rho}\overline{\psi}\gamma^{\sigma}\boldsymbol{\pi}\psi)
\end{aligned}
\tag{211}
$$

in general. In particular, as the electro-dynamic and torsional contributions are positive, we will consider only

$$
\begin{aligned}
E^{\rho\sigma} = {} & \tfrac{i}{4}(\overline{\psi}\gamma^{\rho}\nabla^{\sigma}\psi - \nabla^{\sigma}\overline{\psi}\gamma^{\rho}\psi + \\
& + \overline{\psi}\gamma^{\sigma}\nabla^{\rho}\psi - \nabla^{\rho}\overline{\psi}\gamma^{\sigma}\psi) - \\
& - \tfrac{1}{2}X(W^{\sigma}\overline{\psi}\gamma^{\rho}\boldsymbol{\pi}\psi + W^{\rho}\overline{\psi}\gamma^{\sigma}\boldsymbol{\pi}\psi)
\end{aligned}
\tag{212}
$$

as pure spinorial contribution and which is not positive.

To better see this, we go in the frame where the spinor assumes the polar form (105) in which

$$
\begin{aligned}
E_{\rho\sigma} = {} & \phi^2\Big[P_{\sigma}u_{\rho} + P_{\rho}u_{\sigma} + \\
& + [\tfrac{1}{4}(\Omega^{ij}{}_{\sigma}\varepsilon_{\rho ijk} + \Omega^{ij}{}_{\rho}\varepsilon_{\sigma ijk})\eta^{ka} + \\
& + \xi^{a}_{\rho}(\nabla\beta/2 - XW)_{\sigma} + \xi^{a}_{\sigma}(\nabla\beta/2 - XW)_{\rho}]s_a\Big]
\end{aligned}
\tag{213}
$$

whose time-time component is not positive defined as the straightforward check would immediately show.

In it, the momentum is given by (164) as

$$
P^{\nu} = m\cos\beta u^{\nu} + Y_{\mu}u^{[\mu}s^{\nu]} + Z_{\mu}s_{\rho}u_{\sigma}\varepsilon^{\mu\rho\sigma\nu}
\tag{214}
$$

whose time component is also not positive defined.

However, if we could justify the assumption in terms of which we neglect all contributions coming from the spin, then the energy would reduce to

$$
E_{00} = 2\phi^2 P_0 u_0
\tag{215}
$$

for the time-time component.

The momentum becomes

$$P^0 = m \cos \beta u^0 \tag{216}$$

for the time component.

Therefore, if now the Yvon–Takabayashi angle vanishes, then the energy is ensured to be positive defined.

Summarizing, we can say that, if

$$\beta \to 0 \tag{217}$$
$$s_a \to 0 \tag{218}$$

then the energy of spinor fields is necessarily positive [41].

These two conditions together condense a very simple situation, as we are going to discuss in what follows.

### 6.2. Macroscopic Limit

In the previous part, we have discussed how the energy is positive if $\beta \to 0$ and $s_a \to 0$ happen to occur.

To understand the meaning of these conditions, let us consider again the field equations for the gravitational field and for electro-dynamics (142) and (143) and compute the divergences: they are respectively given by

$$\nabla_\rho [\tfrac{1}{4} F^2 g^{\rho\sigma} - F^{\rho\alpha} F^\sigma{}_\alpha +$$
$$+ \tfrac{1}{4} (\partial W)^2 g^{\rho\sigma} - (\partial W)^{\sigma\alpha} (\partial W)^\rho{}_\alpha +$$
$$+ M^2 (W^\rho W^\sigma - \tfrac{1}{2} W^2 g^{\rho\sigma}) +$$
$$+ \tfrac{i}{4} (\overline{\psi} \gamma^\rho \nabla^\sigma \psi - \nabla^\sigma \overline{\psi} \gamma^\rho \psi + \overline{\psi} \gamma^\sigma \nabla^\rho \psi - \nabla^\rho \overline{\psi} \gamma^\sigma \psi) -$$
$$- \tfrac{1}{2} X (W^\sigma \overline{\psi} \gamma^\rho \pi \psi + W^\rho \overline{\psi} \gamma^\sigma \pi \psi)] = 0 \tag{219}$$

and

$$\nabla_\mu (\overline{\psi} \gamma^\mu \psi) = 0 \tag{220}$$

identically, as we already know from the first chapter.

By substituting the polar form (105) and implementing the above conditions $\beta \to 0$ and $s_a \to 0$, we get

$$\nabla_\rho (\tfrac{1}{4} F^2 g^{\rho\sigma} - F^{\rho\alpha} F^\sigma{}_\alpha + 2m\phi^2 u^\rho u^\sigma) = 0 \tag{221}$$

and

$$\nabla_\mu (2\phi^2 u^\mu) = 0 \tag{222}$$

as it is straightforward to see.

Evaluating the divergence of the former and employing the latter, we obtain the expression

$$-2q\phi^2 u^\alpha F^\sigma{}_\alpha + 2m\phi^2 u^\rho \nabla_\rho u^\sigma = 0 \tag{223}$$

having used the Maxwell field equations.

After the necessary simplifications, we get

$$m u^\rho \nabla_\rho u^\sigma = q F^{\sigma\alpha} u_\alpha \tag{224}$$

which is just the Newton law in the presence of Lorentz force.

This is what we have in macroscopic approximation.

Thus, we can interpret $\beta \to 0$ and $s_a \to 0$ as the conditions that implement the known macroscopic approximation.

This is reasonable because vanishing the internal dynamics and all information about internal structures essentially means that we are considering situations where internal contributions are concealed within the spinorial field, which means we are in macroscopic approximation.

Spinor fields have energy density that can be negative as a consequence of all contributions of spin and internal dynamics, and it is only when these are hidden that the positivity of the energy density is also ensured for spinors.

*6.3. Three: Special Models*

This third and last chapter will be about the application of the above theory for phenomenological cases: we will in fact consider what the effects are of torsion for the two standard models of particles and cosmology.

## 7. Particles and Cosmology

In the first chapter, we have encountered the theorem of the polar form, which specified that, if both scalars $\Theta$ and $\Phi$ do not vanish identically, then we can always find special frames where the most general spinor is as in (105).

However, what if $\Theta = \Phi \equiv 0$ everywhere? The answer to this question has already been given in [15], and it is that we could still find a special frame in which the most general spinor can be written in some type of polar form.

Specifically, if $\Theta = \Phi \equiv 0$, then we can always find special frames where the most general spinor is given by

$$\psi = \tfrac{1}{\sqrt{2}} (\mathbb{I} \cos \tfrac{\alpha}{2} - \boldsymbol{\pi} \sin \tfrac{\alpha}{2}) \begin{pmatrix} 1 \\ 0 \\ 0 \\ 1 \end{pmatrix} \tag{225}$$

up to the reversal of the third axis.

Spinor fields undergoing these constraints are called flag-dipoles, and they contain two special cases: one with constraint $S^a = 0$ and called flagpoles, written as

$$\psi = \tfrac{1}{\sqrt{2}} \begin{pmatrix} 1 \\ 0 \\ 0 \\ 1 \end{pmatrix} \tag{226}$$

up to the reversal of the third axis and extinguishing the class of Majorana spinors; the other with a constraint given by $M^{ab} = 0$ and called dipoles, written as

$$\psi = \begin{pmatrix} 1 \\ 0 \\ 0 \\ 0 \end{pmatrix} \tag{227}$$

up to the reversal of the third axis and the switch between chiral parts and accounting for the Weyl spinors.

Therefore, as it can be seen quite clearly, Majorana as well as Weyl spinors can always be Lorentz transformed into a frame in which they remain with a fixed structure, and, consequently, they have no degree of freedom at all.

This may sound surprising, and thus we are going to give a direct proof of this statement for the Weyl spinors.

To do that, consider a general Weyl spinor, for instance left-handed, in the form

$$\psi = \begin{pmatrix} ae^{i\alpha} \\ be^{i\beta} \\ 0 \\ 0 \end{pmatrix}$$

where the two complex components have been written in polar form. Consider now as Lorentz transformation the rotation of angle $\theta$ around the second axis given by

$$\Lambda_{R2} = \begin{pmatrix} \cos\theta/2 & -\sin\theta/2 & 0 & 0 \\ \sin\theta/2 & \cos\theta/2 & 0 & 0 \\ 0 & 0 & \cos\theta/2 & -\sin\theta/2 \\ 0 & 0 & \sin\theta/2 & \cos\theta/2 \end{pmatrix}$$

followed by the rotation of angle $\varphi$ around the third axis

$$\Lambda_{R3} = \begin{pmatrix} e^{i\varphi/2} & 0 & 0 & 0 \\ 0 & e^{-i\varphi/2} & 0 & 0 \\ 0 & 0 & e^{i\varphi/2} & 0 \\ 0 & 0 & 0 & e^{-i\varphi/2} \end{pmatrix}$$

applied to the spinor. The results are given by expression

$$\psi' = \begin{pmatrix} \cos\theta/2 & -\sin\theta/2 & 0 & 0 \\ \sin\theta/2 & \cos\theta/2 & 0 & 0 \\ 0 & 0 & \cos\theta/2 & -\sin\theta/2 \\ 0 & 0 & \sin\theta/2 & \cos\theta/2 \end{pmatrix} \cdot$$

$$\cdot \begin{pmatrix} e^{i\varphi/2} & 0 & 0 & 0 \\ 0 & e^{-i\varphi/2} & 0 & 0 \\ 0 & 0 & e^{i\varphi/2} & 0 \\ 0 & 0 & 0 & e^{-i\varphi/2} \end{pmatrix} \begin{pmatrix} ae^{i\alpha} \\ be^{i\beta} \\ 0 \\ 0 \end{pmatrix}$$

and that is

$$\psi' = \begin{pmatrix} a\cos\theta/2 e^{i\varphi/2}e^{i\alpha} - b\sin\theta/2 e^{-i\varphi/2}e^{i\beta} \\ a\sin\theta/2 e^{i\varphi/2}e^{i\alpha} + b\cos\theta/2 e^{-i\varphi/2}e^{i\beta} \\ 0 \\ 0 \end{pmatrix}$$

after multiplication. The spin-down component is zero if

$$a\sin\theta/2 e^{i\varphi/2}e^{i\alpha} + b\cos\theta/2 e^{-i\varphi/2}e^{i\beta} = 0$$

which can be worked out to be

$$\frac{a}{b}e^{i(\alpha-\beta)} = -e^{-i\varphi}\cot\theta/2$$

splitting into

$$\cot\theta/2 = -a/b$$
$$\varphi = \beta - \alpha$$

for the two angles. Thus, we can always find a combination of two rotations that brings the spin-down component to vanish identically. When this is done, we have

$$\psi' = \sqrt{a^2 + b^2} e^{i(\beta + \alpha)/2} \begin{pmatrix} 1 \\ 0 \\ 0 \\ 0 \end{pmatrix}$$

for spin-up Weyl spinors. With another rotation of angle $\zeta = \beta + \alpha$ around the third axis given as the above

$$\mathbf{\Lambda}_{R3} = \begin{pmatrix} e^{-i\zeta/2} & 0 & 0 & 0 \\ 0 & e^{i\zeta/2} & 0 & 0 \\ 0 & 0 & e^{-i\zeta/2} & 0 \\ 0 & 0 & 0 & e^{i\zeta/2} \end{pmatrix}$$

the phase can also be vanished. Thus, we have

$$\psi'' = \sqrt{a^2 + b^2} \begin{pmatrix} 1 \\ 0 \\ 0 \\ 0 \end{pmatrix}$$

and employing a boost of rapidity $\eta = \ln |a^2 + b^2|$ along the third axis given by

$$\mathbf{\Lambda}_{B3} = \begin{pmatrix} e^{-\eta/2} & 0 & 0 & 0 \\ 0 & e^{\eta/2} & 0 & 0 \\ 0 & 0 & e^{\eta/2} & 0 \\ 0 & 0 & 0 & e^{-\eta/2} \end{pmatrix}$$

the module is also removed, and we get

$$\psi''' = \begin{pmatrix} 1 \\ 0 \\ 0 \\ 0 \end{pmatrix}$$

for the final form of the Weyl spinor. Obviously, the same would be true if we intended to keep only the spin-down component. In addition, of course, the same remains true for the right-handed case. This result for Weyl spinors is general.

Although more calculations would be needed, it would still be straightforward to see that, by employing exactly the same method, we would obtain exactly the same result also if we were to consider the Majorana spinors.

Such a result may be surprising, but it is a mathematical consequence of the definition of Majorana and Weyl spinors alone and therefore it is true in full generality.

Thus, these spinors do not have degrees of freedom.

If we take this to conclude that these spinors cannot be physical, then we are bound to accept that such spinors cannot form the matter content of any theory, in particular, the standard model of particle physics as we know.

This leaves us with a remarkable consequence: if these spinors, and in particular Weyl spinors, cannot be used in physics, and in particular in the standard model of particle physics, then we cannot employ neutrinos as defined at the moment. Neutrinos need be right-handed too, and, after the symmetry breaking, they must get a Dirac mass.

Because the charge count of the standard model cannot change, neutrinos are sterile.

We will next try to see what happens when sterile neutrinos with a Dirac mass term are then included. Of course, the first application is neutrino oscillations.

We now try to see what the effects of torsion can be. However, in order to do so, we have first to make one little digression in order to generalize the theory.

Throughout the entire presentation, we have been considering single spinor fields, but clearly the treatment of two spinor fields, or even more spinor fields, is doable, and it is achieved by replicating the spinor field Lagrangian as many times as the number of independent spinor fields.

For instance, in the case of two spinor fields, we have

$$
\begin{aligned}
\mathscr{L} = &-\tfrac{1}{4}(\partial W)^2 + \tfrac{1}{2}M^2 W^2 - \tfrac{1}{k}R - \tfrac{2}{k}\Lambda - \tfrac{1}{4}F^2 + \\
&+ i\overline{\psi}_1 \gamma^\mu \nabla_\mu \psi_1 + i\overline{\psi}_2 \gamma^\mu \nabla_\mu \psi_2 + \\
&- X_1 \overline{\psi}_1 \gamma^\mu \boldsymbol{\pi} \psi_1 W_\mu - X_2 \overline{\psi}_2 \gamma^\mu \boldsymbol{\pi} \psi_2 W_\mu + \\
&- m_1 \overline{\psi}_1 \psi_1 - m_2 \overline{\psi}_2 \psi_2
\end{aligned}
\tag{228}
$$

as it is reasonable to expect.

Taking the variation with respect to torsion gives

$$
\begin{aligned}
\nabla_\nu (\partial W)^{\nu\mu} + M^2 W^\mu = &\, X_1 \overline{\psi}_1 \gamma^\mu \boldsymbol{\pi} \psi_1 + \\
&+ X_2 \overline{\psi}_2 \gamma^\mu \boldsymbol{\pi} \psi_2
\end{aligned}
\tag{229}
$$

as the torsion field equations with two sources.

In effective approximation, we obtain expressions

$$
M^2 W^\mu \approx X_1 \overline{\psi}_1 \gamma^\mu \boldsymbol{\pi} \psi_1 + X_2 \overline{\psi}_2 \gamma^\mu \boldsymbol{\pi} \psi_2
\tag{230}
$$

which can be plugged back into the Lagrangian giving

$$
\begin{aligned}
\mathscr{L} = &-\tfrac{1}{k}R - \tfrac{2}{k}\Lambda - \tfrac{1}{4}F^2 + i\overline{\psi}_1 \gamma^\mu \nabla_\mu \psi_1 + i\overline{\psi}_2 \gamma^\mu \nabla_\mu \psi_2 + \\
&+ \tfrac{1}{2}\left|\tfrac{X_1}{M}\right|^2 \overline{\psi}_1 \gamma^\mu \psi_1 \overline{\psi}_1 \gamma_\mu \psi_1 + \tfrac{1}{2}\left|\tfrac{X_2}{M}\right|^2 \overline{\psi}_2 \gamma^\mu \psi_2 \overline{\psi}_2 \gamma_\mu \psi_2 - \\
&- \tfrac{X_1}{M}\tfrac{X_2}{M} \overline{\psi}_1 \gamma^\mu \boldsymbol{\pi} \psi_1 \overline{\psi}_2 \gamma_\mu \boldsymbol{\pi} \psi_2 - m_1 \overline{\psi}_1 \psi_1 - m_2 \overline{\psi}_2 \psi_2
\end{aligned}
\tag{231}
$$

in which each spinor has self-interaction and between spinors there is mutual interaction.

The extension to three spinor fields, or $n$ spinor fields, is similar: there are $n$ self-interactions, always attractive, and $\tfrac{1}{2}n(n-1)$ mutual interactions, being either attractive or repulsive according to $X_i X_j$ being positive or negative.

This extension is interesting for $n = 3$ because this is the situation we have for neutrinos. By neglecting all the interactions apart from the effective interactions, and in them neglecting the self-interaction so to have only the mutual interactions, one may calculate the Hamiltonian

$$
\mathscr{H} = \sum_{ij} \overline{\nu}_i (U_{ij} - X_i X_j \gamma^\mu \boldsymbol{\pi} \nu_i \overline{\nu}_j \boldsymbol{\pi} \gamma_\mu) \nu_j
\tag{232}
$$

where the Latin indices run over the three labels associated with the three different flavors of neutrinos. Hence, the matrix $U_{ij} - X_i X_j \gamma^\mu \boldsymbol{\pi} \nu_i \overline{\nu}_j \boldsymbol{\pi} \gamma_\mu$ is the combination of the constant matrix $U_{ij}$ describing kinematic phases that arise from the mass terms, as usual, plus the field-dependent matrix $X_i X_j \gamma^\mu \boldsymbol{\pi} \nu_i \overline{\nu}_j \boldsymbol{\pi} \gamma_\mu$ describing the dynamical phases that arise from the torsionally-induced nonlinear potentials, those of the present theory.

Dealing with the nonlinear potentials is problematic, but, in reference [42], this problem is solved by taking neutrinos dense enough to make the torsion field background homogeneous and thus constant. The phase difference is

$$
\Delta\Phi \approx \left( \frac{\Delta m^2}{2E} + \frac{1}{4}|W^0 - W^3| \right) L
\tag{233}
$$

having assumed $W_1 = W_2 = 0$ and where $L$ is the length of the oscillations. In [43], it was seen that (233) in the case in which the neutrino mass difference is small becomes

$$\Delta\Phi \approx \left( \Delta m^2 + m \frac{X^2_{\text{eff}}}{4M^2} |\bar{\nu}\gamma_\mu\nu\bar{\nu}\gamma^\mu\nu|^{\frac{1}{2}} \right) \frac{L}{2E} \tag{234}$$

where $m$ is the value of the nearly-equal masses of neutrinos while $X^2_{\text{eff}}$ is a combination of the coupling constants and with the dependence $L/E$ as the ratio between length and energy of the oscillations, as it is expected.

The phase difference due to the oscillation has the kinematic contribution, as a difference of the squared masses, plus a dynamic contribution, proportional to the neutrino mass density distribution. The novelty torsion introduced is that, even in the case in which neutrino masses were to be non-zero but with insufficient non-degeneracy in mass spectrum, we might still have oscillations, and therefore an ampler margin of freedom before having some tension. Notice also that both $m$ and $X^2_{\text{eff}}$ depend on the masses and coupling constants of the two neutrinos involved so that they would be different for another pair of neutrinos, making it clear how the parameters of the oscillation depend on the specific pair of neutrinos, as it should be.

This is an immediate and clear effect that the neutrinos with Dirac mass term and interacting in terms of torsion give to us for some new physical insight beyond what is commonly expected from the standard model of particles.

What about the standard model of cosmology? To give an answer to this question, we move on to examine some consequences torsion may have for Dark Matter.

To begin with, we specify that, although we still do not exactly know what dark matter is, nevertheless it has to be a form of matter: albeit many models may fit galactic rotation curves, only dark matter as a real form of matter fits all galactic behaviors [44].

Hence, given dark matter as a form of matter, massive and weakly interacting, we will additionally take it to be described by $\frac{1}{2}$-spin spinor fields. This makes it possible to have the effects due to torsional interactions.

In reference [45], the torsional effects have been studied in a classical context to see how galactic dynamics could be modified by torsion, and, in [46], we have applied those results to the case in which torsion was coupled to spinors to see how galactic dynamics could be modified by torsion and how torsion could be sourced by dark matter.

Thus, here as before, torsion is not used as an alternative but as a correction over pre-existing physics. Having this in mind, we recall that, in [46], we showed how, if spinors are the source of torsion, the gravitational field in galaxies turns out to be increased: from (142), we see that, in the case of the effective approximation (165), we get

$$R^{\rho\sigma} - \frac{1}{2}Rg^{\rho\sigma} - \Lambda g^{\rho\sigma} = \frac{k}{2}\left( E^{\rho\sigma} - \frac{1}{2}\frac{X^2}{M^2}S^\mu S_\mu g^{\rho\sigma} \right) \tag{235}$$

showing that the spinor field with the torsionally-induced nonlinear interactions has an effective energy, which is written as the usual term plus a nonlinear contribution.

For this contribution, we have to recall that we are not considering a single spinor field, as we have done when in particle physics, but collective states of spinor fields, as it is natural to assume in cosmology, with the consequence that it is not possible to employ the re-arrangements we used before and thus $S^\mu S_\mu$ cannot be reduced. Generally, we do not know how to compute it, but, as the square of a density, it may be positive.

In Ref. [46], we have been discussing precisely what would happen if the spin density square happened to be positive, and we have found that the contribution to the energy would change the gravitational field as to allow for a constant behavior of the rotation curves of galaxies, discussing the value of the torsion–spin coupling constant that is required to fit the galactic observations.

The details of the calculations were based on the fact that in this occurrence and within the approximations of slow rotational velocity and weak gravitational field, the acceleration felt by a point-particle was given by

$$\text{div } \vec{a} \approx -m\rho - K^2\rho^2 \tag{236}$$

in which the Newton gravitational constant has been normalized and where $K$ is the effective value of the torsional constant, with constant tangential velocity obtained for densities scaling down as $r^{-2}$ in general. In the standard approach to dark matter, there are only Newtonian source contributions scaling down as $r^{-1}$ and so a modification to the density distribution has to be devised, and it is the well known Navarro–Frenk–White profile. In the presence of torsional corrections, the Newtonian profile suffices because, even if the density drops as $r^{-1}$, it is squared in the torsional correction and the $r^{-2}$ drop is obtained. These similarities suggest that the torsion correction might be what gives the Navarro–Frenk–White profile. After all, the NFW profile is obtained in $n$-body dynamics as those assumed here provided that the $n$ spinors interact through torsion in terms of some axial-vector simplified model.

Nor is the idea of modeling dark matter, through the NFW profile, unexpected in terms of torsion, since this is precisely what a specific type of effective theories does.

In quite recent years, there has been a shift of approach in looking for physics beyond the standard model, and in particular dark matter. The new way of tackling the issue is based on the idea of studying all types of effective interactions that can be put in a Lagrangian, and, among all of them, there is the axial-vector spin-contact interaction.

However, in even more recent years, this approach has been generalized, shifting the attention from the effective interactions to the mediated interactions, known as simplified models [47]. However, the story does not change, since among all these there is the axial-vector mediated term

$$\Delta \mathscr{L} = -g\overline{\chi}\gamma^\mu\boldsymbol{\pi}\chi B_\mu \tag{237}$$

where $\chi$ is the dark matter particle and $B_\mu$ is the axial-vector mediator, and where the structure of the interaction is that of the torsion–spin coupling, as it should be quite easily recognizable for the reader at this moment.

When the standard model has been acknowledged to need a complementation, we have been striving to have it placed within a more general model, which should have also contained some new physics, and in particular dark matter. It has been the constant failure in this project that prompted us to reverse the strategy, pushing us to look for simplified models, namely models that can immediately describe dark matter, or in general new physics, and leaving the task of including them, together with the standard model, into a more general model for later, and better, times. If we were, therefore, to see that the dark matter, or generally some new physics, were actually described by one of these simplified models, the following step would be to include it beside the standard model within a more general model, and at this point it should be clear what is our ultimate claim for this entire section.

Our claim is that, if such a simplified model is the one described by the axial-vector mediator, then we will need not look very far, as the general model would be torsion.

We next move to study a more direct effect about a cosmological situation [48,49].

The problem is quite simply the fact that the cosmological constant has a measured value that, in natural units, is about one hundred and twenty orders of magnitude off of the theoretically predicted one. Normally, this would have made physicists reject the theories in which its value is calculated, but those theories are quantum field theory and the standard model, being very successful otherwise.

Philosophers may argue that, in the face of a bad result disproving a theory, there can be no good result that can support it: the history of physics is loaded with examples of good agreements between observations and predictions that were based on theories



later seen to be false. In addition, in this specific situation, the bad agreement is not only bad, but it is the worst in all of physics ever. Nowadays, the common behavior would be to claim that this is not really a bad agreement, since new physics might intervene to make the agreement acceptable. It does not take very experienced philosophers to see that this argument could always be invoked to push the problems under the carpet of an even higher energy frontier, and when this frontier will be unreachable, the predictivity of the theory will be annihilated. In this work, we try to embrace a philosophic approach, or merely be reasonable, admitting that such a discrepancy is lethal.

As a consequence, it follows that all theories predicting contributions to the cosmological constant must be dramatically re-adjusted. As we said above, these are the theory of quantum fields, with cosmological constant contributions due to zero-point energy, and the standard model, with cosmological constant contributions given by the same mechanism that gives mass to all fields and that is the spontaneous symmetry breaking.

As for the contribution coming from the general theory of quantum fields in terms of the zero-point energies, we have to recall that the zero-point energies are the result of quantization implemented with commutation relationships. In a normal-ordered quantum theory of fields, or simply in the classical theory of fields, zero-point energy does not appear, and thus no further contribution arises in the cosmological constant.

Leaving us without zero-point energy, it becomes necessary to find a way to compute the Casimir force without using any zero-point energy. It is worth remembering that Casimir forces derived from van der Waals forces was indeed the very first way to describe this phenomenon in the original paper by Casimir and Polder. A more recent account can be found in [50]. See also [51,52].

As for the contribution of the standard model in terms of the mechanism of symmetry breaking, recall that, after the break-down of the symmetry, we have the generation of the masses of all particles interacting with the Higgs field plus that of an effective cosmological constant $\frac{1}{2}\lambda^2 v^4$ with a value around $10^{120}$ in natural units. If this term is to disappear, we need to vanish either $\lambda$ or $v$, but, as vanishing the former would imply no symmetry breaking, the only possibility is to vanish $v$ so that symmetry breaking can still occur though not spontaneously. We may look for a dynamical symmetry breaking.

To begin our investigation, the very first thing we want to do is remark that, as the reader may have noticed, we never treated the scalar field. The reason was merely to keep an already heavy presentation from being heavier.

Still, it is now time to put in some scalar field. The scalar field complementing the Lagrangian (166) gives

$$\mathcal{L} = -\tfrac{1}{4}(\partial W)^2 + \tfrac{1}{2}M^2 W^2 - \tfrac{1}{k}R - \tfrac{2}{k}\Lambda - \tfrac{1}{4}F^2 + $$
$$+ i\overline{\psi}\gamma^\mu \nabla_\mu \psi + \nabla^\mu \phi^\dagger \nabla_\mu \phi - $$
$$- X\overline{\psi}\gamma^\mu \boldsymbol{\pi}\psi W_\mu - \tfrac{1}{2}\Xi\phi^2 W^2 - Y\overline{\psi}\psi\phi - $$
$$- m\overline{\psi}\psi + \mu^2\phi^2 - \tfrac{1}{2}\lambda^2\phi^4 \tag{238}$$

where the $X$, $\Xi$, $Y$ are the constants related to torsion with spinor and scalar interactions.

It is quite interesting to notice that, within this complementation, there is also the term $\phi^2 W^2$ coupling torsion to the scalar. This may look strange, since torsion is supposed to be sourced by the spin density, which is equal to zero for scalar fields. Therefore, we should expect to have torsion without a pure source of scalar fields, although we will have scalar contributions in torsion field equations.

In fact, upon variation of the Lagrangian, we obtain

$$\nabla_\alpha(\partial W)^{\alpha\nu} + (M^2 - \Xi\phi^2)W^\nu = X\overline{\psi}\gamma^\nu \boldsymbol{\pi}\psi \tag{239}$$

in which there is indeed a scalar contribution, although in the form of an interaction giving an effective mass term.

There is, immediately, something rather striking about this expression: in a cosmic scenario, for a universe in an FLRW metric, we would have that the torsion, to respect the same symmetries of isotropy and homogeneity, would have to possess only the temporal component. However, in this case, the dynamical term would disappear leaving

$$(M^2 - \Xi\phi^2)W^\nu = X\overline{\psi}\gamma^\nu\boldsymbol{\pi}\psi \tag{240}$$

as the torsion field equations we would have had in the effective limit, though now the result is exact. The source would have to be the sum of the spin density of all spinors in the universe, and because the spin vector points in all directions, statistically the source vanishes too and

$$(M^2 - \Xi\phi^2)W^\nu = 0 \tag{241}$$

which tells us that, if torsion is present, then

$$M^2 = \Xi\phi^2 \tag{242}$$

and, if $\Xi$ is positive, the scalar acquires the value

$$\phi^2 = M^2/\Xi \tag{243}$$

which is constant throughout the universe.

A constant scalar all over the universe is the condition needed for slow-roll in inflationary scenarios, and in this case there arises an effective cosmological constant

$$\Lambda_{\text{effective}} = \Lambda + \tfrac{1}{2}\left|\tfrac{\lambda}{2}\left|\tfrac{M}{\Xi}\right|^2\right|^2 \tag{244}$$

in the Lagrangian (238), driving the scale factor of the FLRW metric and therefore driving the inflation itself. Inflation will last, so long as symmetry conditions hold, but as the universe expands and the density of sources decreases, local anisotropies are no longer swamped, and their presence will spoil the symmetries that engaged the above mechanism, bringing inflation to an end [53].

As the universe expands in a non-inflationary scenario, the torsional field equation would no longer lose the dynamic term due to the symmetries, but it might still lose it due to the fact that massive torsion can have an effective approximation. In this case, we would still have

$$(M^2 - \Xi\phi^2)W^\nu \approx X\overline{\psi}\gamma^\nu\boldsymbol{\pi}\psi \tag{245}$$

although only as an approximated form. We may plug it back into the initial Lagrangian (238) obtaining

$$\begin{aligned}
\mathscr{L} = &-\tfrac{1}{k}R - \tfrac{2}{k}\Lambda - \tfrac{1}{4}F^2 + i\overline{\psi}\gamma^\mu\nabla_\mu\psi + \nabla^\mu\phi^\dagger\nabla_\mu\phi + \\
&+ \tfrac{1}{2}X^2(M^2 - \Xi\phi^2)^{-1}\overline{\psi}\gamma^\nu\psi\overline{\psi}\gamma_\nu\psi - \\
&- Y\overline{\psi}\psi\phi - m\overline{\psi}\psi + \mu^2\phi^2 - \tfrac{1}{2}\lambda^2\phi^4
\end{aligned} \tag{246}$$

as the resulting effective Lagrangian. The presence of an effective interaction involving spinors and scalars, having a structure much richer than that of the Yukawa interaction, is obvious. In addition, we observe that, if for vanishingly small scalar this reduces to the above effective interaction for spinors, in the presence of larger values for the scalar, it can even become singular. We might speculate that such a value is the maximum allowed for the scalar as the one at which the above mechanism of inflation takes place.

Consider now the case $\mu = 0$ in the above Lagrangian.

The scalar potential is minimized by $\phi^2 = v^2$ such that

$$\lambda^2 v^2 = \tfrac{1}{2}\Xi X^2 (M^2 - \Xi v^2)^{-2}|\overline{\psi}\gamma^\nu\psi\overline{\psi}\gamma_\nu\psi|_{\rm v} \tag{247}$$

linking the square of the Higgs vacuum to the square of the density of the spinor vacuum. Therefore, the dynamical symmetry breaking mechanism occurs eventually.

This break-down of symmetry is a dynamical one because the vacuum is not a constant, but it is the vacuum expectation value of the spinor distribution.

After dynamical symmetry breaking is implemented in the Lagrangian, the effective cosmological constant is still proportional to the Higgs vacuum, but the Higgs vacuum is now proportional to the spinor vacuum. Where material distributions tend to zero, as we would have in cosmology the vacuum for the spinor trivializes, the vacuum for the Higgs trivializes as well and the cosmological constant is no longer generated [54].

The picture that emerges is one for which symmetry breaking is no longer a mechanism that happens throughout the universe but only when spinors are present, with the consequence that, if spinors are not present, the effective cosmological constant is similarly not present. The cosmological constant due to spontaneous symmetry breaking in the standard model is thus avoidable.

No zero-point energy leaves no contribution apart from those due to phase transitions, which can be quenched by a symmetry breaking that is not spontaneous but dynamical, and no effective cosmological constant arises.

In this third chapter, we presented and discussed the possible torsional dynamics in the cosmology and particle physics standard models. Now, it is time to pull together all the loose ends in order to display the general overview.

We have seen and stated repeatedly that torsion can be thought as an axial-vector massive field coupling to the axial-vector bi-linear spinor field according to the term

$$\Delta\mathscr{L}^{\rm Q-spinor}_{\rm interaction} = -X\overline{\psi}\gamma^\mu\boldsymbol{\pi}\psi W_\mu \tag{248}$$

of which we have one for every spinor. Effective approximations involving two, three, or even more spinor fields have been discussed, with a particular care for the case of neutrino oscillations, for which we have detailed in what way the results of [42] can be generalized in order to have

$$\Delta\Phi \approx \tfrac{L}{2E}\left(\Delta m^2 + m\tfrac{X^2_{\rm eff}}{4M^2}|\overline{\nu}\gamma_\mu\nu\overline{\nu}\gamma^\mu\nu|^{\frac{1}{2}}\right) \tag{249}$$

describing the phase difference for almost degenerate neutrino masses, as consisting of the $L/E$ dependence modulating the usual kinetic contribution, plus a new dynamic contribution, so that, even if the neutrino mass spectrum were to be degenerate, torsion would still induce an effective mechanism of oscillation. As these considerations have nothing special about neutrinos, and thus they may as well be extended to all leptons, we then proceeded in studying such extension. However, once the Lagrangian terms of the weak interaction after symmetry breaking and the torsion for an electron and a left-handed neutrino were taken in the effective approximation, we saw that, due to the cleanliness of the scattering and the precision of the measurements, the standard model correction induced by the torsion had to be very small, and, if this occurs because the torsion mass is large, then the effective approximation is no longer viable. We have then re-considered the case without effective approximations, allowing also for sterile right-handed neutrinos in order to maintain the feasibility of the dynamical neutrino oscillations discussed above, therefore reaching the general Lagrangian

$$\begin{aligned}\Delta\mathscr{L}^{\rm Q/weak\text{-}spinor}_{\rm interaction} = {}&-X_e\overline{e}\gamma^\mu\boldsymbol{\pi}eW_\mu - X_\nu\overline{\nu}\gamma^\mu\boldsymbol{\pi}\nu W_\mu + \\ &+ \tfrac{g}{\sqrt{2}}\left(W^-_\mu\overline{\nu}\gamma^\mu e_L + W^+_\mu\overline{e}_L\gamma^\mu\nu\right) + \\ &+ \tfrac{g}{\cos\theta}Z_\mu[\tfrac{1}{2}(\overline{\nu}\gamma^\mu\nu - \overline{e}_L\gamma^\mu e_L) + |\sin\theta|^2\overline{e}\gamma^\mu e]\end{aligned} \tag{250}$$

showing that, while the sterile right-handed neutrino is by construction insensitive to weak interactions, it is sensitive to the universal torsion interaction, suggesting that, to see torsional interactions, we must pass for neutrino physics.

After having extensively wandered in the microscopic domain of particle physics, we move to see what type of effect torsion might have for a macroscopic application of a yet unseen particle, dark matter, and we have seen that, in the case of effective approximation, the spinor source in the gravitational field equations becomes of the form

$$R^{\rho\sigma} - \tfrac{1}{2}Rg^{\rho\sigma} - \Lambda g^{\rho\sigma} = \tfrac{k}{2}\left(E^{\rho\sigma} - \tfrac{1}{2}\tfrac{X^2}{M^2}S^\mu S_\mu g^{\rho\sigma}\right) \tag{251}$$

showing that, if the spin density square happens to be positive, the contribution to the energy would change the gravitational field as to allow for a constant behavior of the rotation curves of galaxies. We have discussed that this behavior comes from having a matter density scaling according to $r^{-2}$ for large distances. Such behavior, usually, is granted by the Navarro–Frenk–White profile or, here, is due to the presence of torsion, suggesting that the NFW profile is just the manifestation of torsional interactions, and ultimately that dark matter may be described in terms of the axial-vector simplified model, sorting out one privileged type among all possible simplified models now in fashion in particle physics. Then, we proceeded to include into the picture also the scalar fields, getting

$$\begin{aligned}
\mathscr{L} = &-\tfrac{1}{4}(\partial W)^2 + \tfrac{1}{2}M^2 W^2 - \tfrac{1}{k}R - \tfrac{2}{k}\Lambda - \tfrac{1}{4}F^2 + \\
&+ i\overline{\psi}\gamma^\mu \nabla_\mu \psi + \nabla^\mu \phi^\dagger \nabla_\mu \phi - \\
&- X\overline{\psi}\gamma^\mu \pi \psi W_\mu - \tfrac{1}{2}\Xi \phi^2 W^2 - Y\overline{\psi}\psi\phi - \\
&- m\overline{\psi}\psi + \mu^2 \phi^2 - \tfrac{1}{2}\lambda^2 \phi^4
\end{aligned} \tag{252}$$

showing that, in general, the torsion, besides its coupling to the spinor, may also couple to the scalar, with the scalar behaving as a sort of correction to the mass of torsion and a kind of re-normalization factor in the torsion–spinor effective interactions. We discussed how, within a homogeneous isotropic universe, the torsion field equations grant the condition $M^2 = \Xi \phi^2$ so that, if $\Xi$ were positive, then the scalar field would acquire a constant value, slow-roll will take place, and inflation could engage. After inflation has ended, torsion contributions to the scalar sector may induce a dynamical symmetry breaking. This may solve the cosmological constant problem in a new manner.

## 8. Conclusions

In this review, we have constructed the most general geometry with torsion as well as curvature, and, after having also introduced gauge fields in a similarly geometric manner, we have also built a genuinely geometric theory of spinor fields. We have seen how, under the assumption of being at the least-order derivative, the most general fully covariant system of field equations has been found for all physical fields in interaction. Separating torsion from all other fields and splitting spinor fields in their irreducible components allowed us to better see that torsion can be seen as an axial-vector massive field mediating the interaction between chiral projections. A formal integration of the spinorial degrees of freedom has also been discussed in some detail. Studying special situations, we have seen that the torsionally-induced spin-contact interactions are attractive. In addition, we have examined the conditions under which they can be removed with a choice of chiral gauge.

We have then seen that torsional effects for spinor fields can give rise to the conditions for which the gravitational singularities are no longer bound to form. We have hence established a parallel to the instance of the Pauli exclusion principle. We have discussed the problem of positive energies for spinors. In addition, we have determined the conditions under which positivity of the energy can be ensured.

Eventually, we have discussed problems inherent to the standard model of particles, specifically for neutrino oscillations and dark matter. In addition, we have commented on a possible solution to the cosmological constant problem.

In a geometry which, in its most general form, is naturally equipped with torsion, and for a physics which, for the most exhaustive form of coupling, has to couple the spin of matter, the fact that torsion couples to the spin of spinor material field distributions is just as well suited as a coupling can possibly be. In addition, its consequences about the stability of such field distributions are certainly worth receiving further attention.

In the standard model of particles, there are different facts to consider. Assuming the existence of right-handed sterile neutrinos, the torsion–spin coupling gives dynamic corrections to the oscillation pattern that can become important contributions if we were to see that the neutrino masses were too close to one another to fit the observed patterns. By assuming dark matter as constituted by a form of matter with spinorial structure, the torsion field, with its axial-vector massive character, might give rise to the known NFW profile. In cosmology, the most urgent of the problems is that of the cosmological constant, which can be solved, or at least quenched, by a theory in which spontaneous symmetry breaking is replaced by dynamical symmetry breaking, as is the case when torsion is allowed to interact through spinors with the scalar.

In the first two instances, that is, for the case of neutrino oscillations and dark matter, the new contributions are condensed in $\Delta\mathscr{L} = -X\overline{\psi}\gamma^{\mu}\boldsymbol{\pi}\psi W_{\mu}$ as axial-vector coupling. In the last instance that is for the dynamical symmetry breaking, new physics are described in terms of the $\Delta\mathscr{L} = -X\overline{\psi}\gamma^{\mu}\boldsymbol{\pi}\psi W_{\mu} - \frac{1}{2}\Xi\phi^2 W^2$ contribution as what gives the torsion–spin and scalar interaction. These two potentials for torsion are the only two potentials that can be added into the Lagrangian within the restriction of renormalizability.

It is important to call attention to the fact that, as the general presentation goes, there are two ways to have torsion in differential geometry: the first is the one followed here, and it is the one more mathematical in essence, based on the general argument that torsion is present simply because there is no reason to set it to zero. The second one is more physical, based on the argument that torsion *must* be present since it necessarily arises after gauging translations much in the same way in which curvature is present as it arose after gauging rotations. In this approach, torsion and curvature are the Yang–Mills fields that inevitably emerge because we are considering the gauge theory of the full Poincaré group [55]. The usefulness of this approach has been remarkable in addressing problems related to supersymmetry, and especially supergravity. For more details, we refer the reader to [56,57], and in particular [58]. More recent papers that deal with torsion-gravity as a gauge theory are [59–61]. Still important is the work in [62].

This latter approach of gauging the full Poincaré group is based on the tetradic formalism, on which an overview by Tecchiolli can be found in this Special Issue [63].

Readers might not have failed to notice that there is a great absent in the presentation: field quantization, and there are reasons for it to be so. In spite of all successful predictions and precise measurements, it would not be a proper behavior to deny all mathematical problems the theory of quantum fields still has. From the fact that the equal-time commutator relations may not make sense at all [64] to the non-existence of interaction pictures [65], to mention only the most important of the problems, the rigorous mathematical treatment of quantum fields is yet to be achieved. What can torsion do for this? Honestly, it seems unlikely that any change in the field content can change things for the general structure of the theory from its roots. However, it may still be possible that, after all, torsion could address problems appearing later on in the development of the theory. For example, we have already discussed how torsion could be responsible for avoiding singularities in the case of spinor fields. This tells us that torsion may similarly be responsible for the fact that the elementary particles might not be point-like. If this were the case, then torsion would certainly have something to say about the problem of ultraviolet divergences. Torsion may be what gives a physical meaning to regularization and normalization, with the torsion mass giving the scale of the physical cut-off. There is still quite a way to go in fixing, or at least alleviating, the problems of the quantum field theoretical approach. It does not look unreasonable, however, that torsion might be there to help again.

Then, there are all the possible extensions. To begin, there is the fact that we wrote all field equations under the constraint of being at the least-order derivative possible. This requirement also coincides with renormalizability for all equations except those for gravity. If we wish to have renormalizability for all equations, then the gravitational field equations must be taken at the fourth-order derivative in the metric tensor [66]. This is certainly an opportunity for further research, especially for the effects torsion may have for the problems of singularity formation and positive energies. Another point needing some strengthening is related to the fact that torsion has always been taken completely antisymmetrically. Such a symmetry was duly justified in terms of fundamental arguments, and, for that matter, the completely antisymmetric torsion couples neatly with the completely antisymmetric spin of the Dirac field. However, if higher-spin fields were to be found or more general geometric backgrounds were to be needed, more general torsion would make their appearance in the theory. Studying what may be the role of a trace torsion or of the remaining irreducible torsion component is also an important task.

Regarding the mass generation of spinors through spin–torsion interactions, it is necessary to direct the attention toward [67–69], and recently [70].

Again, a recent work is that of Diether and Christian later in this Special Issue [71].

In particle physics more in general, high-energy experimental constraints on torsion have been placed, especially in [72–76]. Going to cosmology, dark matter has also been studied in the presence of torsion: after the already-mentioned [45], the reader may find it interesting to also have a look at [77–81]. As for the problem of singularity formation during the Big Bang, the following references may be of help [82–86].

More mathematical extensions have been addressed along the years in various manners. In fact, all possible alternatives and extensions of Einstein gravity can also be generalized for the torsional case: for instance, conformal gravity with torsion has been established in [87], while $f(R)$-types of torsion-gravity have been studied in [88]. As for the latter, the reader may also find the problem of junction conditions interesting [89].

For a review of such a problem, we invite the reader to the paper by Vignolo that can also be found later in this Special Issue [90].

From a purely mathematical, general point of view, interesting features of the torsional background in the presence of spinors have been investigated in [91,92].

The dynamics of the torsion field may also in principle allow the propagation of parity violating modes, although many constraints have been placed recently [93,94].

Anomalies and constraints on torsion were studied in [95,96].

The possibilities introduced by not neglecting torsion in gravity for Dirac fields can also be more mathematical in essence. Above all, it is paramount to mention all the exact solutions for the coupled system of field equations that may be found. In [97–99], we found exact solutions for the Dirac field in its own gravitational field. Including torsion in gravity and allowing the coupling to the spinor can only increase the interesting features that exact solutions could have. Of course, finding exact solutions for a system of interacting fields is a very difficult enterprise and so we must expect a slow evolution.

It is clearly impossible to draw a complete list of references. Nevertheless, those presented here, and their own references, might be taken as a fair list to help the reader.

We wish to conclude this exposition with one personal note on aesthetics. It is very often stated in philosophical debates that a theory is considered to be beautiful when it has some sense of inevitability built into itself. That is, a sense for which there is nothing that can be modified, or removed, from the theory without looking like a form of unnecessary assumption. We see torsion gravity precisely like that. Such a theory is formed by requiring that some very general principles of symmetry be respected for the four-dimensional continuum space–time. If these hypotheses are put in, they determine the development of the theory without anything else to be postulated. It is at this point therefore that a theory of gravity with no torsion, where we would remove an object that would otherwise be naturally present, has to be regarded as something arbitrary.

The most beautiful, in the sense of the most necessary, theory of gravity is the one in which torsion is allowed to occupy the place that is its own a priori.

**Funding:** This research received no external funding.

**Institutional Review Board Statement:** Not applicable.

**Informed Consent Statement:** Not applicable.

**Data Availability Statement:** Not applicable.

**Acknowledgments:** In the last sixteen years, the single constant source of material (and more importantly immaterial) support has been Marie-Hélène Genest, my wife. It is to her that I give my continuous and everlasting thanks for everything.

**Conflicts of Interest:** The author declares no conflict of interest.

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
