# Peer review of "Fundamental Theory of Torsion Gravity"

_universe, doi:10.3390/universe7080305_

Round 1
Reviewer 1 Report
The author has included the references suggested in the previous report and the conclusions are now in better shape.
Nevertheless, the author did not address the first comment of the previous report successfully. Indeed, Section 2 still has several unnumbered equations, repeated equations, and unnecessary details that yields the reader to miss the big picture. For instance:
- Equation (120) is repeated in (124), below line 947 modulo a redefinition, and below line 957.
- Equation (123) is repeated in (125) and in (130) by changing k->a.
- Unnumbered equations between lines 773 and 777 are repeated between lines 952 and 955.
- Equation (121) is repeated in (127), while Eq. (132) is repeated in (135).
- Equation (126) is somehow included in Eq. (131) by including Maxwell fields into the definition of the energy-momentum tensor.
- Equation below line 1052 is repeated below line 1061, below line 1089 by identifying properly J^\mu with the spinorial current, below line 1109 and 1118 by a redefinition of coupling constants.
This list is far from being exhaustive and the author should identify and fix all these issues throughout Section 2.
Additionally, there are several unnecessary details in the derivation of the field equations once the torsion is replaced by the spin density of matter. This fact renders Sec. 2 difficult to read. For instance, it seems to me that it is redundant to write equation below 996 as the trace of the equation below 995: One can just mention it in the text. Moreover, once the tensor below line 1011 is defined, there is no need to write explicitly the unnumbered equation below line 1013, since the reader can refer to the unnumbered equation below 960 by replacing the former. Again, this list is not exhaustive and one finds these issues in different places throughout Section 2.
I recommend rewriting this part completely, keeping essential equations only, namely, field equations Eqs. (120) and (123), definitions of spin density and energy-momentum tensor only once, implications of Bianchi identities on the matter sector through Eqs. (121) and (122), main implications of the Velo-Zwanziger method, and include Maxwell fields from the beginning to avoid repetition.
Until these changes are implemented, I cannot recommend this manuscript for publication.
Reviewer 2 Report
See attached file.

Round 2
Reviewer 1 Report
The author addressed all the issues raised in the previous report. I recommend the article for publication.
This manuscript is a resubmission of an earlier submission. The following is a list of the peer review reports and author responses from that submission.
Round 1
Reviewer 1 Report
In my opinion, the manuscript is too long. It is overloaded with well-known and many new but less interesting formulas. The presentation wishes to be better. The author introduces his own notations for well-known quanatities. For example, instead of the Dirac matrix γ5 the author uses "π" that should not be useful for a reader. Then, it is well-known that torsion, taken in its general form, violates gauge invariance of the electromagnetic field. Gauge invariance can only be preserved by a 4-vector part of torsion, when such a 4-vector is a gradient of a scalar field. This means that gravitational theory with torsion, taken in its general form, and gauge fields should not be gauge invariant. I am sorry, but I cannot recommend this manuscript for publication in Universe.
Reviewer 2 Report
Perhaps I could begin by making a comment about the subject of the review. The vierbein formulation of general relativity may go back to Weyl in the 1920's but it was, I believe, Sciama and then Kibble who introduced the spin connection which they took to transform as a Yang-Mills field for the Lorentz group. Kibble then considered the translations of the Poincare group $x^\mu\to x^\mu + \xi^\mu$ and took the constant parameter $\xi^\mu$ to be a function of spacetime $\xi^\mu(x)$. In this way one finds a general coordinate transformation as introduced by Einstein. One can, as Kibble did, introduce a field, the vierbein, that allows one to construct invariants. In this way one finds a theory in which torsion can appear. However it is not what Yang-Mills did and so it is not the gauge theory of the Poincare group. Instead it starts with general coordinate transformations.
\par
The gauge theory of the Poincare group was first introduced in
A. Chamsedine and P. West, Supergravity as a Gauge Theory of Supersymmetry; Nucl. Phys. B129, 39 (1977).
This paper considered the Poincare group and introduced the corresponding gauge fields $e_\mu{}^a$ and $w_\mu{}^{ab}$ which {\bf transformed as Yang-Mills fields for the Poincare group}. Actually this paper took the larger super Poincare group but it was the first time that Einstein gravity had been derived from the gauging of the Poincare view point. There were a subsequent papers by Mac Dowell and Mansouri and one by Stelle and West which the authors do reference. This gauge theory of the Poincare group approach has been of great use. The paper by Chansedine and West contained the first algebraic proof that supergravity theories were actually invariant under supersymmetry, lead to the construction of conformal supergravity theories and is the technique that is used extensively today to construct higher spin theories.
\par
Thus we have two different approaches to constructing theories of gravity. Only the latter takes the gauge theory of the Poincare group as a Yang-Mills theory. However, it is the former approach that has so often been referred to in this way.
\par
The review follows the first approach but like so much of the literature it very briefly mentions the above two papers but regards the second approach as part of the first approach so confuses the two approaches. The authors should add a section to their review that makes clear what are two approaches and why they are different. It should also make clear that the subject of their review is the first approach. Should they do this it could well form a useful addition to the literature.
Reviewer 3 Report
In this manuscript, the author reviews the Einstein-Cartan-Sciama-Kibble (ECSK) theory of gravity coupled with matter fields, specifically with fermions, and extensions thereof. The extension is based upon the most general action for torsion such that the theory remains the least order in derivatives. The spin density of the latter acts as a source of torsion and induces a spin-spin interaction that modifies the energy-momentum tensor. This interaction resembles the Nambu-Jona-Lasinio model and it can be used to address current problems in gravitation and particle physics.
The first part is devoted to present the geometrical framework underlying the theory: the Riemann-Cartan geometry. Then, the Lorentz and U(1) internal groups were introduced alongside spinorial representations. The interaction between geometry and matter is also discussed in different limits: massless, massive, non-relativistic, and ultra-relativistic.
The author comment on the possibility that torsion could cure some problems arising in general relativity and the standard model of particle physics, e.g. formation of singularities, nature of dark matter, neutrino oscillations, and the nature of the cosmological constant through a dynamical symmetry breaking.
In general terms, I believe that the review gathers different interesting topics of torsion and their interaction with matter. The manuscript could be published after some moderate changes are applied. These are given next.
First, Section 1.2 possesses several unnumbered and repeated equations. For instance, the equation below line 781 is repeated below lines 783, 826, 1003 (it is just 781 in vacuum), and so on; equation below line 825 is repeated below lines 831, 862 (replacing k by a), 880, etc; equations below lines 812 and 813 are repeated below lines 834, 835, 842, 843, 865, 866, so on and so forth. Indeed, the previous list is not exhaustive and there are still several unnumbered and repeated equations. Due to this, Sec. 1.2 is endowed with several unnecessary equations rendering the manuscript difficult to read. I recommend rewriting this part completely, leaving essential equations only, writing them once, and referring them throughout the text when necessary.
Second, although the manuscript aims to review the current status of torsion-matter interaction, the examination of the literature is far from being exhaustive. Since this is a review article, the latter is mandatory. The following references should be added to the main text.
Regarding the mass generation of fermions through torsion interaction, the author missed the following references
- Mod. Phys. Lett. A 25 (2010) 2885, (arXiv:1003.5473), M. A. Zubkhov,
- Mod. Phys. Lett. A 29 (2014) 1450111, (arXiv:1310.8034), by M. A. Zubkov,
- JHEP 1309 (2013) 044, (arXiv:1301.6971), by M. A. Zubkhov,
- Phys. Rev. D 88 (2013) 124022, (arXiv:1310.4124) by O. Castillo-Felisola, C. Corral, C. Villavicencio, A. R. Zerwekh.
Cosmology with spin density in the framework of ECSK theory has been studied in
- Phys. Rev. D 87 (2013) 6, 063504, (arXiv:1212.0585), J. Magueijo, T. G. Zlosnik, and T. W. B. Kibble.
- Phys. Rev. D 90 (2014) 12, 123510, (arXiv:1402.5880), S. Alexander, C. Bambi, A. Marciano, and L. Modesto.
- Astrophys. J. 870 (2019) 2, 78, (arXiv:1808.08327), G. Unger and N. Poplawski.
- Phys. Rev. D 96 (2017) 8, 083530, (arXiv:1709.03171), S. Farnsworth, J-L., Lehners, and T. Qiu.
Particle physics constraints on torsion gravity have been explored in
- Phys. Rev. D 72 (2005) 104002, (arXiv:hep-th/0507253), L. Freidel, D. Minic, and T. Takeuchi.
- Phys. Rev. D 90 (2014) 12, 124068, (arXiv:1410.6197), Y. N. Obukhov, A. J. Silenko, and O. V. Teryaev.
- Mod. Phys. Lett. A 29 (2014) 1450081, (arXiv:1404.5195), O. Castillo-Felisola, C. Corral, I. Schmidt, and A. R. Zerwekh.
- Phys. Rev. D 90 (2014) 2, 024005, (arXiv:1405.0397), O. Castillo-Felisola, C. Corral, S. Kovalenko, and I. Schmidt.
- Phys. Rev. D 97 (2018) 7, 075036, (arXiv:1712.09632), F. M. L. De Almeida, F. R. De Andrade, M. A. B. do Vale, and A. A. Nepomuceno.
In the context of dark matter, different proposals have been put forward; see for instance:
- Phys. Rev. D 95 (2017) 9, 095033, (arXiv:1611.03651), A. S. Belyaev, M. C. Thomas, and I. L. Shapiro.
- JHEAp 13-14 (2017) 10-16 (arXiv:1611.02137), D. Alvarez-Castillo, D. J. Cirilo-Lombardo, and J. Zamora-Saa.
- Gen. Rel. Grav. 43 (2011) 2965-2978, (arXiv:1104.0160) A. Tilquin and T. Schucker.
- Phys. Rev. D 91 (2015) 8, 085017, (arXiv:1502.03694), O. Castillo-Felisola, C. Corral, S. Kovalenko, I. Schmidt, and V. E. Lyubovitskij.
- Gen. Rel. Grav. 53 (2021) 4, 49 (arXiv:2001.02159), G. Grensing.
- Phys. Rev. Lett. 126 (2021) 16, 161301, (arXiv:2008.11686), M. Shaposhnikov, A. Shkerin, I. Timiryasov, and S. Zell.
These and other references the author may find relevant should be included in the next version of this manuscript.
Until these changes are not implemented into the main text, I cannot recommend the article for publication.